# LoRA Meets Second-Order Optimization: Towards Optimal Low-Rank Updates

## Abstract

Low-rank fine-tuning is widely applied for the effective adaptation of large models. Most existing methods rely on low-rank matrix factorization, whose performance is limited by the condition number of the associated Jacobi operator. Although these methods are computationally efficient, their performance still falls short compared to full fine-tuning. To address this, we propose SoLoRA, which leverages an adaptive metric to find a low-rank approximation of the full fine-tuning gradient. This low-rank approximation can be viewed as an approximation of Hessian, effectively incorporating second-order information to achieve faster convergence and higher optimization efficiency. Furthermore, the low-rank approximation in SoLoRA is computationally simple and easy to implement, achieving a close approximation to the performance of full fine-tuning with almost no additional computational overhead. We conduct fine-tuning experiments on large language models and diffusion models, and the results consistently demonstrate that SoLoRA achieves superior performance advantages over state-of-the-art low-rank fine-tuning methods.

## 1 Introduction

Large language models (LLMs) (Liu et al., 2024a; Yang et al., 2024) and vision-language models (Achiam et al., 2023) have demonstrated outstanding performance in various applications, such as chatbot, image generation, and editing. With their strong generalization capabilities and versatility, they have been widely adopted for a range of downstream tasks.To better adapt LLMs to specific downstream tasks, it is often necessary to fine-tune their parameters. However, full fine-tuning is evidently expensive, incurring significant computational and storage costs. To address this, parameter-efficient fine-tuning (PEFT) has emerged to reduce the overhead of fine-tuning.

Low-Rank Adaptation (LoRA) (Hu et al., 2022) is a representative PEFT method. It assumes that weight updates during fine-tuning exhibit a low "intrinsic rank". By freezing the pretrained weights and introducing two low-rank matrices, $\boldsymbol{B} \in \mathbb{R}^{m \times r}$ and $\boldsymbol{A} \in \mathbb{R}^{r \times n}$, for updates, LoRA reduces the number of trainable parameters. Compared to full fine-tuning, the number of trainable parameters in LoRA is $\mathcal{O}((m+n)r)$, where $r \ll \{m, n\}$, significantly lowering the number of trainable parameters, memory consumption, and fine-tuning costs. Owing to these advantages, LoRA and its numerous variants (Hu et al., 2022; Hayou et al., 2024; Zhang and Pilanci, 2024; Wang et al., 2024; Zhao et al., 2024; Zhu et al., 2024; Wang et al., 2025; Mo et al., 2025; Zhang et al., 2025b) have been widely applied in practical applications.

Although LoRA offers significant advantages, most existing fine-tuning algorithms are based on a factorization framework that updates the two low-rank factors separately. Such factorization-based methods are sensitive to the condition number of the low-rank factors, which can result in slow convergence. ScaledGD (Tong et al., 2021; Zhang and Pilanci, 2024) addresses this issue by introducing two preconditioners, effectively eliminating the dependency on the condition number and making its convergence rate condition-number-independent. However, ScaledGD still suffers from parameter redundancy, and its fine-tuning efficiency falls short of matching that of full-parameter fine-tuning.

LoRA-Pro (Wang et al., 2025) demonstrates that applying gradients $\boldsymbol{G}_A$ and $\boldsymbol{G}_B$ to the low-rank factors $\boldsymbol{A}$ and $\boldsymbol{B}$ is equivalent to performing full fine-tuning on the weight matrix $\boldsymbol{W}$ with a low-rank gradient $\tilde{\boldsymbol{G}}$. Building on this insight, LoRA-Pro reduces the discrepancy between $\tilde{\boldsymbol{G}}$ and the full

fine-tuning gradient $G$ by solving the optimization problem $\min \|\tilde{G} - G\|_F^2$, thereby bridging the performance gap between LoRA and full fine-tuning. LoRA-Pro employs the standard metric inherited from the Euclidean space of the weight matrices to approximate $G$. However, approximation is often more effective under a weighted metric rather than the standard metric. For example, AdaGrad (Duchi et al., 2011) and SOAP (Vyas et al., 2025) leverage historical gradient information to adaptively adjust the step size of each gradient component, effectively utilizing weighted metrics in the Euclidean space of the weight matrix. K-FAC (Martens and Grosse, 2015; Eschenhagen et al., 2023) uses a weighted metric based on the Kronecker product to approximate the Hessian, thereby constructing an efficient preconditioner.

Inspired by this, we propose a novel algorithm called **S**econd-**O**rder **Lo**w-**R**ank **A**daption (**SoLoRA**), which aims to further narrow the performance gap between low-rank fine-tuning and full fine-tuning. SoLoRA leverages an adaptive metric derived from AdaGrad (Duchi et al., 2011) and SOAP (Vyas et al., 2025) to identify a low-rank approximation of the full fine-tuning gradient. Notably, this low-rank approximation can also serves as an approximation of the Hessian, enabling SoLoRA to effectively incorporate second-order information from the loss function for faster convergence. Moreover, the optimal low-rank approximation identified by SoLoRA does not directly depend on the full fine-tuning gradient, making SoLoRA simple and easy to implement. Experiments on GPT-2 and diffusion models demonstrate that SoLoRA, by adopting a weighted metric-based approximation, outperforms both standard metric-based approximations and existing low-rank fine-tuning methods, achieving superior performance.

## 2 Low-Rank Fine-Tuning of Large Language Models

In this section, we revisit existing low-rank fine-tuning methods from a fresh theoretical perspective, highlighting their gaps compared to full fine-tuning. Based on this analysis, we discuss the limitations of these low-rank fine-tuning algorithms and elucidate their fundamental distinctions.

### 2.1 Rethinking Low-Rank Fine-Tuning: Connections and Limitations

As a representative parameter-efficient fine-tuning method, low-rank fine-tuning works by freezing the pretrained weights $W_0 \in \mathbb{R}^{m \times n}$ and assuming that the weight update $W$ exhibits a low-rank structure during downstream task adaptation. Consequently, the adaptation process is formulated as a low-rank constrained optimization problem:

$$\min_{W \in \mathbb{R}^{m \times n}} \mathcal{L}(W_0 + W), \quad \text{subject to } \text{rank}(W) = r,$$

where $\mathcal{L}(\cdot)$ denotes the training loss function and $r \ll \min\{m, n\}$. Proximal gradient descent is a widely adopted method for solving the above low-rank optimization problem by updating the weight matrix via

$$W_{t+1} = \mathcal{H}_r(W_t - \alpha_t \nabla_{W_t} \mathcal{L}(W_0 + W_t)),$$

where $\mathcal{H}_r$ represents the $r$-truncated singular value decomposition (SVD) applied to each weight matrix, $\alpha_t$ is the learning rate of $W_t$. This requires performing SVD on every layer at each optimization step, which has a computational complexity $O(m^3)$, leading to high-computation cost.

LoRA and its variants (Hu et al., 2022; Wang et al., 2024; Hayou et al., 2024; Liu et al., 2024b; Wang et al., 2025; Zhang et al., 2025b; Yen et al.; Zhang et al., 2025a) train the network directly via a low-rank factorization, thereby avoiding the expensive SVD computation at each training step. These methods aim to solve the following non-convex optimization problem based on the factorization:

$$\min_{W \in \mathbb{R}^{m \times n}} \mathcal{L}(W_0 + W), \quad \text{subject to} \quad W = BA,$$

where $B \in \mathbb{R}^{m \times r}, A \in \mathbb{R}^{r \times n}$. Here, we define $\mathcal{G}([B, A]) = W$ as a generator that constructs weight matrices from the low-rank factors. Under this definition, the optimization problem can be reformulated as:

$$\min_{B \in \mathbb{R}^{m \times r}, A \in \mathbb{R}^{r \times n}} \mathcal{L}(W_0 + \mathcal{G}([B, A])).$$

For factorization-based gradient algorithms, the updates can be expressed as follows, leveraging the chain rule:

$$[B_{t+1}, A_{t+1}] = [B_t, A_t] - \eta_t J_{\mathcal{G}}^*([B_t, A_t]) \nabla_{W_t} \mathcal{L}(W_0 + \mathcal{G}([B_t, A_t])), \tag{1}$$

where $J_{\mathcal{G}}^*$ is the adjoint of the Jacobian operator of $\mathcal{G}$ and $\eta_t$ is the learning rate of $\boldsymbol{B}_t$ and $\boldsymbol{A}_t$. To further analyze the gap between low-rank fine-tuning and full fine-tuning, we return to the update of the weight matrix $\boldsymbol{W}$. By applying the generator operator $\mathcal{G}$ to both sides of (1), we get

$$\mathcal{G}([\boldsymbol{B}_{t+1}, \boldsymbol{A}_{t+1}]) = \mathcal{G}\Big([\boldsymbol{B}_t, \boldsymbol{A}_t] - \eta_t J_{\mathcal{G}}^*([\boldsymbol{B}_t, \boldsymbol{A}_t])\nabla_{\boldsymbol{W}_t}\mathcal{L}\big(\boldsymbol{W}_0 + \mathcal{G}([\boldsymbol{B}_t, \boldsymbol{A}_t])\big)\Big).$$

To facilitate comparison with the gradient descent algorithm based on the weight matrix $\boldsymbol{W}$, we perform a Taylor expansion around $[\boldsymbol{B}_t, \boldsymbol{A}_t]$,

$$\boldsymbol{W}_{t+1} \approx \boldsymbol{W}_t - \alpha_t J_{\mathcal{G}}([\boldsymbol{B}_t, \boldsymbol{A}_t]) J_{\mathcal{G}}^*([\boldsymbol{B}_t, \boldsymbol{A}_t])\nabla_{\boldsymbol{W}_t}\mathcal{L}(\boldsymbol{W}_0 + \boldsymbol{W}_t), \tag{2}$$

where $J_{\mathcal{G}}([\boldsymbol{B}_t, \boldsymbol{A}_t])[\cdot, \cdot] : [\mathbb{R}^{m \times r}, \mathbb{R}^{r \times n}] \to \mathbb{R}^{m \times n}$ is the Jacobian operator. From this update form, it becomes clear that, compared with full fine-tuning, a key limitation of low-rank fine-tuning lies in the explicit dependence of the factor gradients on $J_{\mathcal{G}}J_{\mathcal{G}}^*$, whose condition number is determined by the condition numbers of the low-rank factors $\boldsymbol{B}$ and $\boldsymbol{A}$ (Chen et al., 2019; Chi et al., 2019). This dependency introduces potential instability during training, particularly when fine-tuning complex neural networks or large language models, which often results in performance degradation (Hayou et al., 2024; Zhang and Pilanci, 2024).

## 2.2 Preconditioned Low-Rank Adaption Fine-Tuning

Under the widely adopted generator form $\mathcal{G}([\boldsymbol{B}, \boldsymbol{A}]) = \boldsymbol{B}\boldsymbol{A}$, the Jacobian operator $J_{\mathcal{G}}([\boldsymbol{B}_t, \boldsymbol{A}_t])[\cdot, \cdot] :$ $[\mathbb{R}^{m \times r}, \mathbb{R}^{r \times n}] \to \mathbb{R}^{m \times n}$ and its adjoint operator $J_{\mathcal{G}}^*([\boldsymbol{B}_t, \boldsymbol{A}_t])(\cdot) : \mathbb{R}^{m \times n} \to [\mathbb{R}^{m \times r}, \mathbb{R}^{r \times n}]$ are given by

$$J_{\mathcal{G}}([\boldsymbol{B}_t, \boldsymbol{A}_t])[\boldsymbol{P}, \boldsymbol{Q}] = \boldsymbol{P}\boldsymbol{A}_t + \boldsymbol{B}_t\boldsymbol{Q},$$

for any factor pairs $[\boldsymbol{P}, \boldsymbol{Q}] \in [\mathbb{R}^{m \times r}, \mathbb{R}^{r \times n}]$, and

$$J_{\mathcal{G}}^*([\boldsymbol{B}_t, \boldsymbol{A}_t])(\boldsymbol{C}) = [\boldsymbol{C}\boldsymbol{A}_t^\top, \boldsymbol{B}_t^\top\boldsymbol{C}],$$

for any matrices $\boldsymbol{C} \in \mathbb{R}^{m \times n}$. For detailed derivations and additional information regarding the Jacobian, please refer to Appendix D.1. Substituting $J_{\mathcal{G}}$ and $J_{\mathcal{G}}^*$ into (2), we can rewrite (2) as

$$\begin{aligned}
\boldsymbol{W}_{t+1} &\approx \boldsymbol{W}_t - \alpha_t\boldsymbol{G}_t \cdot \boldsymbol{A}_t^\top\boldsymbol{A}_t - \alpha_t\boldsymbol{B}_t\boldsymbol{B}_t^\top \cdot \boldsymbol{G}_t \\
&\approx (\boldsymbol{B}_t - \eta_t\boldsymbol{G}_t \cdot \boldsymbol{A}_t^\top)(\boldsymbol{A}_t - \eta_t\boldsymbol{B}_t^\top \cdot \boldsymbol{G}_t) \\
&= (\boldsymbol{B}_t - \eta_t\boldsymbol{G}_{\boldsymbol{B}_t})(\boldsymbol{A}_t - \eta_t\boldsymbol{G}_{\boldsymbol{A}_t}),
\end{aligned}$$

where $\boldsymbol{G}_t = \nabla_{\boldsymbol{W}_t}\mathcal{L}(\boldsymbol{W}_0 + \boldsymbol{W}_t)$, $\boldsymbol{G}_{\boldsymbol{B}_t} = \nabla_{\boldsymbol{B}_t}\mathcal{L}(\boldsymbol{W}_0 + \boldsymbol{W}_t)$ and $\boldsymbol{G}_{\boldsymbol{A}_t} = \nabla_{\boldsymbol{A}_t}\mathcal{L}(\boldsymbol{W}_0 + \boldsymbol{W}_t)$ are the gradient of the loss function $\mathcal{L}$ with respect to $\boldsymbol{W}_t$, $\boldsymbol{B}_t$ and $\boldsymbol{A}_t$. This formulation aligns with the update rule of standard LoRA (Vanilla LoRA) (Hu et al., 2022), in which the factors $\boldsymbol{B}$ and $\boldsymbol{A}$ are updated with the same learning rate. Consequently, the convergence rate of standard LoRA depends on the condition number of $J_{\mathcal{G}}$.

To mitigate this dependence on the condition number of $J_{\mathcal{G}}$, several improvements have been proposed. LoRA+ (Hayou et al., 2024) enhances feature learning efficiency by scaling the update $\eta_t\boldsymbol{G}_{\boldsymbol{B}_t}$ with a factor of $2^4$ when training Roberta (Liu et al., 2019) with LeCun initialization (LeCun et al., 2002). This adjustment can be regarded as applying a constant preconditioner on $\boldsymbol{G}_{\boldsymbol{B}_t}$. However, LoRA+ does not completely eliminate the dependence on the condition number of $J_{\mathcal{G}}$. Imbalance-Regularized LoRA (Zhu et al., 2024) further alleviates the impact of $J_{\mathcal{G}}$ by introducing regularization terms on the low-rank factors $\boldsymbol{B}_t$ and $\boldsymbol{A}_t$, which effectively reduce parameter redundancy. Going further, Riemannian preconditioned LoRA (Zhang and Pilanci, 2024) applies $r \times r$ preconditioners $(\boldsymbol{A}_t\boldsymbol{A}_t^\top)^{-1}$ and $(\boldsymbol{B}_t^\top\boldsymbol{B}_t)^{-1}$ to $\boldsymbol{G}_{\boldsymbol{B}_t}$ and $\boldsymbol{G}_{\boldsymbol{A}_t}$ respectively, making the update of $\boldsymbol{W}_t$ equivalent to projecting the gradient onto the row space of $\boldsymbol{A}_t$ and the column space of $\boldsymbol{B}_t$. Specifically,

$$\begin{aligned}
\boldsymbol{B}_{t+1}\boldsymbol{A}_{t+1} &= (\boldsymbol{B}_t - \eta_t\boldsymbol{G}_{\boldsymbol{B}_t} \cdot (\boldsymbol{A}_t\boldsymbol{A}_t^\top)^{-1})(\boldsymbol{A}_t - \eta_t(\boldsymbol{B}_t^\top\boldsymbol{B}_t)^{-1} \cdot \boldsymbol{G}_{\boldsymbol{A}_t}) \\
&\approx \boldsymbol{W}_t - \alpha_t\boldsymbol{G}_t \cdot \boldsymbol{A}_t^\top(\boldsymbol{A}_t\boldsymbol{A}_t^\top)^{-1}\boldsymbol{A}_t - \alpha_t\boldsymbol{B}_t(\boldsymbol{B}_t^\top\boldsymbol{B}_t)^{-1}\boldsymbol{B}_t^\top \cdot \boldsymbol{G}_t \\
&= \boldsymbol{W}_t - \alpha_t\text{Proj}_{\text{row}(\boldsymbol{A}_t)}(\boldsymbol{G}_t) - \alpha_t\text{Proj}_{\text{col}(\boldsymbol{B}_t)}(\boldsymbol{G}_t).
\end{aligned}$$

Although Riemannian preconditioned LoRA alleviates the influence of condition number of $J_{\mathcal{G}}$ to some extent via two preconditioners, it still has an important limitation: it ignores the projection onto the intersection of the row space of $\boldsymbol{A}_t$ and the column space of $\boldsymbol{B}_t$. Specifically, the term

$B_t(B_t^\top B_t)^{-1}B_t^\top \cdot G_t \cdot A_t^\top(A_t A_t^\top)^{-1}A_t$ is omitted, which causes the update direction to deviate from the steepest descent direction. To compensate for the missing information in this cross subspace, LoRA-Pro (Wang et al., 2025) proposes solving

$$\min_{\Delta_{B_t},\Delta_{A_t}} \|G_t - (B_t\Delta_{A_t} + \Delta_{B_t}A_t)\|_F^2,$$

to more accurately approximate the full fine-tuning gradient and obtain an equivalent low-rank gradient for the factors $B_t$ and $A_t$. Optimizing this objective yields the factor updates

$$\begin{cases} \Delta_{B_t} = [I - B_t(B_t^\top B_t)^{-1}B_t^\top]G_t A_t^\top(A_t A_t^\top)^{-1} - B_t X_t, \\ \Delta_{A_t} = (B_t^\top B_t)^{-1}B_t^\top G_t + X_t A_t, \end{cases}$$

for some $X_t \in \mathbb{R}^{r\times r}$. The corresponding update of the weight matrix is

$$\begin{aligned} &B_t\Delta_{A_t} + \Delta_{B_t}A_t \\ &= B_t(B_t^\top B_t)^{-1}B_t^\top G_t + B_t X_t A_t + [I - B_t(B_t^\top B_t)^{-1}B_t^\top]G_t A_t^\top(A_t A_t^\top)^{-1}A_t - B_t X_t A_t \\ &= \big(B_t(B_t^\top B_t)^{-1}B_t^\top\big)G_t + G_t\big(A_t^\top(A_t A_t^\top)^{-1}A_t\big) - \big(B_t(B_t^\top B_t)^{-1}B_t^\top\big)G_t\big(A_t^\top(A_t A_t^\top)^{-1}A_t\big) \\ &= \text{Proj}_{\text{col}(B_t)}(G_t) + \text{Proj}_{\text{row}(A_t)}(G_t) - \text{Proj}_{\text{col}(B_t)\cap\text{row}(A_t)}(G_t) = \mathcal{P}_{\mathbb{T}_t}(G_t). \end{aligned}$$

$$(3)$$

where $\mathcal{M}_r$ is the Riemannian manifold of all rank $r$ matrices, and $\mathbb{T}_t$ denotes the tangent space of $\mathcal{M}_r$ at the point $W_t$. By Proposition D.2, $\mathcal{P}_{\mathbb{T}_t}(G_t)$ is the orthogonal projection of $G_t$ onto $\mathbb{T}_t$.

Although LoRA-Pro is capable of finding a low-rank approximation of the full fine-tuning gradient under a standard metric, it incurs higher computational overhead due to solving a Sylvester equation at each step. (Lu, 1971; Dmytryshyn et al., 2025). In comparison, gradient approximations based on weighted metrics are often more effective, as they better utilize the second-order information of the loss function. For instance, classical methods such as the Broyden–Fletcher–Goldfarb–Shanno (BFGS) algorithm (Fletcher, 2000) and the Gauss–Newton method (Nocedal and Wright, 2006) both benefit from weighted metrics. Previous research has demonstrated that weighted metrics can significantly enhance algorithmic efficiency across a variety of problems (Duchi et al., 2011; Bian et al., 2024). Motivated by this, we propose designing a novel weighted metric to further improve the approximation of full fine-tuning gradients and to fully exploit the second-order information embedded in the loss function, enabling a more effective low-rank approximation.

## 3 THE PROPOSED ALGORITHMS

Empirical evidence suggests that the weighted metric are often more effective than the standard metric in deep learning. For instance, AdaGrad(Duchi et al., 2011; Shazeer and Stern, 2018) and SOAP (Gupta et al., 2018; Morwani et al., 2024; Vyas et al., 2025) adaptively adjust the step size of each gradient component based on historical gradient information, which is equivalent to using a weighted metric for the weight matrix. Similarly, K-FAC (Martens and Grosse, 2015; Eschenhagen et al., 2023) employs a Kronecker product-based weighted metric to approximate the Hessian, thereby constructing an efficient preconditioner. In this section, we introduce a novel weighted metric and derive a low-rank approximation of the full fine-tuning gradient $G$ based on this metric. This low-rank approximation can be viewed as an approximation of the Hessian, allowing our algorithm to effectively exploit the second-order information of the loss function, thereby narrowing the gap between the performance of low-rank fine-tuning and full fine-tuning.

### 3.1 CONSTRUCTION OF THE ADAPTIVE METRIC

The core idea of AdaGrad (Duchi et al., 2011) and Adam (Kingma and Ba, 2014) is to construct a weighted operator $h_t$ through the outer product of gradients, followed by a diagonalization operation. Specifically, by vectorizing the gradient matrix $G_t \in \mathbb{R}^{m\times n}$ into $g_t \in \mathbb{R}^{mn\times 1}$, the linearized weighted operator $h_t$ is denoted as $h_t = \big(h_{t-1}^2 + \text{diag}(g_t g_t^\top)\big)^{\frac{1}{2}}$, where $\text{diag}(\cdot)$ extracts the diagonal elements of the matrix. This operator $h_t$ is then used to define a new weighted inner product, under which the gradient descent update to linearized weight $w_t$ is derived $w_{t+1} = w_t - g_t/h_t$. AdaGrad employs the elements of $h_t$ to rescale the gradient element-wise. However, when applied to gradient

updates in matrix form, this element-wise rescaling approach ignores the structural information of the matrix, i.e., the relationships between rows and columns. To better utilize matrix structures, Shampoo (Gupta et al., 2018) adopts the Kronecker product to approximate the construction of the weighted matrix. Specifically, Shampoo constructs two matrices, $\boldsymbol{L}_t = \boldsymbol{L}_{t-1} + \boldsymbol{G}_t \boldsymbol{G}_t^\top$ and $\boldsymbol{R}_t = \boldsymbol{R}_{t-1} + \boldsymbol{G}_t^\top \boldsymbol{G}_t$, and defines a weighted inner product based on these matrices. Under this inner product, the matrix update is performed by $\boldsymbol{W}_{t+1} = \boldsymbol{W}_t - \boldsymbol{L}_t^{-\frac{1}{4}} \boldsymbol{G}_t \boldsymbol{R}_t^{-\frac{1}{4}}$. SOAP (Morwani et al., 2024; Vyas et al., 2025) further improves upon Shampoo by noting that the square root operation in Shampoo is equivalent to running Adafactor (Shazeer and Stern, 2018) in the eigenbasis of the Shampoo preconditioner. To enhance the computational efficiency of Shampoo, SOAP runs Adam in the eigenbasis of the Shampoo preconditioner. However, frequent eigen-decomposition computations result in high computational costs. To balance leveraging matrix structural information and maintaining computational efficiency, we proposes a hybrid weighted inner product that is easier to implement, better aligns with matrix structures, and fully utilizes the relationships between matrix rows and columns.

Specifically, we define the weighted factors $\boldsymbol{L}_t$ and $\boldsymbol{R}_t$ as follows:

$$\boldsymbol{L}_t = \mathrm{diag}(\boldsymbol{l}_t/\sqrt{\|\boldsymbol{l}_t\|_1}) \ \text{ with } \ \boldsymbol{l}_t = \beta_1 \boldsymbol{l}_{t-1} + (1-\beta_1)\sum_{j=1}^n (\boldsymbol{G}_t \odot \boldsymbol{G}_t)_{i,j},$$

$$\boldsymbol{R}_t = \mathrm{diag}(\boldsymbol{r}_t/\sqrt{\|\boldsymbol{r}_t\|_1}) \ \text{ with } \ \boldsymbol{r}_t = \beta_2 \boldsymbol{r}_{t-1} + (1-\beta_2)\sum_{i=1}^m (\boldsymbol{G}_t \odot \boldsymbol{G}_t)_{i,j},$$

(4)

where $\odot$ denotes the Hadamard (element-wise) product, $\|\cdot\|_1$ denotes the $l_1$-norm, $\beta_1, \beta_2$ are decay factors in the range $[0, 1]$. The term $\sum_{j=1}^n (\boldsymbol{G}_t \odot \boldsymbol{G}_t)_{i,j}$ forms a vector of the diagonal elements of the matrix $\boldsymbol{G}_t \boldsymbol{G}_t^\top$, and similarly, $\sum_{i=1}^m (\boldsymbol{G}_t \odot \boldsymbol{G}_t)_{i,j}$ forms a vector of the diagonal elements of the matrix $\boldsymbol{G}_t^\top \boldsymbol{G}_t$. As stated in (Shazeer and Stern, 2018), $\boldsymbol{l}_t \boldsymbol{r}_t^\top$ is a rank-1 approximation of $\boldsymbol{G} \odot \boldsymbol{G}$, which is optimal with respect to the generalized Kullback-Leibler divergence. In this way, the memory requirement is reduced from $\mathcal{O}(mn)$ to $\mathcal{O}(m+n)$. At the same time, compared to Shampoo, the computational complexity of $\boldsymbol{L}_t$ and $\boldsymbol{R}_t$ is reduced to $\mathcal{O}(mn)$.

Based on $\boldsymbol{L}_t$ and $\boldsymbol{R}_t$, we define an adaptive weighted inner product in $\mathbb{R}^{m \times n}$. For any $\boldsymbol{Y}, \boldsymbol{Z} \in \mathbb{R}^{m \times n}$, the adaptive weighted inner product is given by:

$$\langle \boldsymbol{Y}, \boldsymbol{Z} \rangle_{\boldsymbol{H}_t} = \langle \boldsymbol{H}_t \boldsymbol{Y}, \boldsymbol{Z} \rangle = \langle \boldsymbol{L}_t^{\frac{1}{2}} \boldsymbol{Y} \boldsymbol{R}_t^{\frac{1}{2}}, \boldsymbol{Z} \rangle. \tag{5}$$

For any matrix $\boldsymbol{K} \in \mathbb{R}^{m \times n}$, the inverse operation of the operator $\boldsymbol{H}_t$ is defined as

$$\boldsymbol{H}_t^{-1} \boldsymbol{K} = \boldsymbol{L}_t^{-\frac{1}{2}} \boldsymbol{K} \boldsymbol{R}_t^{-\frac{1}{2}}.$$

### 3.2 Second-order Low-Rank Adaption for Fine-tuning

Based on the adaptive weighted inner product, we aim to incorporate second-order information into the update of the weight matrix $\boldsymbol{W}$. To achieve this, we consider the update of the weight matrix $\boldsymbol{W}$ in step $t$. Let the update to $\boldsymbol{W}$ at step $t$ be denoted as $\boldsymbol{\Delta}_t$. In this step, we solve the problem:

$$\min_{\boldsymbol{\Delta}_t} \mathcal{L}\big((\boldsymbol{W}_0 + \boldsymbol{W}_t) - \boldsymbol{\Delta}_t\big).$$

We then expand the loss function $\mathcal{L}(\boldsymbol{W})$ around the point $\boldsymbol{W}_0 + \boldsymbol{W}_t$ using its second-order Taylor expansion. By utilizing the weighted inner product as an approximation of Hessian, the optimization problem can be formulated as:

$$\arg\min_{\boldsymbol{\Delta}_t} \ \mathcal{L}((\boldsymbol{W}_0 + \boldsymbol{W}_t) - \boldsymbol{\Delta}_t)$$

$$\approx \arg\min_{\boldsymbol{\Delta}_t} \ \mathcal{L}(\boldsymbol{W}_0 + \boldsymbol{W}_t) - \langle \boldsymbol{\Delta}_t, \boldsymbol{G}_t \rangle + \frac{1}{2}\langle \boldsymbol{H}_t \boldsymbol{\Delta}_t, \boldsymbol{\Delta}_t \rangle,$$

$$= \arg\min_{\boldsymbol{\Delta}_t} \ \mathcal{L}(\boldsymbol{W}_0 + \boldsymbol{W}_t) - \langle \boldsymbol{\Delta}_t, \boldsymbol{H}_t^{-1}\boldsymbol{G}_t \rangle_{\boldsymbol{H}_t} + \frac{1}{2}\langle \boldsymbol{\Delta}_t, \boldsymbol{\Delta}_t \rangle_{\boldsymbol{H}_t} + \frac{1}{2}\langle \boldsymbol{H}_t^{-1}\boldsymbol{G}_t, \boldsymbol{H}_t^{-1}\boldsymbol{G}_t \rangle_{\boldsymbol{H}_t}$$

$$= \arg\min_{\boldsymbol{\Delta}_t} \ \mathcal{L}(\boldsymbol{W}_0 + \boldsymbol{W}_t) + \frac{1}{2}\|\boldsymbol{\Delta}_t - \boldsymbol{H}_t^{-1}\boldsymbol{G}_t\|_{\boldsymbol{H}_t}^2.$$

From this expression, it is evident that the optimization problem is equivalent to finding the optimal $\boldsymbol{\Delta}_t$ for the following objective:

$$\min_{\boldsymbol{\Delta}_t} \|\boldsymbol{\Delta}_t - \boldsymbol{H}_t^{-1}\boldsymbol{G}_t\|_{\boldsymbol{H}_t}^2. \tag{6}$$

Let the optimal update be denoted as $\boldsymbol{\Delta}_t^{\mathrm{opt}}$. From the form of (6), it becomes clear that $\boldsymbol{\Delta}_t^{\mathrm{opt}}$ serves as an approximation of the Newton direction

$$-\boldsymbol{\Delta}_t^{\mathrm{opt}} \approx -\nabla^2 \mathcal{L}(\boldsymbol{W}_0 + \boldsymbol{W}_t) \cdot \nabla \mathcal{L}(\boldsymbol{W}_0 + \boldsymbol{W}_t).$$

Thus, the weight matrix is updated as $\boldsymbol{W}_{t+1} = \boldsymbol{W}_t - \boldsymbol{\Delta}_t^{\mathrm{opt}}$. The advantages of this update are evident:

- It completely eliminates the adverse effects of the condition number of the Jacobian operator $J_{\mathcal{G}}$, thereby improving the stability of the algorithm.
- It effectively incorporates the second-order information of the loss function, enhancing optimization efficiency.

To further reduce memory consumption, we adopt the low-rank factorization strategy of LoRA, representing the update $\boldsymbol{\Delta}_t$ in terms of updates to the low-rank factors $\boldsymbol{A}_t$ and $\boldsymbol{B}_t$, denoted as $\boldsymbol{\Delta}_{\boldsymbol{A}_t}$ and $\boldsymbol{\Delta}_{\boldsymbol{B}_t}$, respectively. As noted in (Wang et al., 2025), the changes in the factors $\boldsymbol{A}_t$ and $\boldsymbol{B}_t$ are intrinsically related to the updates in the weight matrix $\boldsymbol{W}_t$, which can be expressed as

$$\boldsymbol{\Delta}_t = \boldsymbol{\Delta}_{\boldsymbol{B}_t}\boldsymbol{A}_t + \boldsymbol{B}_t\boldsymbol{\Delta}_{\boldsymbol{A}_t}.$$

Therefore, the minimization problem (6) can be equivalently transformed into:

$$\min_{\boldsymbol{\Delta}_{\boldsymbol{B}_t}, \boldsymbol{\Delta}_{\boldsymbol{A}_t}} \|\boldsymbol{\Delta}_{\boldsymbol{B}_t}\boldsymbol{A}_t + \boldsymbol{B}_t\boldsymbol{\Delta}_{\boldsymbol{A}_t} - \boldsymbol{H}_t^{-1}\boldsymbol{G}_t\|_{\boldsymbol{H}_t}^2. \tag{7}$$

To make the optimization process more explicit, we first rewrite (7) as:

$$\arg\min_{\boldsymbol{\Delta}_{\boldsymbol{B}_t}, \boldsymbol{\Delta}_{\boldsymbol{A}_t}} \|\widetilde{\mathcal{P}}_{\mathbb{T}_t}\big(\boldsymbol{\Delta}_{\boldsymbol{B}_t}\boldsymbol{A}_t + \boldsymbol{B}_t\boldsymbol{\Delta}_{\boldsymbol{A}_t} - \boldsymbol{H}_t^{-1}\boldsymbol{G}_t\big) + \widetilde{\mathcal{P}}_{\mathbb{T}_t}^{\perp}\big(\boldsymbol{\Delta}_{\boldsymbol{B}_t}\boldsymbol{A}_t + \boldsymbol{B}_t\boldsymbol{\Delta}_{\boldsymbol{A}_t} - \boldsymbol{H}_t^{-1}\boldsymbol{G}_t\big)\|_{\boldsymbol{H}_t}^2$$

$$= \arg\min_{\boldsymbol{\Delta}_{\boldsymbol{B}_t}, \boldsymbol{\Delta}_{\boldsymbol{A}_t}} \|\boldsymbol{\Delta}_{\boldsymbol{B}_t}\boldsymbol{A}_t + \boldsymbol{B}_t\boldsymbol{\Delta}_{\boldsymbol{A}_t} - \widetilde{\mathcal{P}}_{\mathbb{T}_t}(\boldsymbol{H}_t^{-1}\boldsymbol{G}_t)\|_{\boldsymbol{H}_t}^2 + \|\widetilde{\mathcal{P}}_{\mathbb{T}_t}^{\perp}\big(\boldsymbol{H}_t^{-1}\boldsymbol{G}_t\big)\|_{\boldsymbol{H}_t}^2, \tag{8}$$

where $\widetilde{\mathcal{P}}_{\mathbb{T}_t}^{\perp}(\cdot)$ denotes the projection onto the space orthogonal to the tangent space. This equivalence holds because $\boldsymbol{\Delta}_{\boldsymbol{B}_t}\boldsymbol{A}_t + \boldsymbol{B}_t\boldsymbol{\Delta}_{\boldsymbol{A}_t}$ lies in the tangent space $\mathbb{T}_t$ (see Proposition D.4). This implies that, to find the optimal $\boldsymbol{\Delta}_{\boldsymbol{B}_t}$ and $\boldsymbol{\Delta}_{\boldsymbol{A}_t}$, we ultimately need to solve the following equivalent problem:

$$\min_{\boldsymbol{\Delta}_{\boldsymbol{B}_t}, \boldsymbol{\Delta}_{\boldsymbol{A}_t}} \|\boldsymbol{\Delta}_{\boldsymbol{B}_t}\boldsymbol{A}_t + \boldsymbol{B}_t\boldsymbol{\Delta}_{\boldsymbol{A}_t} - \widetilde{\mathcal{P}}_{\mathbb{T}_t}(\boldsymbol{H}_t^{-1}\boldsymbol{G}_t)\|_{\boldsymbol{H}_t}^2. \tag{9}$$

Here, $\widetilde{\mathcal{P}}_{\mathbb{T}_t}(\boldsymbol{H}_t^{-1}\boldsymbol{G}_t)$ represents the projection of $\boldsymbol{H}_t^{-1}\boldsymbol{G}_t$ onto $\mathbb{T}_t$, with its explicit form given as

$$\widetilde{\mathcal{P}}_{\mathbb{T}_t}(\boldsymbol{L}_t^{-\frac{1}{2}}\boldsymbol{G}_t\boldsymbol{R}_t^{-\frac{1}{2}}) = \widetilde{\boldsymbol{P}}_{B_t}\boldsymbol{L}_t^{-\frac{1}{2}}\boldsymbol{G}_t\boldsymbol{R}_t^{-\frac{1}{2}} + \boldsymbol{L}_t^{-\frac{1}{2}}\boldsymbol{G}_t\boldsymbol{R}_t^{-\frac{1}{2}}\widetilde{\boldsymbol{Q}}_{A_t} - \widetilde{\boldsymbol{P}}_{B_t}\boldsymbol{L}_t^{-\frac{1}{2}}\boldsymbol{G}_t\boldsymbol{R}_t^{-\frac{1}{2}}\widetilde{\boldsymbol{Q}}_{A_t}, \tag{10}$$

where $\widetilde{\boldsymbol{P}}_{B_t} = \boldsymbol{B}_t(\boldsymbol{B}_t^{\top}\boldsymbol{L}_t^{\frac{1}{2}}\boldsymbol{B}_t)^{-1}\boldsymbol{B}_t^{\top}\boldsymbol{L}_t^{\frac{1}{2}}$ and $\widetilde{\boldsymbol{Q}}_{A_t} = \boldsymbol{R}_t^{\frac{1}{2}}\boldsymbol{A}_t^{\top}(\boldsymbol{A}_t\boldsymbol{R}_t^{\frac{1}{2}}\boldsymbol{A}_t^{\top})^{-1}\boldsymbol{A}_t$. The detailed derivation is provided in Appendix D.3.

For problem (9), we provide its explicit solution in the following Theorem 3.1. For the proof of Theorem 3.1, please refer to Appendix D.3.

**Theorem 3.1** (Optimal updates for low-rank factors). *Let $\boldsymbol{W}_t = \boldsymbol{B}_t\boldsymbol{A}_t$ be a rank-$r$ factorization at $t$-th step, and let $\widetilde{\mathcal{P}}_{\mathbb{T}_t}(\boldsymbol{L}_t^{-\frac{1}{2}}\boldsymbol{G}_t\boldsymbol{R}_t^{-\frac{1}{2}})$ denote the projection of the preconditioned gradient $\boldsymbol{L}_t^{-\frac{1}{2}}\boldsymbol{G}_t\boldsymbol{R}_t^{-\frac{1}{2}}$ onto the tangent space $\mathbb{T}_t$ at $\boldsymbol{W}_t$. Consider the following optimization problem:*

$$\min_{\boldsymbol{\Delta}_{\boldsymbol{B}_t}, \boldsymbol{\Delta}_{\boldsymbol{A}_t}} \frac{1}{2}\|\boldsymbol{\Delta}_{\boldsymbol{B}_t}\boldsymbol{A}_t + \boldsymbol{B}_t\boldsymbol{\Delta}_{\boldsymbol{A}_t} - \widetilde{\mathcal{P}}_{\mathbb{T}_t}(\boldsymbol{L}_t^{-\frac{1}{2}}\boldsymbol{G}_t\boldsymbol{R}_t^{-\frac{1}{2}})\|_{\boldsymbol{H}_t}^2, \tag{11}$$

*where $\|\cdot\|_{\boldsymbol{H}_t}$ is the norm induced by the operator $\boldsymbol{H}_t$. Then the optimal solutions for $\boldsymbol{\Delta}_{\boldsymbol{B}_t}$ and $\boldsymbol{\Delta}_{\boldsymbol{A}_t}$ are given by*

$$\boldsymbol{\Delta}_{\boldsymbol{B}_t}^{opt} = [\boldsymbol{I} - \boldsymbol{B}_t(\boldsymbol{B}_t^{\top}\boldsymbol{L}_t^{\frac{1}{2}}\boldsymbol{B}_t)^{-1}\boldsymbol{B}_t^{\top}\boldsymbol{L}_t^{\frac{1}{2}}]\boldsymbol{L}_t^{-\frac{1}{2}}\boldsymbol{G}_{B_t}(\boldsymbol{A}_t\boldsymbol{R}_t^{\frac{1}{2}}\boldsymbol{A}_t^{\top})^{-1} - \boldsymbol{B}_t\boldsymbol{X}_t,$$

$$\boldsymbol{\Delta}_{\boldsymbol{A}_t}^{opt} = (\boldsymbol{B}_t^{\top}\boldsymbol{L}_t^{\frac{1}{2}}\boldsymbol{B}_t)^{-1}\boldsymbol{G}_{A_t}\boldsymbol{R}_t^{-\frac{1}{2}} + \boldsymbol{X}_t\boldsymbol{A}_t,$$

*where $\boldsymbol{X}_t \in \mathbb{R}^{r \times r}$ is an arbitrary matrix.*

From Theorem 3.1, we observe that although $\boldsymbol{G}_t$ appears in the solution, it does not directly appear in the closed-form expression. Instead, the solution depends on the low-rank gradients $\boldsymbol{G}_{\boldsymbol{A}_t}$ and $\boldsymbol{G}_{\boldsymbol{B}_t}$, ensuring low memory overhead. This efficient representation allows for straightforward gradient updates: first, compute the gradients using standard backpropagation, and then adjust $\boldsymbol{\Delta}_{\boldsymbol{B}_t}$ and $\boldsymbol{\Delta}_{\boldsymbol{A}_t}$ according to the closed-form solution. While $\boldsymbol{\Delta}_{\boldsymbol{B}_t}$ and $\boldsymbol{\Delta}_{\boldsymbol{A}_t}$ depend on $\boldsymbol{X}_t$, the choice of $\boldsymbol{X}_t$ is critical for balancing the updates. Next, we minimize the weighted norm of the difference between the two update components, $\boldsymbol{\Delta}_{\boldsymbol{B}_t}\boldsymbol{A}_t$ and $\boldsymbol{B}_t\boldsymbol{\Delta}_{\boldsymbol{A}_t}$. This yields the optimal $\boldsymbol{X}_t$ in Theorem 3.2 (proof provided in Appendix D.3).

Once the matrix $\boldsymbol{X}_t$ is computed, $\boldsymbol{\Delta}_{\boldsymbol{B}_t}$ and $\boldsymbol{\Delta}_{\boldsymbol{A}_t}$ can be derived. Using the updates $\boldsymbol{\Delta}_{\boldsymbol{B}_t}$ and $\boldsymbol{\Delta}_{\boldsymbol{A}_t}$, we propose **S**econd-**o**rder **Lo**w-**R**ank **A**daption (**SoLoRA**), summarized in Algorithm 1. The computational complexity is analyzed in Appendix C.

**Theorem 3.2** (Optimal Solution for Balancing Matrix $\boldsymbol{X}_t$)**.** *Let $\boldsymbol{X}_t \in \mathbb{R}^{r \times r}$. Consider the following optimization problem with respect to $\boldsymbol{X}_t$,*

$$\min_{\boldsymbol{X}_t \in \mathbb{R}^{r \times r}} \frac{1}{2}\|\boldsymbol{\Delta}_{\boldsymbol{B}_t}\boldsymbol{A}_t - \boldsymbol{B}_t\boldsymbol{\Delta}_{\boldsymbol{A}_t}\|^2_{\boldsymbol{H}_t}, \tag{12}$$

*where $\boldsymbol{\Delta}_{\boldsymbol{B}_t}$ and $\boldsymbol{\Delta}_{\boldsymbol{A}_t}$ are functions of $\boldsymbol{X}_t$ given in Theorem 3.1. Then the optimal solution for $\boldsymbol{X}_t$ is given by*

$$\boldsymbol{X}_t^{opt} = -\frac{1}{2}(\boldsymbol{B}_t^\top \boldsymbol{L}_t^{\frac{1}{2}}\boldsymbol{B}_t)^{-1}\boldsymbol{B}_t^\top \boldsymbol{G}_t \boldsymbol{A}_t^\top (\boldsymbol{A}_t \boldsymbol{R}_t^{\frac{1}{2}}\boldsymbol{A}_t^\top)^{-1}.$$

---

**Algorithm 1** Second-order Low-Rank Adaption (SoLoRA) with SGD for Fine-tuning.

---

1: Initialize $\boldsymbol{B}_1 = \boldsymbol{0}_{m \times r}$, $\boldsymbol{A}_1 = $ Kaiming uniform$_{r \times n}$, $\boldsymbol{l}_0 = \boldsymbol{0}_m$, $\boldsymbol{r}_0 = \boldsymbol{0}_n$, $\epsilon = 1e-6$.
2: **for** $t = 1, \cdots, T$ **do**
3: $\quad \boldsymbol{l}_t = \beta_1 \boldsymbol{l}_{t-1} + (1 - \beta_1)\sum_{j=1}^n (\boldsymbol{G}_t \odot \boldsymbol{G}_t)_{i,j}, \boldsymbol{L}_t = \mathrm{diag}(\boldsymbol{l}_t/\sqrt{\|\boldsymbol{l}_t\|_1})$.
4: $\quad \boldsymbol{r}_t = \beta_2 \boldsymbol{r}_{t-1} + (1 - \beta_2)\sum_{i=1}^m (\boldsymbol{G}_t \odot \boldsymbol{G}_t)_{i,j}, \boldsymbol{R}_t = \mathrm{diag}(\boldsymbol{r}_t/\sqrt{\|\boldsymbol{r}_t\|_1})$.
5: $\quad \boldsymbol{\Delta}_{\boldsymbol{B}_t} = \left[\boldsymbol{I} - \frac{1}{2}\boldsymbol{B}_t(\boldsymbol{B}_t^\top \boldsymbol{L}_t^{\frac{1}{2}}\boldsymbol{B}_t)^{-1}\boldsymbol{B}_t^\top \boldsymbol{L}_t^{\frac{1}{2}}\right]\boldsymbol{L}_t^{-\frac{1}{2}}\boldsymbol{G}_{\boldsymbol{B}_t}(\boldsymbol{A}_t \boldsymbol{R}_t^{\frac{1}{2}}\boldsymbol{A}_t^\top)^{-1}$.
6: $\quad \boldsymbol{\Delta}_{\boldsymbol{A}_t} = (\boldsymbol{B}_t^\top \boldsymbol{L}_t^{\frac{1}{2}}\boldsymbol{B}_t)^{-1}\boldsymbol{G}_{\boldsymbol{A}_t}\boldsymbol{R}_t^{-\frac{1}{2}}\left[\boldsymbol{I} - \frac{1}{2}\boldsymbol{R}_t^{\frac{1}{2}}\boldsymbol{A}_t^\top (\boldsymbol{A}_t \boldsymbol{R}_t^{\frac{1}{2}}\boldsymbol{A}_t^\top)^{-1}\boldsymbol{A}_t\right]$.
7: $\quad \boldsymbol{B}_{t+1} = \boldsymbol{B}_t - \eta_t \boldsymbol{\Delta}_{\boldsymbol{B}_t}, \boldsymbol{A}_{t+1} = \boldsymbol{A}_t - \eta_t \boldsymbol{\Delta}_{\boldsymbol{A}_t}$.
8: **end for**
9: Note: Add $\epsilon \boldsymbol{I}$ to matrix $\boldsymbol{B}_t^\top \boldsymbol{L}_t^{\frac{1}{2}}\boldsymbol{B}_t$ if it is not invertible.

---

### 3.3 SECOND-ORDER LOW-RANK ADAPTION WITH MOMENTUM FOR FINE-TUNING.

First-order momentum methods, such as Adam and AdamW (Kingma and Ba, 2014; Loshchilov and Hutter, 2017), have been shown to be highly effective in stochastic optimization. By maintaining an exponential moving average of both the per-coordinate gradient statistics and the raw gradients, Adam stabilizes updates, reduces gradient variance, and minimizes sensitivity to manual learning rate tuning. To incorporate these advantages into our second-order low-rank adaptation framework, we integrate the exponential moving average of the gradients into SoLoRA. The enhanced method preserves the curvature-aware geometric properties of SoLoRA while inheriting the stability and adaptivity of Adam, resulting in more reliable and efficient fine-tuning. The pseudocode is present in Algorithm 2.

## 4 EXPERIMENTAL RESULTS

To evaluate the performance of our SoLoRA algorithm, we apply it to fine-tuning tasks for the large language model GPT-2 (see Section 4.1 and Appendix A) and diffusion models (see Appendix B). In the experiments, we compare two kinds of optimization algorithms: SGD-based algorithms and AdamW-based algorithms. The SGD-based algorithms include: LoRA with SGD optimizer (referred to as SGD) (Hu et al., 2022), Scaled GD (Zhang and Pilanci, 2024; Tong et al., 2021), LoRA-Pro with SGD optimizer (Wang et al., 2025), and our SoLoRA with SGD optimizer (Algorithm 1). The AdamW-based algorithms include: LoRA with AdamW optimizer (referred to as AdamW) (Hu et al.,

---

**Algorithm 2** **S**econd-**o**rder **Lo**w-**R**ank **A**daption (**SoLoRA**) with Momentum for Fine-tuning.

1: Initialize moment $M_0 = \mathbf{0}_{m \times n}$, $B_1 = \mathbf{0}_{m \times r}$, $A_1 = \text{Kaiming uniform}_{r \times n}$; $l_0 = \mathbf{0}_m$, $r_0 = \mathbf{0}_n$, weight decay $\lambda$, coefficients $\beta_1 = \beta_2$, and $\beta_3$, $\epsilon = 1e - 6$.
2: **for** $t = 1, \cdots, T$ **do**
3: $\quad l_t = \beta_1 l_{t-1} + (1 - \beta_1) \sum_{j=1}^{n} (G_t \odot G_t)_{i,j}$, $L_t = \text{diag}(l_t / \sqrt{\|l_t\|_1})$.
4: $\quad r_t = \beta_2 r_{t-1} + (1 - \beta_2) \sum_{i=1}^{m} (G_t \odot G_t)_{i,j}$, $R_t = \text{diag}(r_t / \sqrt{\|r_t\|_1})$.
5: $\quad M_t = \beta_3 M_{t-1} + (1 - \beta_3) G_t$.
6: $\quad \Delta_{B_t} = \left[ I - \frac{1}{2} B_t \left( B_t^\top L_t^{\frac{1}{2}} B_t \right)^{-1} B_t^\top L_t^{\frac{1}{2}} \right] L_t^{-\frac{1}{2}} M_t A_t^\top \left( A_t R_t^{\frac{1}{2}} A_t^\top \right)^{-1}.$
7: $\quad \Delta_{A_t} = \left( B_t^\top L_t^{\frac{1}{2}} B_t \right)^{-1} B_t^\top M_t R_t^{-\frac{1}{2}} \left[ I - \frac{1}{2} R_t^{\frac{1}{2}} A_t^\top \left( A_t R_t^{\frac{1}{2}} A_t^\top \right)^{-1} A_t \right].$
8: $\quad B_{t+1} = (1 - \lambda \eta_t) B_t - \eta_t \frac{\sqrt{1 - \beta_1^t}}{1 - \beta_3^t} \Delta_{B_t}$, $A_{t+1} = (1 - \lambda \eta_t) A_t - \eta_t \frac{\sqrt{1 - \beta_1^t}}{1 - \beta_3^t} \Delta_{A_t}.$
9: **end for**
10: Note: Add $\epsilon I$ to matrix $B_t^\top L_t^{\frac{1}{2}} B_t$ if it is not invertible.

---

2022), Scaled AdamW (Zhang and Pilanci, 2024), LoRA-Pro with AdamW optimizer (Wang et al., 2025), and our SoLoRA with AdamW optimizer (Algorithm 2). All experiments are implemented using PyTorch (Paszke et al., 2019) and conducted on NVIDIA GeForce RTX 4090 or 3090 GPUs.

## 4.1 GPT-2 FINE-TUNING

In this section, we conduct fine-tuning experiments on the GPT-2 model (Radford et al., 2019) using SoLoRA. First, we perform fine-tuning on the GPT-2 small model with ranks 16 and 64, evaluated on the E2E natural language generation challenge (Novikova et al., 2017). The results are shown in Table 1. The experimental setup follows (Zhang and Pilanci, 2024), but we independently tune the learning rate for each optimizer using grid search. As shown in Table 1, the model trained with SoLoRA outperforms all other methods across all evaluation metrics, regardless of whether the SGD or AdamW optimizer is used. To further validate the efficiency of SoLoRA, we compare the loss reduction trends when employing different optimizers under the same runtime and the same number of iterations. These results are illustrated in Figures 1 and 2. The findings demonstrate that SoLoRA achieves significantly faster loss reduction than other algorithms within the same runtime, thanks to its effective utilization of second-order information of the loss function.

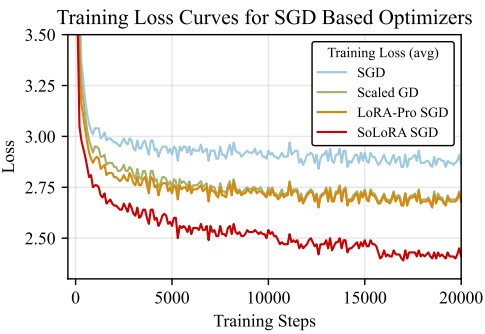
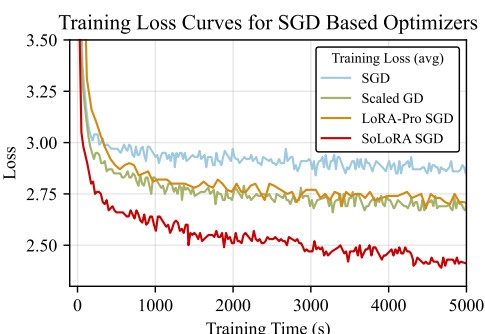

(a) Training loss curve over training step when fine-tuning using SGD-based methods.

(b) Training loss curve over training time when fine-tuning using SGD-based methods.

Figure 1: Training loss GPT-2 small model ($r = 64$) fine-tuned using different SGD-based optimizers. Evaluation is conducted on E2E Natural Language Generation Challenge.

Optimizing low-rank factorization matrices presents inherent challenges, particularly when the weight matrix contains small singular values — a scenario that often arises with larger ranks, such as ranks 16 and 64 in this experiment. Under these conditions, the curvature of Hessian becomes very large, resulting in a high condition number and making the optimization problem ill-conditioned. Despite

these challenges, SoLoRA demonstrates superior performance in both computational efficiency and final evaluation metrics. This highlights the ability of SoLoRA to effectively mitigate the impact of $J_{\mathcal{G}}$'s condition number while leveraging the second-order information from the loss function. To further evaluate SoLoRA, we conducted additional experiments on GPT-2 models of varying sizes with rank 4. The results are presented in Table 2 (see Appendix A), reaffirm the advantages of SoLoRA. Finally, we test the stability of SoLoRA under different learning rates, with the results shown in Figure 3 (see Appendix A). The experiments reveal that, compared to other algorithms, SoLoRA exhibits greater stability across varying ranks and learning rates.

Table 1: Scores of GPT-2 small model fine-tuned using different optimizers. Evaluation is conducted on E2E Natural Language Generation challenge.

| rank | Method | E2E | | | | |
| | | BLEU | NIST | MET | ROUGE-L | CIDEr |
|---|---|---|---|---|---|---|
| 16 | SGD | 65.4 | 8.07 | 40.7 | 67.0 | 2.07 |
| | Scaled GD | 68.8 | 8.75 | 45.0 | 69.2 | 2.39 |
| | LoRA-Pro SGD | 68.3 | 8.67 | 45.1 | 69.3 | 2.37 |
| | SoLoRA SGD (ours) | **70.0** | **8.82** | **46.6** | **71.6** | **2.53** |
| | AdamW | 69.5 | 8.77 | 46.4 | 71.2 | 2.48 |
| | Scaled AdamW | 69.8 | 8.79 | 46.5 | 71.7 | 2.51 |
| | LoRA-Pro AdamW | 69.7 | 8.73 | **46.8** | 71.7 | 2.51 |
| | SoLoRA AdamW (ours) | **70.2** | **8.85** | 46.6 | **71.9** | **2.52** |
| 64 | SGD | 64.7 | 8.08 | 40.8 | 66.7 | 2.04 |
| | Scaled GD | 68.5 | 8.68 | 45.0 | 69.4 | 2.38 |
| | LoRA-Pro SGD | 68.6 | 8.71 | 45.4 | 69.7 | 2.38 |
| | SoLoRA SGD (ours) | **70.1** | **8.85** | **46.7** | **71.8** | **2.53** |
| | AdamW | 69.6 | 8.76 | 46.7 | 71.5 | 2.50 |
| | Scaled AdamW | 70.0 | 8.83 | 46.4 | 71.5 | 2.50 |
| | LoRA-Pro AdamW | 70.0 | 8.82 | 46.6 | 71.5 | 2.51 |
| | SoLoRA AdamW (ours) | **70.2** | **8.84** | **46.8** | **72.1** | **2.52** |

## 5 CONCLUSION

This paper addresses the performance limitations of low-rank fine-tuning in efficiently adapting large models by proposing the second-order low-rank adaptation algorithm, **SoLoRA**. SoLoRA leverages an adaptive metric inspired by AdaGrad (Duchi et al., 2011) and SOAP (Vyas et al., 2025) to efficiently compute a low-rank approximation of the full fine-tuning gradient. This approximation, which can be viewed as an approximation of Hessian, effectively incorporates second-order information, accelerating convergence and improving optimization efficiency. Compared to existing low-rank fine-tuning methods, SoLoRA not only exploits second-order information but also completely eliminates the impact of the condition number of Jacobian operator. Moreover, as its low-rank approximation does not directly depend on the full gradient, SoLoRA is simpler and more efficient to implement. Experiments on GPT-2 and diffusion models consistently demonstrate that SoLoRA outperforms state-of-the-art low-rank fine-tuning methods. It achieves performance close to full fine-tuning while incurring almost no additional computational cost. This strongly demonstrates that second-order low-rank approximations based on our adaptive weighted metric provide a practical path to bridging the gap between parameter efficiency and optimal performance, paving the way for efficient and robust task transfer and personalized customization in large models.

**Ethics statement**   This paper conforms with the ICLR Code of Ethics.

**Reproducibility statement**   We are committed to the reproducibility of our research. To this end, we have made all source code, environmental configurations, and data access instructions available in the supplementary material. Furthermore, the key parameters for our experiments are provided in Table 3, Table 4 , and Table 6 to facilitate the replication of our findings.

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

# CONTENTS

# A  SUPPLEMENTARY EXPERIMENTS OF GPT-2 FINE-TUNING

## A.1  EXPERIMENTAL RESULTS FOR DIFFERENT DATASETS

To further validate the effectiveness of SoLoRA, we also conducted the experiments of GPT-2 fine-tuning on the DART (Nan et al., 2021) dataset. with the results provided in the following table.

Scores of GPT-2 small model (rank=4) fine-tuned using different optimizers. Evaluation is conducted on DART dataset.

| Methods | BLEU↑ | METEOR↑ | chrF++↑ | TER↓ | BLEURT↑ |
|---|---|---|---|---|---|
| SGD | 41.2 | 0.63 | 0.59 | 0.52 | 0.33 |
| Scaled GD | 43.8 | **0.66** | 0.61 | 0.50 | 0.38 |
| LoRA-Pro SGD | 44.1 | **0.66** | 0.61 | 0.50 | 0.38 |
| SoLoRA SGD (ours) | **44.6** | **0.66** | **0.62** | **0.49** | **0.39** |
| AdamW | 43.9 | 0.66 | 0.60 | 0.50 | 0.38 |
| Scaled AdamW | 44.8 | **0.67** | **0.62** | 0.49 | **0.40** |
| LoRA-Pro AdamW | 44.9 | 0.66 | **0.62** | 0.50 | 0.39 |
| SoLoRA AdamW (ours) | **45.4** | **0.67** | 0.60 | **0.49** | **0.40** |

## A.2  EXPERIMENTAL RESULTS FOR DIFFERENT MODEL SIZE

To more comprehensively validate the advantages of SoLoRA, we conduct experiments not only on the small GPT-2 model but also on GPT-2 models of varying sizes for broader evaluation. All models are fine-tuned with rank 4, and the evaluation results are presented in Table 2. The specific parameter settings can be found in Table 3 and Table 4. By testing on models of different sizes, the experimental

results clearly demonstrate that SoLoRA significantly outperforms other algorithms, regardless of whether the SGD optimizer or the AdamW optimizer is used. This further confirms the effectiveness and stability of the SoLoRA algorithm, enabling it to maintain excellent performance across models of different sizes and under different optimizers.

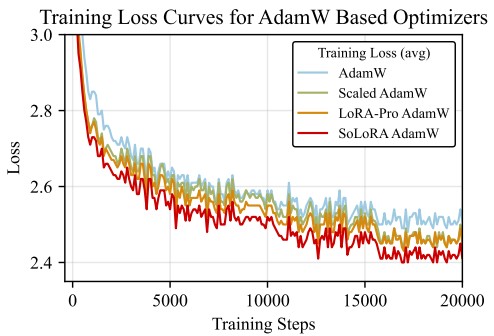
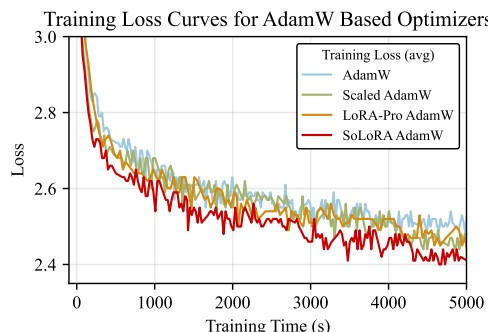

(a) Training loss curve over training steps when fine-tuning using AdamW-based method.

(b) Training loss curve over training time when fine-tuning using AdamW-based method.

Figure 2: Training loss of GPT-2 small model ($r = 64$) fine-tuned using different AdamW-based optimizers. Evaluation is conducted on E2E Natural Language Generation Challenge.

### A.3 TRAINING LOSS CURVE USING DIFFERENT OPTIMIZERS

To further explore the performance advantages of SoLoRA, we compare the runtime of different optimizers when fine-tuning large language models, with the results shown in Figures 1 and 2. These results strongly demonstrate the significant efficiency improvements achieved by the SoLoRA method in fine-tuning tasks. Additionally, Figure 3 illustrates the stability of SoLoRA under different learning rates. The experimental results show that SoLoRA maintains stable performance across a wide range of learning rates, which is crucial for parameter tuning in practical applications. To more comprehensively evaluate the stability of SoLoRA, we also compare it with Scaled AdamW and LoRA-Pro AdamW, under varying learning rates. The results are presented in Figure 3. The comparison reveals that SoLoRA exhibits superior stability across different ranks and learning rates. This indicates that SoLoRA is not only insensitive to changes in learning rates but also robust across varying LoRA ranks. As a result, it reduces the difficulty of hyperparameter tuning and enhances its practicality in fine-tuning.

### A.4 TRAINING EFFICIENCY COMPARISON

To validate the training and inference efficiency of SoLoRA, we report in the table below the total training time required for all algorithms on the GPT-2 small model (rank 64). In addition, we recorded the relationship between training time and the number of steps in Figure 4.

Training and Inference Time of GPT-2 small model (rank=64) fine-tuned using different optimizers. Evaluation is conducted on E2E dataset.

| Methods | SGD | Scaled GD | LoRA-Pro SGD | SoLoRA SGD |
|---|---|---|---|---|
| Total Training Time (Hours) | 1.79 | 1.92 | 2.78 | 2.04 |
| Total Inference Time (Hours) | 1.86 | 1.87 | 1.58 | 1.89 |

| Methods | AdamW | Scaled AdamW | LoRA-Pro AdamW | SoLoRA AdamW |
|---|---|---|---|---|
| Total Training Time (Hours) | 1.79 | 1.94 | 2.93 | 2.04 |
| Total Inference Time (Hours) | 1.87 | 1.88 | 1.89 | 1.86 |

To further validate this, we record GPU memory consumption when the optimizer is called and after the backward is called (fine-tune GPT-2 small model with rank as 4). The results are summarized below.

GPU Memory occupied of GPT-2 small model (rank=4) fine-tuned using different optimizers. Evaluation is conducted on E2E dataset.

| Methods | SGD | Scaled GD | LoRA-Pro SGD | SoLoRA SGD |
|---|---|---|---|---|
| During optimizer computation (MB) | 1395.48 | 1395.57 | 1395.62 | 1401.01 |
| After backward (MB) | 1395.48 | 1395.48 | 1395.48 | 1395.62 |
| Memory Complexity | 0 | 0 | 0 | m+n |
| Methods | AdamW | Scaled AdamW | LoRA-Pro AdamW | SoLoRA AdamW |
| During optimizer computation (MB) | 1396.63 | 1396.72 | 1529.97 | 1462.51 |
| After backward (MB) | 1396.60 | 1396.60 | 1510.98 | 1457.12 |
| Memory Complexity | (m+n)r | (m+n)r | 2mn | mn+m+n |

The results confirm that the memory usage of Algorithm 1 is comparable to other algorithms. Specifically, SoLoRA SGD (Algorithm 1) increases memory usage by only $(1401.01-1395.62)/1395.62 = 0.386\%$ compared to LoRA-SGD. However, with this slight increase in memory, Algorithm 1 demonstrates an effective improvement, as shown in Table 2.

Table 2: Scores of GPT-2 small and medium models ($r = 4$) fine-tuned using different optimizers. Evaluation is conducted on E2E Natural Language Generation challenge. See Appendix A.2 for experimental details.

| Model | Method | E2E | | | | |
|---|---|---|---|---|---|---|
| | | BLEU | NIST | MET | ROUGE-L | CIDEr |
| GPT-2 small | SGD | 54.8 | 4.56 | 34.0 | 63.3 | 1.29 |
| | Scaled GD | 68.5 | 8.72 | 45.5 | 69.4 | 2.40 |
| | LoRA-Pro SGD | 68.4 | 8.72 | 45.5 | 69.6 | 2.43 |
| | SoLoRA SGD (ours) | **69.5** | **8.77** | **46.5** | **71.5** | **2.50** |
| | AdamW | 69.1 | 8.75 | 46.0 | 70.5 | 2.47 |
| | Scaled AdamW | 69.5 | 8.80 | 46.2 | 70.9 | 2.48 |
| | LoRA-Pro AdamW | 69.2 | 8.73 | 45.9 | 70.8 | 2.47 |
| | SoLoRA AdamW (ours) | **70.0** | **8.84** | **46.3** | **71.3** | **2.50** |
| GPT-2 medium | SGD | 66.6 | 8.54 | 44.2 | 68.2 | 2.32 |
| | Scaled GD | 69.2 | 8.71 | 46.3 | 70.9 | 2.48 |
| | LoRA-Pro SGD | 69.7 | 8.77 | 46.5 | 70.9 | 2.50 |
| | SoLoRA SGD (ours) | **70.3** | **8.84** | **46.9** | **71.7** | **2.54** |
| | AdamW | 68.9 | 8.69 | 46.5 | 71.3 | 2.51 |
| | Scaled AdamW | 69.6 | 8.77 | 46.6 | **71.8** | 2.52 |
| | LoRA-Pro AdamW | 69.8 | 8.78 | 46.5 | 71.7 | 2.52 |
| | SoLoRA AdamW (ours) | **70.3** | **8.84** | **46.7** | **71.8** | **2.53** |

## A.5 PARAMETER SETTINGS

To ensure the reproducibility of the experiments described in Section 4 and to facilitate verification and comparison by others, we provide the complete details of the experimental parameter settings. Tables 3 and 4 list the parameters used during the fine-tuning of GPT-2 models and the learning rates corresponding to different optimizers, respectively. Specifically, we conduct experiments with GPT-2 models of various sizes. "Rank 4 (M)" represents a medium-sized model using LoRA with rank 4, while "Rank 4", "Rank 16", and "Rank 64" represent small models using LoRA with ranks 4, 16, and 64, respectively. To ensure the fairness of the experimental setup, we follow the parameter settings in LoRA (Hu et al., 2022) and Riemannian Preconditioned LoRA (Zhang and Pilanci, 2024). However,

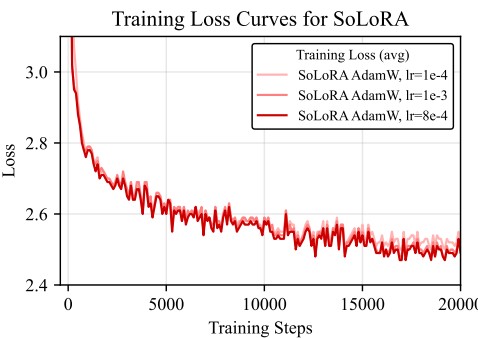
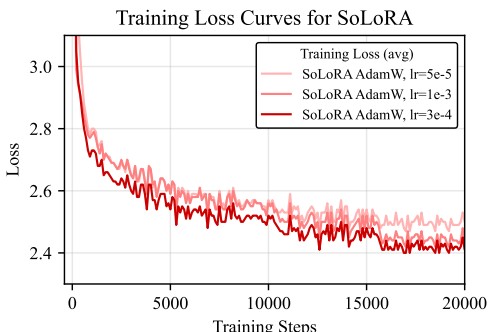

(a) Training loss curve over training step when fine-tuning using SoLoRA AdamW with different learning rates. LoRA rank is 16, with 8e-4 being the optimal learning rate.

(b) Training loss curve over training step when fine-tuning using SoLoRA AdamW with different learning rates. LoRA rank is 64, with 3e-4 being the best learning rate.

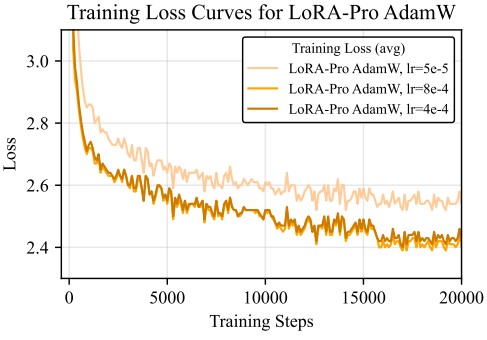
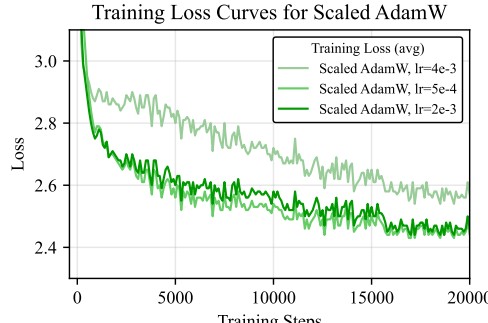

(c) Training loss curve over training step when fine-tuning using LoRA-Pro AdamW with different learning rates. LoRA rank is 64, with 4e-4 being the best learning rate.

(d) Training loss curve over training step when fine-tuning using Scaled AdamW with different learning rates. LoRA rank is 64, with 2e-3 being the best learning rate.

Figure 3: Training loss curve over training step of GPT-2 small model ($r = 16$ and $64$) fine-tuned using different learning rates. Evaluation is conducted on E2E Natural Language Generation Challenge. Our optimizer is stable across different learning rates under varying ranks.

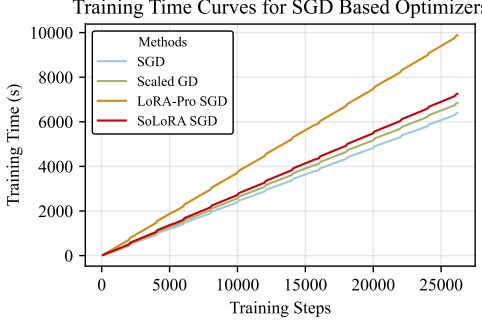
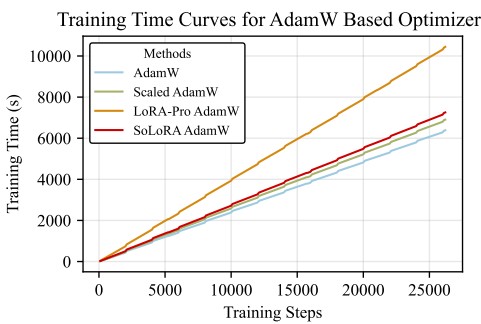

(a) Training Time curve over training steps when fine-tuning using SGD-based method.

(b) Training Time curve over training steps when fine-tuning using AdamW-based method.

Figure 4: Training Time of GPT-2 small model ($r = 64$) fine-tuned using different optimizers. Evaluation is conducted on E2E Natural Language Generation Challenge.

considering the sensitivity of different optimizers to learning rates, we use a grid search strategy to independently tune the optimal learning rate for each optimizer. This ensures that each optimizer operates under its best-performing configuration, providing more objective and reliable experimental results.

Table 3: Training and Inference Configuration for GPT-2 Fine-tuning.

| Training | | LoRA $\alpha$ | | Inference | |
|---|---|---|---|---|---|
| Parameter | Value | Parameter | Value | Parameter | Value |
| Dropout Probability | 0.1 | | | | |
| Batch Size | 8 | | | | |
| Number of Epochs | 5 | $\alpha$ (for Rank 4) | 32 | Beam Size | 10 |
| Warm-up Steps | 500 | $\alpha$ (for Rank 16) | 32 | Length Penalty | 0.8 |
| Learning Rate Scheduler | Linear | $\alpha$ (for Rank 64) | 128 | No Repeat Ngram Size | 4 |
| Label Smoothing | 0.1 | | | | |
| Weight Decay | 0.01 | | | | |

Table 4: Core Optimizer Parameters for GPT-2 fine-tuning.

| Methods | Learning Rate ($\times 10^{-3}$) | | | | $\beta_3$ | $\beta_1 = \beta_2$ |
|---|---|---|---|---|---|---|
| | Rank 4 | Rank 4 (M) | Rank 16 | Rank 64 | | |
| SGD | 90 | 90 | 200 | 90 | / | / |
| Scaled GD | 20 | 20 | 40 | 10 | / | / |
| LoRA-Pro SGD | 40 | 40 | 40 | 40 | / | / |
| SoLoRA SGD | 0.05 | 0.05 | 0.5 | 0.8 | / | 0.98 |
| AdamW | 0.2 | 0.2 | 0.2 | 0.2 | 0.9 | 0.999 |
| Scaled AdamW | 0.8 | 0.8 | 2 | 4 | 0.7 | 0.8 |
| LoRA-Pro AdamW | 0.1 | 0.1 | 0.2 | 0.4 | 0.9 | 0.999 |
| SoLoRA AdamW | 0.5 | 0.1 | 0.8 | 0.3 | 0.9 | 0.98 |

# B SUPPLEMENTARY EXPERIMENTS OF DIFFUSION MODEL FINE-TUNING

As diffusion models increasingly become the mainstream method in image generation, LoRA plays an indispensable role in personalization and style transfer for specific characters. It demonstrates unique advantages, particularly in terms of parameter efficiency, training stability, and rapid convergence. To systematically evaluate the effectiveness of our optimizer SoLoRA in such personalized generation scenarios, we conduct experiments using the Mix-of-Show framework (Gu et al., 2023). This framework integrates Embedding Decomposed LoRA (EDLoRA) into the model, which further reduces the number of trainable parameters while maintaining expressive power. This design better aligns with the dual demands of computational efficiency and generalization stability in real-world applications. To ensure reproducibility and fair comparison, we follow the training and inference settings from (Zhang and Pilanci, 2024; Gu et al., 2023). Specifically, we disable fine-tuning of all embedding vectors and only fine-tune LoRA-related components of the text encoder and U-Net submodules.

Our evaluation encompasses two main aspects: quantitative assessment of the generated images based on objective metrics, as detailed in Section B.1, and qualitative demonstrations of the generated images to visually showcase the effectiveness of the optimizer. For qualitative evaluation, we use examples of Harry Potter and Hermione Granger to visually compare the performance of different optimizers in terms of identity preservation, scene conformity with prompt, and style diversity. As shown in Section B.2 and Section B.3, we conduct image generation under different LoRA scaling factors and compare the performance of various optimizers across multiple learning rates. This design not only evaluates the robustness of the optimizers under multi-scale hyperparameters but also reflects their overall impact on generation quality and consistency in real-world scenarios.

All experimental results consistently demonstrate the advantages of the SoLoRA algorithm. Both the quantitative evaluation metrics and the qualitative image demonstrations highlight the superior performance of SoLoRA. This success can be attributed to the ability of SoLoRA to effectively leverage the second-order information of the loss function, enabling more precise updates to model parameters. Furthermore, the low-rank approximations of gradients derived from our proposed adaptive weighted gradient strategy bring the performance of low-rank fine-tuning closer to that of full-parameter fine-tuning. This allows SoLoRA to achieve comparable performance to full-parameter fine-tuning while significantly reducing computational costs.

## B.1 EVALUATION METRICS OF DIFFUSION MODELS

For quantitative evaluation, we employ two metrics: CLIP score (Hessel et al., 2021) and FID (Heusel et al., 2017). The CLIP score, based on the ViT-B/32 variant of the CLIP model (Radford et al., 2021), measures the consistency between the generated images and the input text prompts. The score ranges from 0 to 100, with higher scores indicating better alignment between the generated image and the text prompt. On the other hand, FID assesses the similarity between the distribution of generated images and the reference images. Lower FID values indicate higher similarity and better overall image quality. The experimental results are shown in Table 5.

In terms of FID, regardless of whether the SGD or AdamW optimizer is used, or whether the scaling factor is 0.7 or 1, our algorithm consistently achieve significantly lower FID values compared to all other methods. This strongly indicates that the distribution of images generated by our algorithm closely matches the distribution of the reference images. For the CLIP score, our algorithm achieve the best performance when the scaling factor is set to 1, outperforming all other methods. However, when the scaling factor is 0.7, the CLIP score of our algorithm is comparable to those of LoRA-Pro and AdamW algorithm. It is important to note that this does not imply that the quality of the images generated by our algorithm is inferior to others. On the contrary, this highlights one of the key strengths of our algorithm: by effectively leveraging second-order information from the loss function, the images generated by our method SoLoRA, not only maintain strong relevance to the text prompt but also exhibit richer details and greater diversity. For instance, in Figure 7, the clothing worn by the generated Harry Potter characters is more diverse, incorporating features that go beyond the simple text prompt. Similarly, in Figure 9, in addition to generating Harry Potter wearing a brown hat, our algorithm introduces more varied gestures for Harry Potter. These richer and more diverse features, while potentially causing a slight decrease in the CLIP score (as CLIP tends to prioritize strict prompt-image alignment and might not fully reward additional details beyond the prompt), actually enhance the overall quality and creativity of the generated images.

Table 5: CLIP and FID scores of different optimizers with different scaling factors for Mix-of-Show.

| Methods | scaling=0.7 | | scaling=1 | |
|---|---|---|---|---|
| | CLIP↑ | FID↓ | CLIP↑ | FID↓ |
| SGD | 27.79 | 69.90 | 31.40 | 40.95 |
| Scaled GD | 31.23 | 35.86 | 30.60 | 29.62 |
| LoRA-Pro SGD | **31.47** | 34.30 | 30.48 | 29.19 |
| SoLoRA SGD (ours) | **31.47** | **30.17** | **31.58** | **28.18** |
| AdamW | **31.47** | 34.15 | 30.68 | 27.80 |
| Scaled AdamW | 24.21 | 48.23 | 24.51 | 34.18 |
| LoRA-Pro AdamW | 31.04 | 29.18 | 30.60 | 28.18 |
| SoLoRA AdamW (ours) | **31.47** | **29.01** | **30.73** | **27.13** |

## B.2 EXPERIMENTAL RESULTS FOR DIFFERENT LoRA SCALING FACTORS

To validate the effectiveness of the proposed optimizer, we compared the generated images of models trained using each optimizer under different LoRA scaling factors $s$. For the sake of fairness, we employed the optimal parameters for each optimizer, detailed in Table 6 for ease of replication.

Figure 5 and 6 show the generated results for Harry Potter and Hermione Granger when fine-tuning the model using different AdamW-based optimizers, with the scaling factor set to 1.0. Figure 7 and 8 show the model's generated results when fine-tuned using different SGD-based optimizers, with scaling factors uniformly set to 1.0. Figure 9 and 10 present the generated results by using different AdamW-based optimizers, employing the scaling factor of 0.7. Experimental results demonstrate that the models trained with our optimizer generate high-quality images, accurately reproducing the identity of Harry Potter and Hermione Granger while demonstrating diverse scene layouts adhering to the input prompts.

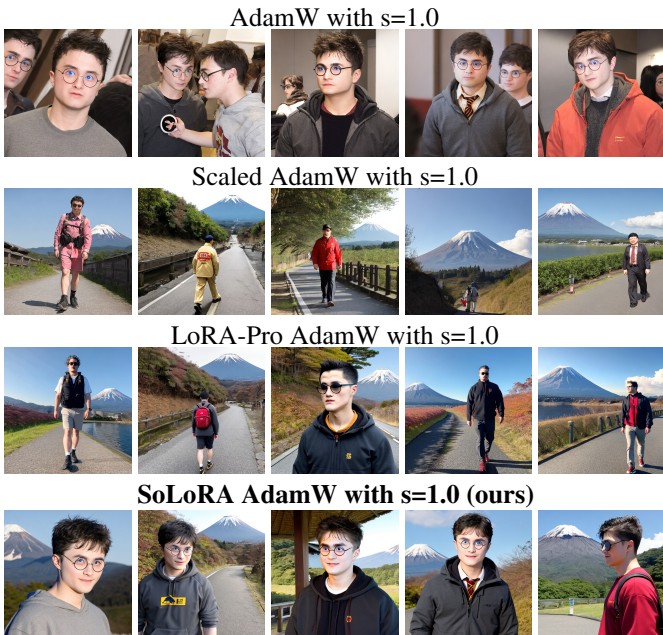

Figure 5: Generated results based on the prompt "Harry Potter is walking near Mount Fuji" when fine-tuned using AdamW-based optimizers. All optimizers employed a LoRA scaling factor of 1.0, with the best learning rate. The results indicate that the output of the model trained with our optimizer incorporates the character "Harry Potter", the action "walking", and the scene "Mount Fuji", yielding superior image quality compared to alternative approaches.

### B.3 Experimental Results for Different Learning Rates

To illustrate the stability of the proposed optimizer, we fix the scaling factor to 1.0 and conduct experiments for each optimizer when using different learning rates. For AdamW-based optimizers, we set AdamW to employ the "Small LR" learning rate combination of 5e-6 and 5e-5 for text-encoder and U-Net, and the "Large LR" learning rate combination of 1e-5 and 1e-4. For Scaled AdamW, LoRA-Pro AdamW, and SoLoRA AdamW, we employed the same learning rate combinations, the "Small LR" of 5e-6 and 5e-6, and the "Large LR" combination of 1e-5 and 1e-5. For SGD-based optimizers, SGD, Scaled GD, and LoRA-Pro SGD, we employ the "Small LR" combination of 1e-2 and 1e-2, and the "Large LR" combination of 1e-1 and 1e-1, whereas SoLoRA SGD utilized the "Small LR" combination of 5e-6 and 5e-6, and the "Large LR" combination of 1e-5 and 1e-5.

The experimental results, presented in Figures 11 and 12, illustrate the effectiveness of our proposed optimizer across both small and large learning rates. This consistent performance signifies a higher degree of stability compared to the alternatives. Such stability is paramount when fine-tuning diffusion models, as their training is characterized by a non-stationary loss landscape. Therefore, the optimizer's ability to remain effective under varying learning rates makes it a robust and advantageous choice for this application.

AdamW with s=1.0

Scaled AdamW with s=1.0

LoRA-Pro AdamW with s=1.0

**SoLoRA AdamW with s=1.0 (ours)**

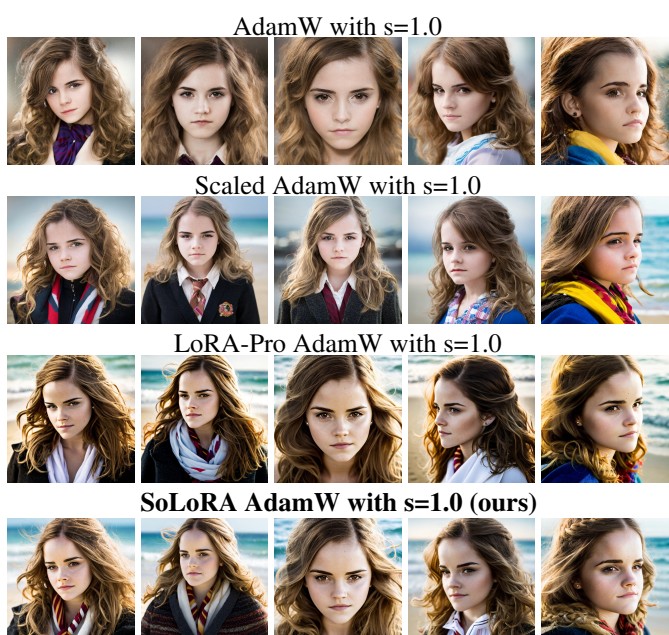

Figure 6: Generation results from the prompt "A photo of Hermione Granger on the beach, small waves, detailed symmetric face, beautiful composition" using AdamW-based optimizers. All the optimizers apply LoRA scaling factor as 1.0, with the best learning rate. Results demonstrate that the model trained with our optimizer generates higher-quality images than others, especially the face of Hermione Granger and the scene.

SGD with s=1.0

Scaled GD with s=1.0

LoRA-Pro SGD with s=1.0

**SoLoRA SGD with s=1.0 (ours)**

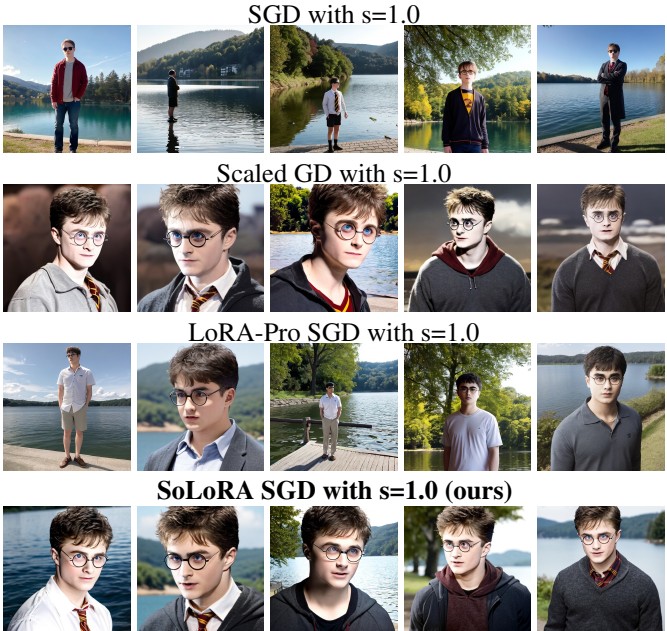

Figure 7: Generated results based on the prompt "Harry Potter standing near the lake" when fine-tuned using SGD-based optimizers. All optimizers employed a LoRA scaling scaling factor of 1.0, with the best learning rate. Results demonstrate that the output images of the model trained with our optimizer have higher-quality than others, especially the face of Harry Potter.

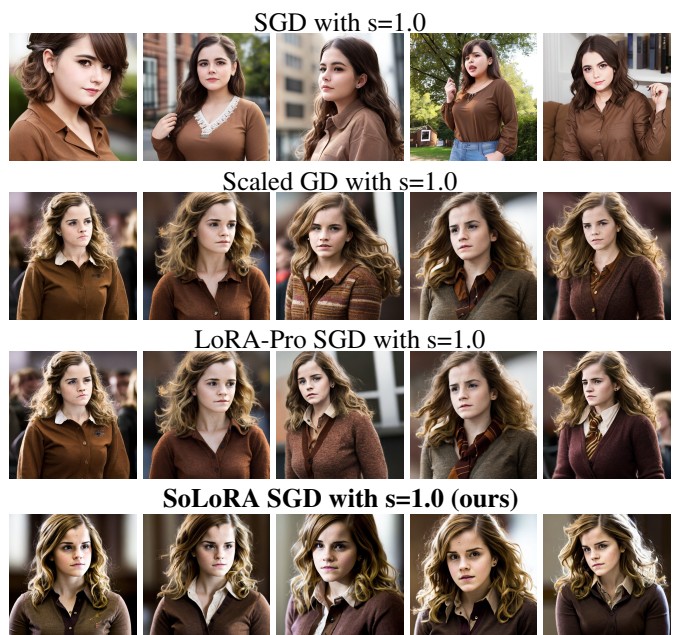

Figure 8: Generated results based on the prompt "Hermione Granger wearing a brown shirt" when fine-tuned using SGD-based optimizers. All optimizers employed a LoRA scaling factor of 1.0, with the best learning rate. Results demonstrate that the model trained with SoLoRA generates higher-quality images than others, especially the face of Hermione Granger.

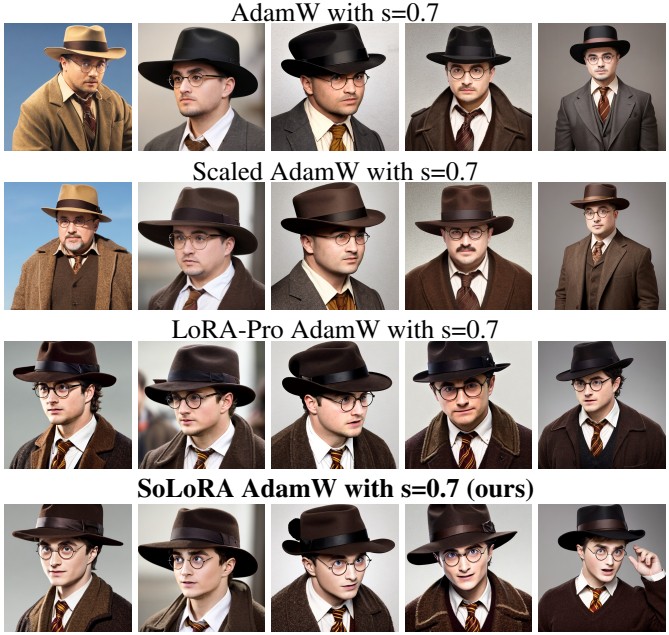

Figure 9: Generated results based on the prompt "Harry Potter wearing a brown hat" when fine-tuned using AdamW-based optimizers. All optimizers employed a LoRA scaling factor of 0.7, with the best learning rate. The results indicate that the output of the model trained with SoLoRA incorporates the character "Harry Potter", and the "hat", yielding superior image quality compared to alternative approaches.

AdamW with s=0.7

Scaled AdamW with s=0.7

LoRA-Pro AdamW with s=0.7

**SoLoRA AdamW with s=0.7 (ours)**

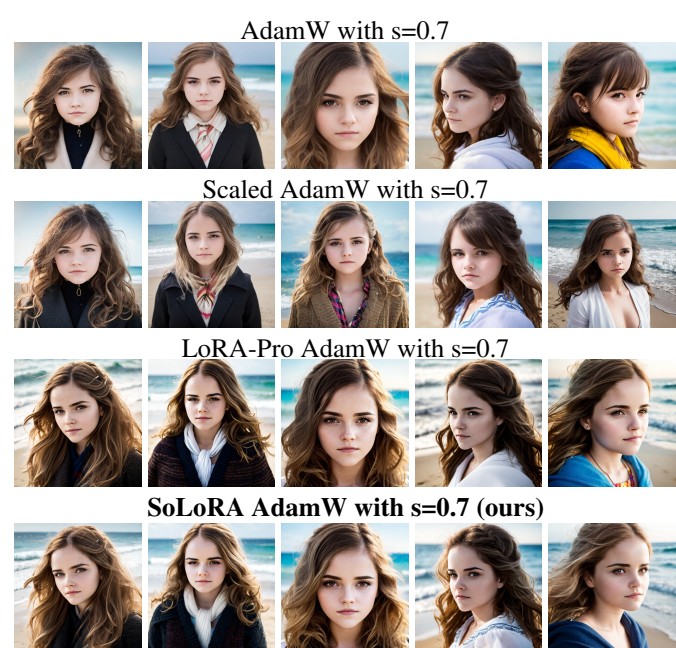

Figure 10: Generation results from the prompt "A photo of Hermione Granger on the beach, small waves, detailed symmetric face, beautiful composition" using AdamW-based optimizers. All the optimizers apply LoRA scaling factor as 0.7. According to the author's recommendation, the optimizer AdamW and Scaled AdamW utilized a learning rate of 1e-5 for text-encoder and 1e-4 for U-Net, whereas LoRA-Pro AdamW and our SoLoRA optimizer adopted 1e-5 for text-encoder and U-Net. Results demonstrate that SoLoRA generates higher-quality images for both scaling factors than others, including the face of Hermione Granger and the scene.

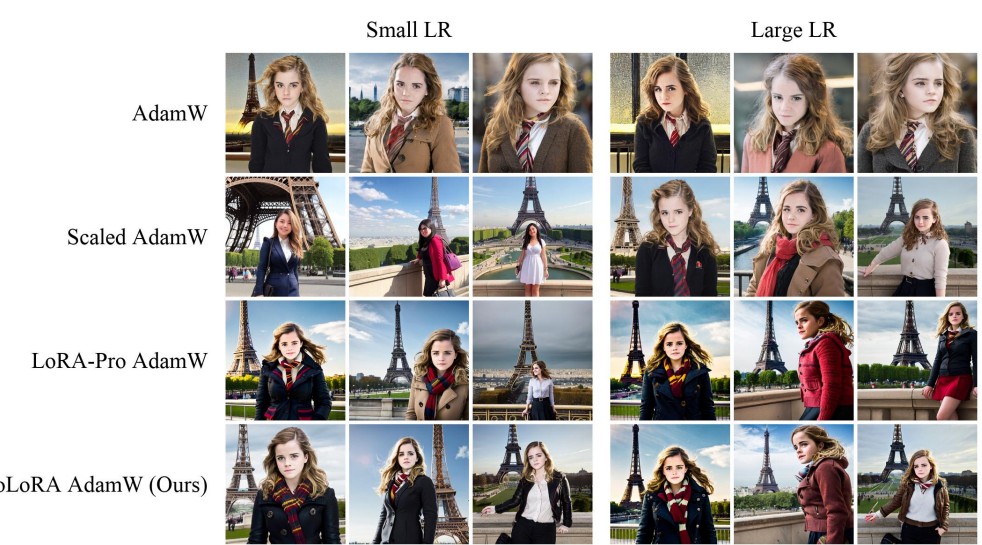

Figure 11: Generated results based on the prompt " Hermione Granger in front of Eiffel Tower" using AdamW-based optimizers. All the optimizers apply LoRA scaling factor as 1.0. "Small LR" and "Large LR" represents using different learning rate, please refer to Appendix B.3 for more details.

Small LR  Large LR

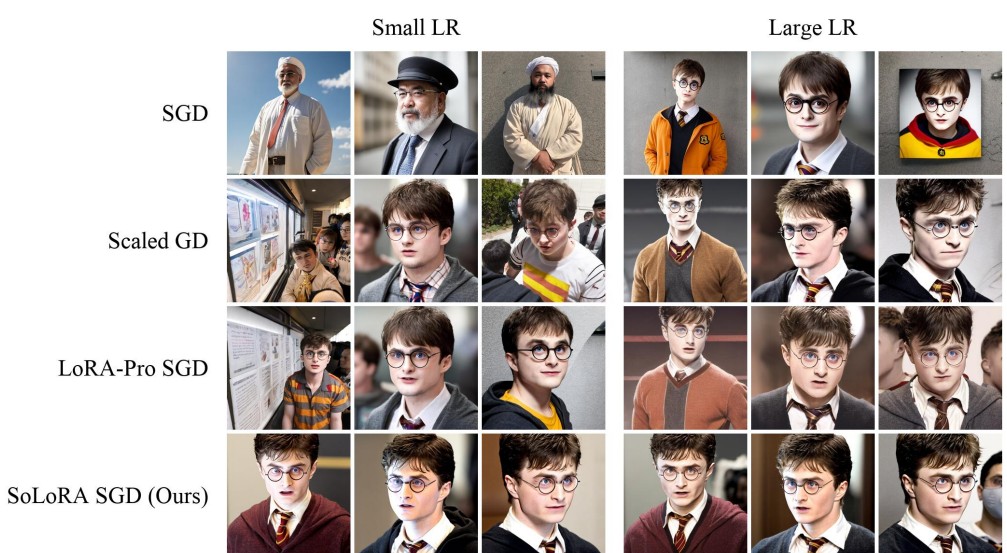

SGD

Scaled GD

LoRA-Pro SGD

SoLoRA SGD (Ours)

Figure 12: Generated results based on the prompt " Photo of Harry Potter" using SGD-based optimizers. All the optimizers apply LoRA scaling factor as 1.0. "Small LR" and "Large LR" represent using different learning rate, please refer to Appendix B.3 for more details.

Table 6: Optimizer Parameters for fine-tuning the Mix-of-Show Model.

| Methods | Learning Rate | | $\beta_3$ | $\beta_1 = \beta_2$ |
| | Text-Encoder | U-Net | | |
| --- | --- | --- | --- | --- |
| SGD | 1e-1 | 1e-1 | / | / |
| Scaled GD | 1e-1 | 1e-1 | / | / |
| LoRA-Pro SGD | 1e-1 | 1e-1 | / | / |
| SoLoRA SGD | 1e-5 | 1e-5 | / | 0.98 |
| AdamW | 1e-5 | 1e-4 | 0.9 | 0.999 |
| Scaled AdamW | 1e-5 | 1e-4 | 0.7 | 0.8 |
| LoRA-Pro AdamW | 1e-5 | 1e-5 | 0.9 | 0.999 |
| SoLoRA AdamW | 1e-5 | 1e-5 | 0.9 | 0.98 |

## C  COMPUTATIONAL AND MEMORY COMPLEXITY ANALYSIS OF SOLORA

The update rule of SoLoRA is given by

$$\boldsymbol{\Delta}_{\boldsymbol{A}_t} = (\boldsymbol{B}_t^\top \boldsymbol{L}_t^{\frac{1}{2}} \boldsymbol{B}_t)^{-1} \underbrace{\boldsymbol{B}_t^\top \boldsymbol{G}_t}_{\boldsymbol{G}_{\boldsymbol{A}_t}} \boldsymbol{R}_t^{-\frac{1}{2}} \left[ \boldsymbol{I} - \frac{1}{2} \boldsymbol{R}_t^{\frac{1}{2}} \boldsymbol{A}_t^\top (\boldsymbol{A}_t \boldsymbol{R}_t^{\frac{1}{2}} \boldsymbol{A}_t^\top)^{-1} \boldsymbol{A}_t \right],$$

$$\boldsymbol{\Delta}_{\boldsymbol{B}_t} = \left[ \boldsymbol{I} - \frac{1}{2} \boldsymbol{B}_t (\boldsymbol{B}_t^\top \boldsymbol{L}_t^{\frac{1}{2}} \boldsymbol{B}_t)^{-1} \boldsymbol{B}_t^\top \boldsymbol{L}_t^{\frac{1}{2}} \right] \boldsymbol{L}_t^{-\frac{1}{2}} \underbrace{\boldsymbol{G}_t \boldsymbol{A}_t^\top}_{\boldsymbol{G}_{\boldsymbol{B}_t}} (\boldsymbol{A}_t \boldsymbol{R}_t^{\frac{1}{2}} \boldsymbol{A}_t^\top)^{-1}.$$

We now analyze the computational complexity of computing the updates $\boldsymbol{\Delta}_{\boldsymbol{A}_t}$ and $\boldsymbol{\Delta}_{\boldsymbol{B}_t}$. For simplicity, we focus on $\boldsymbol{\Delta}_{\boldsymbol{A}_t}$, as the complexity for $\boldsymbol{\Delta}_{\boldsymbol{B}_t}$ is symmetric.

- Compute gradient $\boldsymbol{G}_t$. The stochastic gradient $\boldsymbol{G}_t$ of $\boldsymbol{W}_t$ is obtained during the backpropagation process.

- Row and column sums for $\boldsymbol{l}_t$ and $\boldsymbol{r}_t$. Compute $\boldsymbol{l}_t$ and $\boldsymbol{r}_t$ by summing the square of the element of $\boldsymbol{G}_t$ along rows or columns, which is in the computation $\mathcal{O}(mn)$. $\boldsymbol{L}_t^{\frac{1}{2}}$ and $\boldsymbol{L}_t^{-\frac{1}{2}}$ can computed in $\mathcal{O}(m)$, $\boldsymbol{R}_t^{\frac{1}{2}}$ and $\boldsymbol{R}_t^{-\frac{1}{2}}$ can computed in $\mathcal{O}(n)$.

- Compute $(\boldsymbol{B}_t^\top \boldsymbol{L}_t^{\frac{1}{2}} \boldsymbol{B}_t)^{-1} \boldsymbol{G}_{\boldsymbol{A}_t} \boldsymbol{R}_t^{-\frac{1}{2}}$. First to compute the inverse matrices $(\boldsymbol{B}_t^\top \boldsymbol{L}_t^{\frac{1}{2}} \boldsymbol{B}_t)^{-1}$ in $\mathcal{O}((m+r)r^2)$. Then multiply the inverse $(\boldsymbol{B}_t^\top \boldsymbol{L}_t^{\frac{1}{2}} \boldsymbol{B}_t)^{-1}$ by $\boldsymbol{G}_{\boldsymbol{A}_t}$ in $\mathcal{O}(nr^2)$, and multiply the diagonal matrix $\boldsymbol{R}_t^{-\frac{1}{2}}$ in $\mathcal{O}(nr)$.

- Compute $(\boldsymbol{B}_t^\top \boldsymbol{L}_t^{\frac{1}{2}} \boldsymbol{B}_t)^{-1} \boldsymbol{G}_{\boldsymbol{A}_t} \boldsymbol{A}_t^\top (\boldsymbol{A}_t \boldsymbol{R}_t^{\frac{1}{2}} \boldsymbol{A}_t^\top)^{-1} \boldsymbol{A}_t$. First to compute the inverse matrices $(\boldsymbol{A}_t \boldsymbol{R}_t^{\frac{1}{2}} \boldsymbol{A}_t^\top)^{-1}$ in $\mathcal{O}((n+r)r^2)$. Use the result from the last step, multiply $(\boldsymbol{B}_t^\top \boldsymbol{L}_t^{\frac{1}{2}} \boldsymbol{B}_t)^{-1} \boldsymbol{G}_{\boldsymbol{A}_t}$ by $\boldsymbol{A}_t^\top$ in computation $\mathcal{O}(nr^2)$, then multiply $(\boldsymbol{A}_t \boldsymbol{R}_t^{\frac{1}{2}} \boldsymbol{A}_t^\top)^{-1}$ in computation $\mathcal{O}(r^3)$, and multiply $\boldsymbol{A}_t$ in $\mathcal{O}(nr^2)$.

The computation complexity of $\boldsymbol{\Delta}_{\boldsymbol{A}_t}$ is $\mathcal{O}(mn + (m+n)r^2 + r^3)$. The computation of $\boldsymbol{\Delta}_{\boldsymbol{B}_t}$ follows a similar structure, with symmetric terms. Its complexity is also $\mathcal{O}(mn + (m+n)r^2 + r^3)$. Then we have

- **Per Iteration Computational Complexity.** Combining the computations of $\boldsymbol{\Delta}_{\boldsymbol{A}_t}$ and $\boldsymbol{\Delta}_{\boldsymbol{B}_t}$, the total computation complexity per iteration is $\mathcal{O}(mn + (m+n)r^2 + r^3)$.

- **Memory Complexity.** The algorithm requires storing the vectors $\boldsymbol{l}_t$ and $\boldsymbol{r}_t$ in each iteration, hence the memory complexity is $\mathcal{O}(m+n)$.

## D  PROOF OF THEORETICAL RESULTS

### D.1  COMPUTATION OF JACOBIAN

**Proposition D.1** (Computation of $J_\mathcal{G}$ and $J_\mathcal{G}^*$). *Let $[\boldsymbol{B}, \boldsymbol{A}]$ be a pair of low-rank factors with $\boldsymbol{B} \in \mathbb{R}^{m \times r}, \boldsymbol{A} \in \mathbb{R}^{r \times n}$. Define the generator $\mathcal{G} : [\mathbb{R}^{m \times r}, \mathbb{R}^{r \times n}] \to \mathbb{R}^{m \times n}$ by $\mathcal{G}([\boldsymbol{B}, \boldsymbol{A}]) = \boldsymbol{B} \boldsymbol{A}$. Denote the Jacobian of $\mathcal{G}$ by $J_\mathcal{G}$ and its adjoint by $J_\mathcal{G}^*$. Then, for any $[\boldsymbol{P}, \boldsymbol{Q}] \in [\mathbb{R}^{m \times r}, \mathbb{R}^{r \times n}]$ and any $\boldsymbol{C} \in \mathbb{R}^{m \times n}$,*

- $J_\mathcal{G}([\boldsymbol{B}, \boldsymbol{A}])[\boldsymbol{P}, \boldsymbol{Q}] = \boldsymbol{P} \boldsymbol{A} + \boldsymbol{B} \boldsymbol{Q},$

- $J_\mathcal{G}^*([\boldsymbol{B}, \boldsymbol{A}])(\boldsymbol{C}) = [\boldsymbol{C} \boldsymbol{A}^\top, \boldsymbol{B}^\top \boldsymbol{C}],$

- $J_\mathcal{G}([\boldsymbol{B}, \boldsymbol{A}]) J_\mathcal{G}^*([\boldsymbol{B}, \boldsymbol{A}])(\boldsymbol{C}) = \boldsymbol{C} \boldsymbol{A}^\top \boldsymbol{A} + \boldsymbol{B} \boldsymbol{B}^\top \boldsymbol{C}.$

*Proof.* The Jacobian operator $J_\mathcal{G}([\boldsymbol{B}, \boldsymbol{A}])[\boldsymbol{P}, \boldsymbol{Q}] : [\mathbb{R}^{m \times r}, \mathbb{R}^{r \times n}] \to \mathbb{R}^{m \times n}$ represents the derivative of $\mathcal{G}$ at $[\boldsymbol{B}, \boldsymbol{A}]$ along the direction $[\boldsymbol{P}, \boldsymbol{Q}]$. Similarly, $J_\mathcal{G}^*([\boldsymbol{B}, \boldsymbol{A}])(\boldsymbol{C}) : \mathbb{R}^{m \times n} \to [\mathbb{R}^{m \times r}, \mathbb{R}^{r \times n}]$ is the adjoint of $J_\mathcal{G}$ at $[\boldsymbol{B}, \boldsymbol{A}]$ along the direction $\boldsymbol{C}$. For more details, see (Absil et al., 2009, Section 6.1).

(i) The computation of $J_{\mathcal{G}}$. Let $\boldsymbol{B}(t) : \mathbb{R} \to \mathbb{R}^{m \times r}$ and $\boldsymbol{A}(t) : \mathbb{R} \to \mathbb{R}^{r \times n}$ be differentiable curves with $\boldsymbol{B}(0) = \boldsymbol{B}$ and $\boldsymbol{A}(0) = \boldsymbol{A}$. By the chain rule, the Jacobian of $\mathcal{G}$ at $[\boldsymbol{B}, \boldsymbol{A}]$ along these curves is

$$
\begin{aligned}
J_{\mathcal{G}}([\boldsymbol{B}(t), \boldsymbol{A}(t)])[\dot{\boldsymbol{B}}(t), \dot{\boldsymbol{A}}(t)]\Big|_{t=0} &= \left[\frac{\mathrm{d}\mathcal{G}([\boldsymbol{B}, \boldsymbol{A}])}{\mathrm{d}\boldsymbol{B}}\right]\dot{\boldsymbol{B}}(t)\Big|_{t=0} + \left[\frac{\mathrm{d}\mathcal{G}([\boldsymbol{B}, \boldsymbol{A}])}{\mathrm{d}\boldsymbol{A}}\right]\dot{\boldsymbol{A}}(t)\Big|_{t=0} \\
&= \dot{\boldsymbol{B}}(t)\boldsymbol{A}(t)\Big|_{t=0} + \boldsymbol{B}(t)\dot{\boldsymbol{A}}(t)\Big|_{t=0} \\
&= \dot{\boldsymbol{B}}(0)\boldsymbol{A} + \boldsymbol{B}\dot{\boldsymbol{A}}(0),
\end{aligned}
$$

where $\dot{\boldsymbol{B}}(t)$ and $\dot{\boldsymbol{A}}(t)$ denote the derivatives of $\boldsymbol{B}(t)$ and $\boldsymbol{A}(t)$ with respect to $t$. The second line follows because $\mathcal{G}([\boldsymbol{B}, \boldsymbol{A}]) = \boldsymbol{B}\boldsymbol{A}$, hence $\frac{\mathrm{d}\mathcal{G}([\boldsymbol{B}, \boldsymbol{A}])}{\mathrm{d}\boldsymbol{B}}$ and $\frac{\mathrm{d}\mathcal{G}([\boldsymbol{B}, \boldsymbol{A}])}{\mathrm{d}\boldsymbol{A}}$ are both linear operators.

Since $\dot{\boldsymbol{B}}(0)$ and $\dot{\boldsymbol{A}}(0)$ are arbitrary, for any $[\boldsymbol{P}, \boldsymbol{Q}] \in [\mathbb{R}^{m \times r}, \mathbb{R}^{r \times n}]$, we obtain

$$
J_{\mathcal{G}}([\boldsymbol{B}, \boldsymbol{A}])[\boldsymbol{P}, \boldsymbol{Q}] = \boldsymbol{P}\boldsymbol{A} + \boldsymbol{B}\boldsymbol{Q}.
$$

(ii) The computation of $J_{\mathcal{G}}^*$. For brevity, write $J_{\mathcal{G}}[\boldsymbol{P}, \boldsymbol{Q}]$ for $J_{\mathcal{G}}([\boldsymbol{B}, \boldsymbol{A}])[\boldsymbol{P}, \boldsymbol{Q}]$ and $J_{\mathcal{G}}^*(\boldsymbol{C})$ for $J_{\mathcal{G}}^*([\boldsymbol{B}, \boldsymbol{A}])(\boldsymbol{C})$. By definition of the adjoint (with respect to the Frobenius inner product), for any $[\boldsymbol{P}, \boldsymbol{Q}] \in (\mathbb{R}^{m \times r}, \mathbb{R}^{r \times n})$ and $\boldsymbol{C} \in \mathbb{R}^{m \times n}$,

$$
\langle J_{\mathcal{G}}[\boldsymbol{P}, \boldsymbol{Q}], \boldsymbol{C} \rangle = \langle [\boldsymbol{P}, \boldsymbol{Q}], J_{\mathcal{G}}^*(\boldsymbol{C}) \rangle.
$$

For the left-hand side,

$$
\begin{aligned}
\langle J_{\mathcal{G}}[\boldsymbol{P}, \boldsymbol{Q}], \boldsymbol{C} \rangle &= \langle \boldsymbol{P}\boldsymbol{A} + \boldsymbol{B}\boldsymbol{Q}, \boldsymbol{C} \rangle \\
&= \langle \boldsymbol{P}\boldsymbol{A}, \boldsymbol{C} \rangle + \langle \boldsymbol{B}\boldsymbol{Q}, \boldsymbol{C} \rangle \\
&= \langle \boldsymbol{P}, \boldsymbol{C}\boldsymbol{A}^\top \rangle + \langle \boldsymbol{Q}, \boldsymbol{B}^\top \boldsymbol{C} \rangle.
\end{aligned}
$$

For the right-hand side, writing $J_{\mathcal{G}}^*(\boldsymbol{C}) = [\boldsymbol{C}_1, \boldsymbol{C}_2]$, then

$$
\begin{aligned}
\langle [\boldsymbol{P}, \boldsymbol{Q}], J_{\mathcal{G}}^*(\boldsymbol{C}) \rangle &= \langle [\boldsymbol{P}, \boldsymbol{Q}], [\boldsymbol{C}_1, \boldsymbol{C}_2] \rangle \\
&= \langle \boldsymbol{P}, \boldsymbol{C}_1 \rangle + \langle \boldsymbol{Q}, \boldsymbol{C}_2 \rangle.
\end{aligned}
$$

Hence $\boldsymbol{C}_1 = \boldsymbol{C}\boldsymbol{A}^\top$ and $\boldsymbol{C}_2 = \boldsymbol{B}^\top \boldsymbol{C}$, and therefore $J_{\mathcal{G}}^*([\boldsymbol{B}, \boldsymbol{A}])(\boldsymbol{C}) = [\boldsymbol{C}\boldsymbol{A}^\top, \boldsymbol{B}^\top \boldsymbol{C}]$.

(iii) Finally, $J_{\mathcal{G}}([\boldsymbol{B}, \boldsymbol{A}])J_{\mathcal{G}}^*([\boldsymbol{B}, \boldsymbol{A}])(\boldsymbol{C}) = J_{\mathcal{G}}([\boldsymbol{B}, \boldsymbol{A}])[\boldsymbol{C}\boldsymbol{A}^\top, \boldsymbol{B}^\top \boldsymbol{C}] = \boldsymbol{C}\boldsymbol{A}^\top \boldsymbol{A} + \boldsymbol{B}\boldsymbol{B}^\top \boldsymbol{C}$ as claimed.

$\square$

## D.2 ORTHOGONAL PROJECTION TO TANGENT SPACE

In this subsection, we derive the orthogonal projection onto the tangent space under both the standard metric and the weighted metric. The specific forms of $\boldsymbol{L}_t$ and $\boldsymbol{R}_t$ are presented here and will not be repeated in subsequent propositions and proofs. For the sake of simplicity, the subscript $t$ will be omitted in this subsection.

$$
\begin{aligned}
\boldsymbol{L}_t &= \mathrm{diag}(\boldsymbol{l}_t / \sqrt{\|\boldsymbol{l}_t\|_1}) \text{ with } \boldsymbol{l}_t = \beta_2 \boldsymbol{l}_{t-1} + (1 - \beta_2)\sum_{j=1}^{n}(\boldsymbol{G}_t \odot \boldsymbol{G}_t)_{i,j}, \\
\boldsymbol{R}_t &= \mathrm{diag}(\boldsymbol{r}_t / \sqrt{\|\boldsymbol{r}_t\|_1}) \text{ with } \boldsymbol{r}_t = \beta_3 \boldsymbol{r}_{t-1} + (1 - \beta_3)\sum_{i=1}^{m}(\boldsymbol{G}_t \odot \boldsymbol{G}_t)_{i,j},
\end{aligned}
\tag{13}
$$

where $\odot$ denotes the Hadamard (elementwise) product and $\boldsymbol{G}_t = \nabla\mathcal{L}(\boldsymbol{W}_0 + \boldsymbol{W}_t)$.

**Proposition D.2** (Orthogonal Projection to Tangent Space Under the Standard Metric). *Let $\boldsymbol{W} \in \mathcal{M}_r$ be a rank-$r$ matrix with a low-rank decomposition $\boldsymbol{W} = \boldsymbol{B}\boldsymbol{A}$, where $\boldsymbol{B} \in \mathbb{R}^{m \times r}, \boldsymbol{A} \in \mathbb{R}^{r \times n}$. Denote by $\mathbb{T}_{\boldsymbol{W}}$ the tangent space of the smooth manifold $\mathcal{M}_r$ at the point $\boldsymbol{W}$. Then, the orthogonal projection of any matrix $\boldsymbol{Z} \in \mathbb{R}^{m \times n}$ onto $\mathbb{T}_{\boldsymbol{W}}$ is given by*

$$
\mathcal{P}_{\mathbb{T}_{\boldsymbol{W}}}(\boldsymbol{Z}) = \boldsymbol{B}(\boldsymbol{B}^\top \boldsymbol{B})^{-1}\boldsymbol{B}^\top \boldsymbol{Z} + \boldsymbol{Z}\boldsymbol{A}^\top(\boldsymbol{A}\boldsymbol{A}^\top)^{-1}\boldsymbol{A} - \boldsymbol{B}(\boldsymbol{B}^\top \boldsymbol{B})^{-1}\boldsymbol{B}^\top \boldsymbol{Z}\boldsymbol{A}^\top(\boldsymbol{A}\boldsymbol{A}^\top)^{-1}\boldsymbol{A}.
$$

*Proof.* Suppose $\boldsymbol{W}$ has a compact singular value decomposition, given by $\boldsymbol{W} = \boldsymbol{U\Sigma V}^\top$, where $\boldsymbol{U} \in \mathbb{R}^{m \times r}, \boldsymbol{\Sigma} \in \mathbb{R}^{r \times r}, \boldsymbol{V} \in \mathbb{R}^{n \times r}$. Then the tangent space $\mathbb{T}_{\boldsymbol{W}}$ at $\boldsymbol{W}$ is characterized as

$$\mathbb{T}_{\boldsymbol{W}} = \{\boldsymbol{UM}^\top + \boldsymbol{NV}^\top, \text{for } \boldsymbol{M} \in \mathbb{R}^{m \times r}, \boldsymbol{N} \in \mathbb{R}^{n \times r}\}.$$

Therefore, the orthogonal projection of $\boldsymbol{Z}$ onto $\mathbb{T}_{\boldsymbol{W}}$ is known to be (Wei et al., 2016)

$$\mathcal{P}_{\mathbb{T}_{\boldsymbol{W}}}(\boldsymbol{Z}) = \boldsymbol{UU}^\top \boldsymbol{Z} + \boldsymbol{ZV}^\top \boldsymbol{V} - \boldsymbol{UU}^\top \boldsymbol{ZV}^\top \boldsymbol{V}. \tag{14}$$

Since the columns of $\boldsymbol{B}$ and $\boldsymbol{U}$ span the same column space (i.e., the column space of $\boldsymbol{W}$), then there exists an invertible matrix $\boldsymbol{S} \in \mathbb{R}^{r \times r}$ such that $\boldsymbol{B} = \boldsymbol{US}$ and $\boldsymbol{U} = \boldsymbol{BS}^{-1}$. Using this relation, we have

$$\boldsymbol{U}^\top \boldsymbol{U} = (\boldsymbol{BS}^{-1})^\top \boldsymbol{BS}^{-1} = \boldsymbol{S}^{-\top}(\boldsymbol{B}^\top \boldsymbol{B})\boldsymbol{S}^{-1}.$$

Since $\boldsymbol{U}^\top \boldsymbol{U} = \boldsymbol{I}_r$, it follows that

$$\boldsymbol{S}^{-\top}(\boldsymbol{B}^\top \boldsymbol{B})\boldsymbol{S}^{-1} = \boldsymbol{I}_r \implies \boldsymbol{B}^\top \boldsymbol{B} = \boldsymbol{S}^\top \boldsymbol{S}.$$

Using this, we compute $\boldsymbol{UU}^\top$

$$\boldsymbol{UU}^\top = \boldsymbol{BS}^{-1}\boldsymbol{S}^{-\top}\boldsymbol{B} = \boldsymbol{B}(\boldsymbol{S}^\top \boldsymbol{S})^{-1}\boldsymbol{B}^\top = \boldsymbol{B}(\boldsymbol{B}^\top \boldsymbol{B})^{-1}\boldsymbol{B}^\top \tag{15}$$

Similarly, since the rows of $\boldsymbol{A}$ and the columns of $\boldsymbol{V}$ span the same row space (i.e., the row space of $\boldsymbol{W}$), there exists an invertible matrix $\boldsymbol{Q} \in \mathbb{R}^{r \times r}$ such that $\boldsymbol{A} = \boldsymbol{QV}^\top$ and $\boldsymbol{V}^\top = \boldsymbol{Q}^{-1}\boldsymbol{A}$. Further, using $\boldsymbol{V}^\top \boldsymbol{V} = \boldsymbol{I}_r$, we obtain

$$\boldsymbol{V}^\top \boldsymbol{V} = \boldsymbol{Q}^{-1}(\boldsymbol{AA}^\top)\boldsymbol{Q}^{-\top} = \boldsymbol{I}_r,$$

hence $\boldsymbol{AA}^\top = \boldsymbol{QQ}^\top$ and

$$\boldsymbol{VV}^\top = \boldsymbol{A}^\top \boldsymbol{Q}^{-\top}\boldsymbol{Q}^{-1}\boldsymbol{A} = \boldsymbol{A}^\top(\boldsymbol{QQ}^\top)^{-1}\boldsymbol{A} = \boldsymbol{A}^\top(\boldsymbol{AA}^\top)^{-1}\boldsymbol{A} \tag{16}$$

Substituting (15) and (16) into (14) yields

$$\mathcal{P}_{\mathbb{T}_{\boldsymbol{W}}}(\boldsymbol{Z}) = \boldsymbol{B}(\boldsymbol{B}^\top \boldsymbol{B})^{-1}\boldsymbol{B}^\top \boldsymbol{Z} + \boldsymbol{ZA}^\top(\boldsymbol{AA}^\top)^{-1}\boldsymbol{A} - \boldsymbol{B}(\boldsymbol{B}^\top \boldsymbol{B})^{-1}\boldsymbol{B}^\top \boldsymbol{ZA}^\top(\boldsymbol{AA}^\top)^{-1}\boldsymbol{A}.$$

$$\square$$

**Proposition D.3** (Orthogonal Projection onto the Tangent Space Under the Weighted Metric). *Let $\boldsymbol{W} \in \mathcal{M}_r$ has a low-rank decomposition $\boldsymbol{W} = \boldsymbol{BA}$, where $\boldsymbol{B} \in \mathbb{R}^{m \times r}, \boldsymbol{A} \in \mathbb{R}^{r \times n}$. Denote the tangent space of the Riemannian manifold $\mathcal{M}_r$ at the point $\boldsymbol{W}$ as $\mathbb{T}_{\boldsymbol{W}}$. The weighted metric is defined as $\langle \boldsymbol{Y}, \boldsymbol{Z} \rangle_{\boldsymbol{H}} = \langle \boldsymbol{L}^{\frac{1}{2}}\boldsymbol{Y}\boldsymbol{R}^{\frac{1}{2}}, \boldsymbol{Z} \rangle$ for any $\boldsymbol{Y}, \boldsymbol{Z} \in \mathbb{R}^{m \times n}$. Then, the orthogonal projection of any matrix $\boldsymbol{Z} \in \mathbb{R}^{m \times n}$ onto $\mathbb{T}_{\boldsymbol{W}}$ under the weighed metric is given by*

$$\mathcal{P}_{\mathbb{T}_{\boldsymbol{W}}}(\boldsymbol{Z}) = \boldsymbol{B}(\boldsymbol{B}^\top \boldsymbol{L}^{\frac{1}{2}}\boldsymbol{B})^{-1}\boldsymbol{B}^\top \boldsymbol{L}^{\frac{1}{2}}\boldsymbol{Z} + \boldsymbol{ZR}^{\frac{1}{2}}\boldsymbol{A}^\top(\boldsymbol{AR}^{\frac{1}{2}}\boldsymbol{A}^\top)^{-1}\boldsymbol{A}$$
$$- \boldsymbol{B}(\boldsymbol{B}^\top \boldsymbol{B})^{-1}\boldsymbol{B}^\top \boldsymbol{L}^{\frac{1}{2}}\boldsymbol{ZR}^{\frac{1}{2}}\boldsymbol{A}^\top(\boldsymbol{AA}^\top)^{-1}\boldsymbol{A}.$$

*Proof.* This proof is inspired by (Bian et al., 2024). Here, we briefly provide a sketch of the proof.

(i) *The new orthonormal basis under the weighted metric.* Let $\boldsymbol{W} = \boldsymbol{U\Sigma V}^\top$ be a be a compact SVD with $\boldsymbol{U} = [\boldsymbol{u}_1, \boldsymbol{u}_2, \cdots, \boldsymbol{u}_r] \in \mathbb{R}^{m \times r}, \boldsymbol{V} = [\boldsymbol{v}_1, \boldsymbol{v}_2, \cdots, \boldsymbol{v}_r] \in \mathbb{R}^{n \times r}$. Normalize the singular vectors under the weighted vector

$$\langle \boldsymbol{x}, \boldsymbol{y} \rangle_{\boldsymbol{L}^{\frac{1}{2}}} = \langle \boldsymbol{L}^{\frac{1}{2}}\boldsymbol{x}, \boldsymbol{y} \rangle \text{ in } \mathbb{R}^m \quad \text{and} \quad \langle \boldsymbol{x}, \boldsymbol{y} \rangle_{\boldsymbol{R}^{\frac{1}{2}}} = \langle \boldsymbol{R}^{\frac{1}{2}}\boldsymbol{x}, \boldsymbol{y} \rangle \text{ in } \mathbb{R}^n$$

to obtain

$$\widetilde{\boldsymbol{U}} = \boldsymbol{U}(\boldsymbol{U}^\top \boldsymbol{L}^{\frac{1}{2}}\boldsymbol{U})^{-\frac{1}{2}} := [\tilde{\boldsymbol{u}}_1, \tilde{\boldsymbol{u}}_2, \cdots, \tilde{\boldsymbol{u}}_r] \in \mathbb{R}^{m \times r},$$
$$\widetilde{\boldsymbol{V}} = \boldsymbol{V}(\boldsymbol{V}^\top \boldsymbol{R}^{\frac{1}{2}}\boldsymbol{V})^{-\frac{1}{2}} := [\tilde{\boldsymbol{v}}_1, \tilde{\boldsymbol{v}}_2, \cdots, \tilde{\boldsymbol{v}}_r] \in \mathbb{R}^{n \times r}.$$

Next, we extend $\widetilde{\boldsymbol{U}}$ and $\widetilde{\boldsymbol{V}}$ to full orthonormal basis of $(\mathbb{R}^m, \langle \cdot, \cdot \rangle_{\boldsymbol{L}^{\frac{1}{2}}})$ and $(\mathbb{R}^n, \langle \cdot, \cdot \rangle_{\boldsymbol{R}^{\frac{1}{2}}})$ respectively. Then, an orthonormal basis of $\mathbb{T}_{\boldsymbol{W}}$ with respect to $\langle \cdot, \cdot \rangle_{\boldsymbol{H}_t}$ is $\{\tilde{\boldsymbol{u}}_i \tilde{\boldsymbol{v}}_j^\top\}_{\min\{i,j\} \leq r}$.

(ii) *Orthogonal projection represented by the new orthonormal basis.* Using the orthonormal bases $\widetilde{U}$ and $\widetilde{V}$, the projection of $Z$ onto $\mathbb{T}_W$ is expressed as:

$$
\widetilde{\mathcal{P}}_{\mathbb{T}_W}(Z) = \sum_{(i,j):\min\{i,j\}\leq r} \langle Z, \tilde{u}_i\tilde{v}_j^\top \rangle_{H_t} \cdot \tilde{u}_i\tilde{v}_j^\top = \sum_{(i,j):\min\{i,j\}\leq r} \langle L^{\frac{1}{2}}ZR^{\frac{1}{2}}, \tilde{u}_i\tilde{v}_j^\top \rangle \cdot \tilde{u}_i\tilde{v}_j^\top
$$

$$
= \sum_{(i,j):\min\{i,j\}\leq r} \tilde{u}_i^\top L^{\frac{1}{2}}ZR^{\frac{1}{2}}\tilde{v}_j \cdot \tilde{u}_i\tilde{v}_j^\top
$$

$$
= \widetilde{U}\widetilde{U}^\top L^{\frac{1}{2}}Z + ZR^{\frac{1}{2}}\widetilde{V}\widetilde{V}^\top - \widetilde{U}\widetilde{U}^\top L^{\frac{1}{2}}ZR^{\frac{1}{2}}\widetilde{V}\widetilde{V}^\top.
$$

(iii) *Express the basis projectors via factors $B$ and $A$.* Since $B$ and $A$ span the same spaces as $U$ and $V$, we derive

$$
\widetilde{U}\widetilde{U}^\top = B\left(B^\top L^{\frac{1}{2}}B\right)^{-1}B^\top, \quad \widetilde{V}\widetilde{V}^\top = A^\top\left(AR^{\frac{1}{2}}A^\top\right)^{-1}A.
$$

Substituting these expressions into the formula for $\widetilde{\mathcal{P}}_{\mathbb{T}_W}$, we obtain

$$
\widetilde{\mathcal{P}}_{\mathbb{T}_W}(Z) = B(B^\top L^{\frac{1}{2}}B)^{-1}B^\top L^{\frac{1}{2}}Z + ZR^{\frac{1}{2}}A^\top(AR^{\frac{1}{2}}A^\top)^{-1}A
$$

$$
- B(B^\top L^{\frac{1}{2}}B)^{-1}B^\top L^{\frac{1}{2}}ZR^{\frac{1}{2}}A^\top(AR^{\frac{1}{2}}A^\top)^{-1}A.
$$

$\square$

**Proposition D.4.** *Suppose $W \in \mathcal{M}_r$ has a low-rank decomposition $W = BA$, where $B \in \mathbb{R}^{m\times r}$ and $A \in \mathbb{R}^{r\times n}$. For any matrix $M \in \mathbb{R}^{m\times r}$, $N \in \mathbb{R}^{r\times n}$, the matrix $MA + BN$ lies in the tangent space $\mathbb{T}_W$ at $W$ of $\mathcal{M}_r$ at the point $W$.*

*Proof.* Let $W \in \mathcal{M}_r$ has a compact singular value decomposition $W = U\Sigma V^\top$, where $U \in \mathbb{R}^{m\times r}, \Sigma \in \mathbb{R}^{r\times r}$, and $V \in \mathbb{R}^{n\times r}$. By definition, the tangent space $\mathbb{T}_W$ at $W$ is given by

$$
\mathbb{T}_W = \{UK_1^\top + K_2V^\top | K_1 \in \mathbb{R}^{n\times r}, K_2 \in \mathbb{R}^{m\times r}\}.
$$

Since $B$ and $A$ are low-rank factors of $W$, there exist invertible matrices $S \in \mathbb{R}^{r\times r}$ and $Q \in \mathbb{R}^{r\times r}$ such that

$$
B = US, \quad A = QV^\top.
$$

Substituting these expressions, the matrix $MA + BN$ can be rewritten as

$$
MA + BN = MQV^\top + USN.
$$

The first term, $MQV^\top$, lies in span$(V^\top)$, and the second term, $USN$, lies in span$(U)$. Thus, the sum $MQV^\top + USN$ lies in the tangent space $\mathbb{T}_W$ by the definition of the tangent space. Then, it follows that $MA + BN$ is on the tangent space $\mathbb{T}_W$. This completes the proof. $\square$

### D.3 PROOFS OF THEOREM 3.1 AND THEOREM 3.2

**Proof of Theorem 3.1.** Define

$$
\Gamma(\Delta_{B_t}, \Delta_{A_t}) := \frac{1}{2}\|\Delta_{B_t}A_t + B_t\Delta_{A_t} - \widetilde{\mathcal{P}}_{\mathbb{T}_t}(L_t^{-\frac{1}{2}}G_tR_t^{-\frac{1}{2}})\|_{H_t}^2.
$$

Differentiating $\Gamma(\Delta_{B_t}, \Delta_{A_t})$ with respect to $\Delta_{B_t}$ and $\Delta_{A_t}$ yields

$$
\nabla_{\Delta_{B_t}}\Gamma(\Delta_{B_t}, \Delta_{A_t}) = L_t^{\frac{1}{2}}\Delta_{B_t}(A_tR_t^{\frac{1}{2}}A_t^\top) + L_t^{\frac{1}{2}}B_t\Delta_{A_t}R_t^{\frac{1}{2}}A_t^\top - G_tA_t^\top, \tag{17}
$$

and

$$
\nabla_{\Delta_{A_t}}\Gamma(\Delta_{B_t}, \Delta_{A_t}) = B_t^\top L_t^{\frac{1}{2}}\Delta_{B_t}A_tR_t^{\frac{1}{2}} + B_t^\top L_t^{\frac{1}{2}}B_t\Delta_{A_t}R_t^{\frac{1}{2}} - B_t^\top G_t. \tag{18}
$$

Setting $\nabla_{\Delta_{B_t}}\Gamma(\Delta_{B_t}, \Delta_{A_t}) = 0$ and using the invertibility of $(A_tR_t^{\frac{1}{2}}A_t^\top)$ and $L_t^{\frac{1}{2}}$ gives

$$
\Delta_{B_t} = L_t^{-\frac{1}{2}}G_tA_t^\top(A_tR_t^{\frac{1}{2}}A_t^\top)^{-1} - B_t\Delta_{A_t}R_t^{\frac{1}{2}}A_t^\top(A_tR_t^{\frac{1}{2}}A_t^\top)^{-1}. \tag{19}
$$

Substituting (18) into $\nabla_{\boldsymbol{\Delta}_{A_t}}\Gamma(\boldsymbol{\Delta}_{B_t}, \boldsymbol{\Delta}_{A_t}) = \boldsymbol{0}$ and using the invertibility of $\boldsymbol{B}_t^\top \boldsymbol{L}_t^{\frac{1}{2}} \boldsymbol{B}_t$ and $\boldsymbol{R}_t^{\frac{1}{2}}$ yields

$$\boldsymbol{\Delta}_{A_t}[\boldsymbol{I} - \widetilde{\boldsymbol{Q}}_{A_t}] = (\boldsymbol{B}_t^\top \boldsymbol{L}_t^{\frac{1}{2}} \boldsymbol{B}_t)^{-1} \boldsymbol{B}_t^\top \boldsymbol{G}_t \boldsymbol{R}_t^{-\frac{1}{2}}[\boldsymbol{I} - \widetilde{\boldsymbol{Q}}_{A_t}],$$

where $\widetilde{\boldsymbol{Q}}_{A_t} = \boldsymbol{R}_t^{\frac{1}{2}} \boldsymbol{A}_t^\top (\boldsymbol{A}_t \boldsymbol{R}_t^{\frac{1}{2}} \boldsymbol{A}_t^\top)^{-1} \boldsymbol{A}_t$, which is the projection matrix onto the row space of $\boldsymbol{A}_t$. Since $\boldsymbol{I} - \widetilde{\boldsymbol{Q}}_{A_t}$ is the residual maker matrix, then a general solution is

$$\boldsymbol{\Delta}_{A_t}^{\mathrm{opt}} = (\boldsymbol{B}_t^\top \boldsymbol{L}_t^{\frac{1}{2}} \boldsymbol{B}_t)^{-1} \boldsymbol{B}_t^\top \boldsymbol{G}_t \boldsymbol{R}_t^{-\frac{1}{2}} + \boldsymbol{X}_t \boldsymbol{A}_t,$$

with arbitrary matrix $\boldsymbol{X}_t \in \mathbb{R}^{r\times r}$. Plugging this $\boldsymbol{\Delta}_{A_t}$ back into (19) gives

$$\boldsymbol{\Delta}_{B_t}^{\mathrm{opt}} = [\boldsymbol{I} - \widetilde{\boldsymbol{P}}_{B_t}]\boldsymbol{L}_t^{-\frac{1}{2}} \boldsymbol{G}_t \boldsymbol{A}_t^\top (\boldsymbol{A}_t \boldsymbol{R}_t^{\frac{1}{2}} \boldsymbol{A}_t^\top)^{-1} - \boldsymbol{B}_t \boldsymbol{X}_t,$$

where $\widetilde{\boldsymbol{P}}_{B_t} = \boldsymbol{B}_t(\boldsymbol{B}_t^\top \boldsymbol{L}_t^{\frac{1}{2}} \boldsymbol{B}_t)^{-1}\boldsymbol{B}_t^\top \boldsymbol{L}_t^{\frac{1}{2}}$, which is the projection matrix onto the column space of $\boldsymbol{B}_t$. $\square$

**Proof of Theorem 3.2.** Let the objective function be $\Psi(\boldsymbol{X}_t) = \frac{1}{2}\|\boldsymbol{\Delta}_{B_t}\boldsymbol{A}_t - \boldsymbol{B}_t\boldsymbol{\Delta}_{A_t}\|_{\boldsymbol{H}_t}^2$. To minimize $\Psi(\boldsymbol{X}_t)$, we compute its gradient with respect to $\boldsymbol{X}_t$,

$$\nabla_{\boldsymbol{X}_t}\Psi(\boldsymbol{X}_t) = \boldsymbol{B}_t^\top \boldsymbol{L}_t^{\frac{1}{2}}(\boldsymbol{\Delta}_{B_t}\boldsymbol{A}_t - \boldsymbol{B}_t\boldsymbol{\Delta}_{A_t})\boldsymbol{R}_t^{\frac{1}{2}}\boldsymbol{A}^\top.$$

Substituting the expressions for $\boldsymbol{A}_t$ and $\boldsymbol{B}_t$ from Theorem 3.1, we have

$$\nabla_{\boldsymbol{X}_t}\Psi(\boldsymbol{X}_t) = \boldsymbol{B}_t^\top \boldsymbol{L}_t^{\frac{1}{2}}\Big( [\boldsymbol{I} - \boldsymbol{B}_t(\boldsymbol{B}_t^\top \boldsymbol{L}_t^{\frac{1}{2}}\boldsymbol{B}_t)^{-1}\boldsymbol{B}_t^\top \boldsymbol{L}_t^{\frac{1}{2}}]\boldsymbol{L}_t^{-\frac{1}{2}}\boldsymbol{G}_t\boldsymbol{A}_t^\top(\boldsymbol{A}_t\boldsymbol{R}_t^{\frac{1}{2}}\boldsymbol{A}_t^\top)^{-1}\boldsymbol{A}_t$$
$$- \boldsymbol{B}_t(\boldsymbol{B}_t^\top \boldsymbol{L}_t^{\frac{1}{2}}\boldsymbol{B}_t)^{-1}\boldsymbol{B}_t^\top \boldsymbol{G}_t\boldsymbol{R}^{-\frac{1}{2}} - 2\boldsymbol{B}_t\boldsymbol{X}_t\boldsymbol{A}_t\Big)\boldsymbol{R}_t^{\frac{1}{2}}\boldsymbol{A}^\top$$
$$= -\boldsymbol{B}_t^\top \boldsymbol{G}_t\boldsymbol{A}_t - 2(\boldsymbol{B}_t^\top \boldsymbol{L}_t^{\frac{1}{2}}\boldsymbol{B}_t)\boldsymbol{X}_t(\boldsymbol{A}_t\boldsymbol{R}_t^{\frac{1}{2}}\boldsymbol{A}_t^\top).$$

Setting $\nabla_{\boldsymbol{X}_t}\Psi(\boldsymbol{X}_t) = \boldsymbol{0}$, we obtain

$$-\boldsymbol{B}_t^\top \boldsymbol{G}_t\boldsymbol{A}_t = 2(\boldsymbol{B}_t^\top \boldsymbol{L}_t^{\frac{1}{2}}\boldsymbol{B}_t)\boldsymbol{X}_t(\boldsymbol{A}_t\boldsymbol{R}_t^{\frac{1}{2}}\boldsymbol{A}_t^\top).$$

Since $\boldsymbol{B}_t^\top \boldsymbol{L}_t^{\frac{1}{2}}\boldsymbol{B}_t$ and $\boldsymbol{A}_t\boldsymbol{R}_t^{\frac{1}{2}}\boldsymbol{A}_t^\top$ are invertible, we solve for $\boldsymbol{X}_t$ as

$$\boldsymbol{X}_t^{\mathrm{opt}} = -\frac{1}{2}(\boldsymbol{B}_t^\top \boldsymbol{L}_t^{\frac{1}{2}}\boldsymbol{B}_t)^{-1}\boldsymbol{B}_t^\top \boldsymbol{G}_t\boldsymbol{A}_t^\top(\boldsymbol{A}_t\boldsymbol{R}_t^{\frac{1}{2}}\boldsymbol{A}_t^\top)^{-1}.$$

Thus, the optimal solution for $\boldsymbol{X}_t$ is derived. $\square$

### D.4 PROOF OF THE LOSS FUNCTION EXHIBITS A DECREASING TREND

Under the framework of LoRA, the infinitesimal change in loss is

$$d\mathcal{L} = \langle \boldsymbol{G}_B, d\boldsymbol{B}\rangle_{\boldsymbol{H}} + \langle \boldsymbol{G}_A, d\boldsymbol{A}\rangle_{\boldsymbol{H}}$$
$$= -\eta(\langle \boldsymbol{G}_B, \boldsymbol{\Delta}_B\rangle_{\boldsymbol{H}} + \langle \boldsymbol{G}_A, \boldsymbol{\Delta}_A\rangle_{\boldsymbol{H}})$$
$$= -\eta\Big(\langle \boldsymbol{G}_B, [\boldsymbol{I} - \boldsymbol{B}(\boldsymbol{B}^\top \boldsymbol{L}^{\frac{1}{2}}\boldsymbol{B})^{-1}\boldsymbol{B}^\top \boldsymbol{L}^{\frac{1}{2}}]\boldsymbol{L}^{-\frac{1}{2}}\boldsymbol{G}_B(\boldsymbol{A}\boldsymbol{R}^{\frac{1}{2}}\boldsymbol{A}^\top)^{-1} - \boldsymbol{B}\boldsymbol{X}\rangle_{\boldsymbol{H}}$$
$$+ \langle \boldsymbol{G}_A, (\boldsymbol{B}^\top \boldsymbol{L}^{\frac{1}{2}}\boldsymbol{B})^{-1}\boldsymbol{G}_A\boldsymbol{R}^{-\frac{1}{2}} + \boldsymbol{X}\boldsymbol{A}\rangle_{\boldsymbol{H}}\Big)$$
$$= -\eta\Big(\langle \boldsymbol{G}_B, [\boldsymbol{I} - \boldsymbol{B}(\boldsymbol{B}^\top \boldsymbol{L}^{\frac{1}{2}}\boldsymbol{B})^{-1}\boldsymbol{B}^\top \boldsymbol{L}^{\frac{1}{2}}]\boldsymbol{L}^{-\frac{1}{2}}\boldsymbol{G}_B(\boldsymbol{A}\boldsymbol{R}^{\frac{1}{2}}\boldsymbol{A}^\top)^{-1}\rangle_{\boldsymbol{H}} + \langle \boldsymbol{G}_A, (\boldsymbol{B}^\top \boldsymbol{L}^{\frac{1}{2}}\boldsymbol{B})^{-1}\boldsymbol{G}_A\boldsymbol{R}^{-\frac{1}{2}}\rangle_{\boldsymbol{H}}$$
$$+ \langle \boldsymbol{G}_B, -\boldsymbol{B}\boldsymbol{X}\rangle_{\boldsymbol{H}} + \langle \boldsymbol{G}_A, \boldsymbol{X}\boldsymbol{A}\rangle_{\boldsymbol{H}}\Big)$$
$$= -\eta\Big(\langle \boldsymbol{G}_B, [\boldsymbol{I} - \boldsymbol{B}(\boldsymbol{B}^\top \boldsymbol{L}^{\frac{1}{2}}\boldsymbol{B})^{-1}\boldsymbol{B}^\top \boldsymbol{L}^{\frac{1}{2}}]\boldsymbol{L}^{-\frac{1}{2}}\boldsymbol{G}_B(\boldsymbol{A}\boldsymbol{R}^{\frac{1}{2}}\boldsymbol{A}^\top)^{-1}\rangle_{\boldsymbol{H}} + \langle \boldsymbol{G}_A, (\boldsymbol{B}^\top \boldsymbol{L}^{\frac{1}{2}}\boldsymbol{B})^{-1}\boldsymbol{G}_A\boldsymbol{R}^{-\frac{1}{2}}\rangle_{\boldsymbol{H}}\Big),$$

$$(20)$$

Where the third line is derived from Theorem 3.1, the forth line is because of the additivity of the inner product, and the last line is because

$$\langle G_B, -BX \rangle_H + \langle G_A, XA \rangle_H = \langle -B^\top G_B + G_A A^\top, X \rangle_H$$
$$= \langle -B^\top G A^\top + B^\top G A^\top, X \rangle_H$$
$$= 0.$$

Therefore, to prove $d\mathcal{L} \leq 0$, it suffices to prove that

$$\langle G_B, [I - B(B^\top L^{\frac{1}{2}}B)^{-1}B^\top L^{\frac{1}{2}}L^{-\frac{1}{2}}G_B(AR^{\frac{1}{2}}A^\top)^{-1} \rangle_H \geq 0,$$
$$\langle G_A, (B^\top L^{\frac{1}{2}}B)^{-1}G_A R^{-\frac{1}{2}} \rangle_H \geq 0.$$

1.  First to prove $\langle G_B, [I - B(B^\top L^{\frac{1}{2}}B)^{-1}B^\top L^{\frac{1}{2}}]L^{-\frac{1}{2}}G_B(AR^{\frac{1}{2}}A^\top)^{-1} \rangle_H \geq 0$.

    *   Prove $[I - B(B^\top L^{\frac{1}{2}}B)^{-1}B^\top L^{\frac{1}{2}}]$ is symmetric and positive semi-definite.

        From Equation (10), we have $\widetilde{P}_B = B(B^\top L^{\frac{1}{2}}B)^{-1}B^\top L^{\frac{1}{2}}$. $\widetilde{P}_B$ is symmetric in the weighted space if it is self-adjoint with respect to the weighted inner product. That means we need to prove for all vectors $x, y \in \mathbb{R}^m$

        $$\langle \widetilde{P}_B x, y \rangle_H = \langle x, \widetilde{P}_B y \rangle_H$$

        For the left-hand side,

        $$\langle \widetilde{P}_B x, y \rangle_H = \langle L^{\frac{1}{2}}\widetilde{P}_B x, y \rangle = \langle L^{\frac{1}{2}}B(B^\top L^{\frac{1}{2}}B)^{-1}B^\top L^{\frac{1}{2}}x, y \rangle.$$

        For the right-hand side,

        $$\langle x, \widetilde{P}_B y \rangle_H = \langle L^{\frac{1}{2}}x, \widetilde{P}_B y \rangle = \langle \widetilde{P}_B^\top L^{\frac{1}{2}}x, y \rangle = \langle L^{\frac{1}{2}}B(B^\top L^{\frac{1}{2}}B)^{-1}B^\top L^{\frac{1}{2}}x, y \rangle.$$

        Thus, the left-hand side equals to the right-hand side, $\widetilde{P}_B$ is symmetric.
        By Proposition D.3, $\widetilde{P}_B$ is the orthogonal projection under the inner product $\langle \cdot, \cdot \rangle_H$. As a projection matrix, the eigenvalues of $\widetilde{P}_B$ are zeros and ones. Thus, $\widetilde{P}_B^\top$ and $I - \widetilde{P}_B^\top$ are both positive semi-definite matrices.
        There exists a Cholesky decomposition of $I - \widetilde{P}_B^\top$, denote as $I - \widetilde{P}_B^\top = KK^\top$, where $K \in \mathbb{R}^{m \times m}$ is a lower triangular matrix.

    *   Prove $(AR^{\frac{1}{2}}A^\top)^{-1}$ is symmetric and positive definite.

        Since $R$ is a diagonal matrix, then symmetric, and therefore $AR^{\frac{1}{2}}A^\top$ is also symmetric.
        For any non-zero vector $x \in \mathbb{R}^r$, $x^\top AR^{\frac{1}{2}}A^\top x = \langle A^\top x, A^\top x \rangle_H = \|A^\top x\|_H^2 > 0$.
        Thus, $AR^{\frac{1}{2}}A^\top$ is a positive definite matrix.
        Therefore, $AR^{\frac{1}{2}}A^\top$ is invertible and $(AR^{\frac{1}{2}}A^\top)^{-1}$ is symmetric and positive definite.
        There exists a Cholesky decomposition of $(AR^{\frac{1}{2}}A^\top)^{-1}$, denote as $(AR^{\frac{1}{2}}A^\top)^{-1} = CC^\top$, where $C \in \mathbb{R}^{r \times r}$ is a lower triangular matrix.

    Since $[I - B(B^\top L^{\frac{1}{2}}B)^{-1}B^\top L^{\frac{1}{2}}]$ and $(AR^{\frac{1}{2}}A^\top)^{-1}$ both have a Cholesky decomposition. The inner product can be rewritten as

    $$\langle G_B, [I - B(B^\top L^{\frac{1}{2}}B)^{-1}B^\top L^{\frac{1}{2}}]L^{-\frac{1}{2}}G_B(AR^{\frac{1}{2}}A^\top)^{-1} \rangle_H$$
    $$= \langle G_B, KK^\top L^{-\frac{1}{2}}G_B CC^\top \rangle_H$$
    $$= \langle L^{-\frac{1}{2}}KK^\top G_B, G_B CC^\top \rangle_H$$
    $$= \langle KK^\top G_B, G_B CC^\top \rangle$$
    $$= \langle K^\top G_B C, K^\top G_B C \rangle$$
    $$= \|K^\top G_B C\|_F^2$$
    $$\geq 0$$

2. Then prove $\langle \boldsymbol{G_A}, (\boldsymbol{B}^\top \boldsymbol{L}^{\frac{1}{2}} \boldsymbol{B})^{-1} \boldsymbol{G_A} \boldsymbol{R}^{-\frac{1}{2}} \rangle_H \geq 0$.

Similar to $(\boldsymbol{A} \boldsymbol{R}^{\frac{1}{2}} \boldsymbol{A}^\top)^{-1}$ is symmetric and positive definite, $(\boldsymbol{B}^\top \boldsymbol{L}^{\frac{1}{2}} \boldsymbol{B})^{-1}$ is symmetric positive definite, and there exists a Cholesky decomposition of $(\boldsymbol{B}^\top \boldsymbol{L}^{\frac{1}{2}} \boldsymbol{B})^{-1}$, denote as $(\boldsymbol{B}^\top \boldsymbol{L}^{\frac{1}{2}} \boldsymbol{B})^{-1} = \boldsymbol{D} \boldsymbol{D}^\top$, where $\boldsymbol{D} \in \mathbb{R}^{r \times r}$ is a lower triangular matrix.

Thus, the inner product can be rewritten as

$$
\begin{aligned}
\langle \boldsymbol{G_A}, (\boldsymbol{B}^\top \boldsymbol{L}^{\frac{1}{2}} \boldsymbol{B})^{-1} &\boldsymbol{G_A} \boldsymbol{R}^{-\frac{1}{2}} \rangle_H \\
&= \langle \boldsymbol{G_A}, \boldsymbol{D} \boldsymbol{D}^\top \boldsymbol{G_A} \boldsymbol{R}^{-\frac{1}{2}} \rangle_H \\
&= \langle \boldsymbol{G_A} \boldsymbol{R}^{\frac{1}{2}}, \boldsymbol{D} \boldsymbol{D}^\top \boldsymbol{G_A} \boldsymbol{R}^{-\frac{1}{2}} \rangle \\
&= \langle \boldsymbol{G_A}, \boldsymbol{D} \boldsymbol{D}^\top \boldsymbol{G_A} \rangle \\
&= \langle \boldsymbol{D}^\top \boldsymbol{G_A}, \boldsymbol{D}^\top \boldsymbol{G_A} \rangle \\
&= \| \boldsymbol{D}^\top \boldsymbol{G_A} \|_F^2 \\
&\geq 0
\end{aligned}
$$

In the conclusion, we have completed the proof of $d\mathcal{L} \leq 0$. This ensures that the SoLoRA model maintains a decreasing trend in the loss.

## E    ADDITIONAL EXPERIMENTS

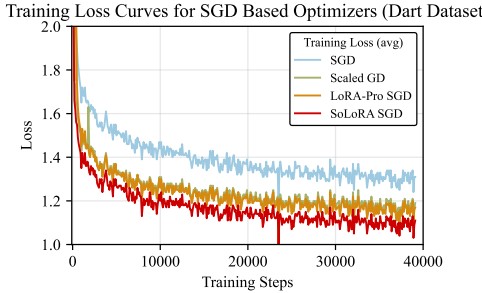

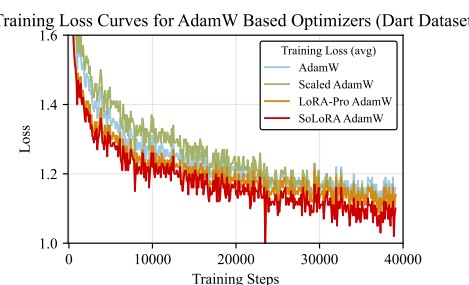

(a) Training loss curve over training steps when fine-tuning using SGD-based method.

(b) Training loss curve over training time when fine-tuning using AdamW-based method.

Figure 13: Training loss of GPT-2 small model ($r = 4$) fine-tuned using different optimizers. Evaluation is conducted on the DART Dataset.

## F    THE USE OF LARGE LANGUAGE MODELS (LLMS)

We use LLMs to polish writing.

