# OpenReview forum: "LoRA Meets Second-Order Optimization: Towards Optimal Low-Rank Updates"
_ICLR.cc/2026/Conference — Submitted to ICLR 2026_

### Official Review · Reviewer_jxGJ · 2025-10-21

**Soundness:** 2
**Presentation:** 2
**Contribution:** 2
**Rating:** 2
**Confidence:** 4

**Summary:**

The proposed method, SoLoRA, extends LoRA-Pro by introducing a metric defined as a weighted average of gradients in the same spirit of AdaGrad and Adam. The approach is evaluated through fine-tuning experiments on GPT-2 and Mix-of-Show models.

**Strengths:**

The proposed approach generalizes LoRA-Pro and demonstrates improved performance in the evaluated cases.

**Weaknesses:**

W1. Conceptually, what motivates the introduction of the gradient-based metric to modify LoRA’s updates? If the intent is to capture second-order information, why not directly employ established second-order optimization methods (e.g., SOAP or Shampoo)? Moreover, comparing the results in Figs. 1 and 2 shows that the improvement under AdamW is considerably smaller than under SGD, suggesting that AdamW may already serve a similar role to the proposed approach.

W2. The specific design choices behind SoLoRA are not sufficiently justified. Why is the $\ell_1$ norm adopted in Eq. (4)? There are several alternative and reasonable options, such as using an $\ell_2$ norm or omitting normalization altogether.

W3. In terms of novelty, the proposed approach appears to reduce to LoRA-Pro when setting $L_t = R_t = I$. Moreover, the idea of using a moving average to adaptively precondition gradients is well established in the literature. As a result, the conceptual distinction between SoLoRA and existing methods remains unclear. The authors are encouraged to clarify the new insights or mechanisms introduced by the proposed metric beyond LoRA-Pro, and to demonstrate concretely how it results in different optimization dynamics or empirical improvements.

W4. Several recent works closely related to this paper are not cited or discussed (e.g., [1, 2]). Both methods are more computationally efficient than the proposed approach, yet the paper does not acknowledge or compare against them. A discussion on how SoLoRA differs from or improves upon these methods would strengthen the contribution.

W5. The experimental validation appears relatively small in scope. Only GPT-2 Small and GPT-2 Medium are tested, which are modest in size by today’s standards. This makes it difficult to assess whether the proposed method scales effectively to larger models. Moreover, the evaluation is conducted on a single dataset (E2E), leaving open the question of how the approach generalizes to other tasks or domains.

W6. The performance gain under AdamW is quite limited compared to baseline methods such as LoRA-Pro, further weakening the empirical support for the proposed design.

W7. The paper does not explicitly report the runtime or memory consumption of the proposed method. Since SoLoRA aims to improve training efficiency, such metrics are essential to substantiate its practical benefits. A quantitative comparison with baseline methods in terms of computational cost and memory usage would make the evaluation more complete.

W8. The writing can be improved significantly.
- The description in Section 2.1 contains inaccuracies. For instance, the statement that “LoRA was designed to avoid the expensive SVD computation at each training step” (lines 95 - 96) is not historically correct. LoRA predates GaLore, and was not motivated by avoiding SVD computations. The authors are encouraged to revise this section for historical and conceptual accuracy.
- Line 94: "leading to time-consuming"
- Line 242 - 243: By the definition of eq.(4), $L_t$ and $R_t$ are clearly not rank-1.


[1] Zhang Y, Li B, Giannakis GB. RefLoRA: Refactored Low-Rank Adaptation for Efficient Fine-Tuning of Large Models. arXiv preprint arXiv:2505.18877. 2025.

[2] Yen JN, Si S, Meng Z, Yu F, Duvvuri SS, Dhillon IS, Hsieh CJ, Kumar S. LoRA Done RITE: Robust Invariant Transformation Equilibration for LoRA Optimization. arXiv preprint arXiv:2410.20625. 2024.

**Questions:**

See above.

---

> ### Author Response · Authors · 2025-11-20
> **Response to Reviewer jxGJ: Part 1**
>
> Thank you for your valuable and detailed comments! We are glad that you find that SoLoRA introduced a weighted metric. In the following, we provide our point-by-point response and hope our response could address your concerns. We also look forward to the subsequent discussion which may further help solve the current issues.
>
> **w1**
>
> **Motivation for introducing the weighted metric:**
>
> + Outer product of gradients can approximate to Hessian: As described in [3], the outer product of gradients can be regarded as an approximation of the Hessian matrix. The second-order moment of AdamW/AdaGrad—namely, the outer product of gradients—thus constitutes an approximation of the Hessian matrix. Specifically, the second-order moment can be vectorized as $\text{diag}(g \otimes g^\top) = \text{diag}(gg^\top)$, where $g$ is the vectorized gradient.
> + Gauss Newton approximation to Hessian: In [4], it has been proved that the preconditioner $L$ and $R$ maintained by Shampoo can be viewed either as an approximation of the Gauss–Newton component of the Hessian.
> + Effectiveness of the weighted metric in matrix recovery tasks: In [5], it has been demonstrated that weighted metrics based on the approximation of the Hessian matrix can accelerate the convergence.
> + The approximation of the Hessian is applied in the Taylor expansion of the loss function: In Section 3.2, we demonstrated the necessity of weighted norms through the second-order Taylor expansion of the loss function.
>
> **Difference with SOAP/Shampoo:**
>
> + SOAP requires computing QR decomposition, with computational complexity of $\mathcal{O}(m^3)$, while matrices $L$ and $R$ require computing $GG^\top$and $G^\top G$, with computational complexity of $\mathcal{O}(m^2n)$ and $\mathcal{O}(mn^2)$, where $G \in \mathbb{R}^{m \times n}$.
> + Shampoo also construct the matrix $L$ and $R$ by computing $GG^\top$ and $G^\top G$, which have computational complexity $\mathcal{O}(m^2n)$ and $\mathcal{O}(mn^2)$, where $G \in \mathbb{R}^{m \times n}$.
> + Given the high computational complexity of SOAP/Shampoo and the simplicity of AdaGrad/AdamW, we construct $L$ and $R$ using intermediate forms. Specifically, $L=\text{diag}(GG^\top)$ and $R=\text{diag}(G^\top G)$. Computation can be achieved through element-wise matrix multiplication, followed by summation along rows or columns.
>
> **Comparison of AdamW methods:**
>
> + Figure 1 presents a comparison of SGD-based methods, with the implementation details of SoLoRA SGD outlined in Algorithm 1. Figure 2 displays a comparison of AdamW-based methods, where the implementation details of SoLoRA AdamW are outlined in Algorithm 2.
> + The distinction between Algorithm 1 and Algorithm 2 lies in the first-order momentum, not AdamW.
> + Algorithm 2 is named SoLoRA AdamW, as its implementation resembles AdamW, combining first-order and second-order momentum.
> + Figure 2 demonstrates that SoLoRA outperforms other methods in both training loss and training efficiency.
> + To further validate the effectiveness of SoLoRA, we plot the relationship between training time and training steps in **Figure 4 (Page 14).**
> + All the implementations of SoLoRA and baseline methods are in `Supplementary Material` for reviewers to check.
>
> **w2**
>
> Thank you very much for your valuable feedback on our paper. Your suggestions have greatly helped us improve our paper.
>
> The origin of the $ l_1$ norm is from the perspective of the update of the weighted matrix $W$. In Section 2.1, we provide a new perspective on existing factorization-based algorithms of LoRA and propose a unified factorization-based update framework:
> $$
> W_{t+1} \approx W_t - \alpha_t J_\mathcal{G} ([B_t, A_t]) J_\mathcal{G}^* ([B_t, A_t]) \nabla_{W_t} \mathcal{L}(W_0 + W_t)
> $$
> Within this framework, we analyze how the convergence rate of factorization-based algorithms is influenced by the condition number of the Jacobian operator $J_\mathcal{G} J_\mathcal{G}^* $, which in turn depends on the condition numbers of $B$ and $A$. Based on this observation, we identify a gap between factorization-based algorithms and algorithms that directly update the full weight matrix $W$.
>
> Furthermore, to accelerate the convergence, we employed the approximation of the Hessian of $W$ to serve as the preconditioner. As described in [2], the idealized Shampoo applied the preconditioner $(GG^\top) \otimes (G^\top G) / \text{trace}(GG^\top)$, where $\text{trace}(GG^\top)=\text{trace}(G^\top G)$ can serve as a normalization to the Kronecker product of $(G G^\top)$ and $(G^\top G)$. The normalizatio term is exactly the summation of diagonal terms of $GG^\top$ or $G^\top G$, also the $l_1$ norm of vectorized $\text{diag}(GG^\top)$ or $\text{diag}(G^\top G)$, i.e. $\text{trace}(GG^\top) = || \text{vec}(\text{diag}(GG^\top)) ||_1$ and $\text{trace}(G^\top G) = || \text{vec}(\text{diag}(G^\top G)) ||_1$.

---

> ### Author Response · Authors · 2025-11-20
> **Response to Reviewer jxGJ: Part 2**
>
> In our algorithm, the weighted metric is derived from $\text{diag}(GG^\top)$ and $\text{diag}(G^\top G)$ to reduce the computation complexity. Therefore, we employ the square root of $|| \text{vec}(\text{diag}(GG^\top)) ||_1$ and $|| \text{vec}(\text{diag}(G^\top G)) ||_1$ to regularize $\text{diag}(GG^\top)$ and $\text{diag}(G^\top G)$, respectively.
> $$
> \begin{split}
> L & =\text{diag}(GG^\top/\sqrt{|| \text{vec}(\text{diag}(GG^\top)) ||_1}), \\
> R & =\text{diag}(G^\top G/\sqrt{|| \text{vec}(\text{diag}(G^\top G)) ||_1}),
> \end{split}
> $$
> where we omit the momentum term. After considering the momentum, Equation (4) of our paper can be derived.
>
> **w3**
>
> Thank you very much for your valuable feedback on our paper. Your suggestions have greatly helped us improve our paper.
>
> First, we want to clarify that even when setting $L=R=I$, SoLoRA and LoRA-Pro still exhibit differences. Specifically, when $L=R=I$, Algorithm 1 (SoLoRA SGD) in our paper becomes SGD. The similarity to LoRA-Pro SGD lies in both updating $B$ and $A$ using Riemannian gradient descent (RGD) on the full weight matrix $W$. However, the key difference is that we do not need to solve the Sylvester equation, significantly saving computation time. This can be verified in **Figure 1**, which shows the relation of training loss and step/time. And the additional **Figure 4**, which shows the relation between training time and step. And **the following table**, which shows the total training time. Additionally, when $L=R=I$, Algorithm 2 (SoLoRA AdamW) in our paper becomes SGD with momentum (SGDM), requiring only the storage of the optimizer state for $\mathcal{O}(mn)$. In contrast, LoRA-Pro AdamW applies both first- and second-order momentum from AdamW to the Riemann gradient for $W$ (LoRA-Pro Algorithm 2 lines 10-11), requiring twice the optimizer state storage ($\mathcal{O}(2mn)$). Moreover, SoLoRA AdamW also avoids solving the Sylvester equation, achieving substantial time savings.
>
> Second, we want to explain the meaning of $L$ and $R$. The adaptive metric is defined as $L=\text{diag}(GG^\top)$ and $R=\text{diag}(G^\top G)$, both of which approximate the diagonal elements of the Hessian matrix. This approximation is derived from the Generalized Gauss-Newton (GGN) matrix and is a computationally efficient method for incorporating curvature information into optimization algorithms. [6] presents the GGN approximation and discusses its properties in neural network optimization, emphasizing how the GGN matrix, often based on the outer product of gradients, serves as an approximation to the Hessian.
>
> We sincerely thank you again for your feedback, which has helped us more comprehensively validate the effectiveness of our algorithm and improve the overall quality of the paper.
>
> **w4**
>
> We will cite RefLoRA [7] and LoRA-RITE [8] in the revised version of our paper. Below, we elaborate on the key differences between our algorithm and RefLoRA and LoRA-RITE, along with their potential implications, while highlighting the advantages of our approach.
>
> **Difference with LoRA-RITE:** Although LoRA-RITE achieves transformation invariance through the use of preconditioners, its underlying framework is still rooted in factorization for updating low-rank factors. LoRA-RITE introduces adaptive preconditioners and integrates first- and second-order momentum by leveraging gradients of the low-rank factors. However, its gradient updates depend on the pseudo-inverse of the low-rank factor gradients to approximate the inverse of the Hessian. Because low-rank factorization is inherently over-parameterized, these gradient matrices are frequently not invertible. Consequently, using pseudo-inverses results in the loss of gradient information along directions orthogonal to the pseudo-inverse, leading to errors that can impact both the optimization trajectory and final model performance.
>
> **Difference with RefLoRA:** RefLoRA focuses on improving low-rank adaptation by dynamically refactoring the low-rank factors $A$ and $B$ at each step using a closed-form solution to minimize an upper bound on the loss function. Still, RefLoRA remains within the factorization paradigm, optimizing the factor matrices separately, which means it can be sensitive to conditioning and imbalance issues inherent in factorized updates.
>
> This underscores a key distinction between our algorithm and factorization-based approaches. Our method directly employs the full gradients $G$ to construct the preconditioner, first-order momentum, and second-order momentum. By avoiding low-rank factorization, our gradient matrices remain invertible, enabling a more accurate and direct approximation of the inverse Hessian. This eliminates the information loss and potential inaccuracies caused by pseudo-inverses in LoRA-RITE. Leveraging the full weight matrix gradients, our approach offers potential improvements in optimization efficiency and final performance.

---

> ### Author Response · Authors · 2025-11-20
> **Response to Reviewer jxGJ: Part 3**
>
> **w5**
>
> Thank you very much for your valuable feedback on our paper. Your suggestions have greatly helped us improve our paper. We have added additional experiments and optimized the presentation in the paper.
>
> **In fact, our experiments already cover two major types of models: GPT-2 (language model) and Mix-of-Show (diffusion model).** For GPT-2, we performed experiments under different rank settings (e.g., $r=4, 16, 64$), with results shown in **Table 1** (Page 9) and **Table 2** (Page 13). Additionally, we tested models of different sizes (small and medium), with results presented in **Table 2** (Page 13). To further validate the robustness of our algorithm, we also tested its sensitivity to different learning rates, as shown in **Figure 3** (Page 14). These diverse experimental setups clearly demonstrate the effectiveness of our algorithm.To further verify the performance of our method, **we add experiments on the DART [1] dataset, with the results provided in the following table.**
>
> | Methods/Metric    | SGD  | Scaled GD | LoRA-Pro SGD | SoLoRA SGD | AdamW | Scaled AdamW | LoRA-Pro AdamW | SoLoRA AdamW |
> | ----------------- | ---- | --------- | ------------ | ---------- | ----- | ------------ | -------------- | ------------ |
> | BLEU $\uparrow$   | 41.2 | 43.8      | 44.1         | 44.6       | 43.9  | 44.8         | 44.9           | 45.4         |
> | METEOR $\uparrow$ | 0.63 | 0.66      | 0.66         | 0.66       | 0.66  | 0.67         | 0.66           | 0.67         |
> | chrF++ $\uparrow$ | 0.59 | 0.62      | 0.61         | 0.62       | 0.60  | 0.62         | 0.62           | 0.60         |
> | TER $\downarrow$  | 0.52 | 0.50      | 0.50         | 0.49       | 0.50  | 0.49         | 0.50           | 0.49         |
> | BLEURT $\uparrow$ | 0.33 | 0.38      | 0.38         | 0.39       | 0.38  | 0.40         | 0.39           | 0.40         |
>
> From the results above, we can observe that the four metrics METEOR, chrF++, TER, and BLEURT all showed no significant differences. Regarding the BLEU metric, SoLoRA outperformed the other approaches.
>
> For Mix-of-Show, we follow the setting of Preconditioned Riemannian LoRA (scaled GD), and conduct experiments within the framework of Mix-of-Show. In this field, each LoRA is trained for each single character. We conducted fine-tuning for two characters with a variety of prompts. The results are shown in **Table 5** (Page 16) and **Figures 5 to 12**. To further verify the performance of our method, **we add experiments on 4 more characters, with the results provided in the following table.**
>
> | Method/Metric    | AdamW | Scaled AdamW | LoRA-Pro AdamW | SoLoRA AdamW |
> | ---------------- | ----- | ------------ | -------------- | ------------ |
> | CLIP $\uparrow$  | 30.46 | 24.50        | 30.60          | 30.69        |
> | FID $\downarrow$ | 27.72 | 34.17        | 28.15          | 27.12        |
>
> **These supplementary experiments span different models, tasks, and datasets, collectively proving the adaptability and effectiveness of our algorithm.** However, for the experiments of even larger models, we are constrained by GPU resources (equipped with only a single Nvidia 4090), and experiments with larger models are difficult to implement.
>
> We sincerely thank you again for your feedback, which has helped us more comprehensively validate the effectiveness of our algorithm and improve the overall quality of the paper.
>
> **w6**
>
> Thank you very much for your valuable feedback on our paper. Actually, our method SoLoRA not only outperforms LoRA-Pro in metric score, but also in computational efficiency.
>
> **Comparison of AdamW methods:**
>
> In Table 2, SoLoRA AdamW improves the BLEU score to 70.0 (LoRA-Pro AdamW achieves 69.2) and the NIST score to 8.84 (LoRA-Pro AdamW achieves 8.73). Additionally, Figure 2 demonstrates that SoLoRA outperforms other methods in both training loss and training efficiency. These curves consistently show that our method achieves faster loss reduction, demonstrating its superior optimization efficiency and effectiveness.
>
> To further validate the effectiveness of SoLoRA, we plot the relationship between training time and training steps in **Figure 4 (Page 14).** The results show that SoLoRA is more time-efficient than LoRA-Pro.

---

> ### Author Response · Authors · 2025-11-20
> **Response to Reviewer jxGJ: Part 4**
>
> **Runtime of training:** We recorded the total training time in the following table for GPT-2 small model (rank 64).
>
> | Methods                     | SGD  | Scaled GD | LoRA-Pro SGD | SoLoRA SGD | AdamW | Scaled AdamW | LoRA-Pro AdamW | SoLoRA AdamW |
> | --------------------------- | ---- | --------- | ------------ | ---------- | ----- | ------------ | -------------- | ------------ |
> | Total Training Time (Hours) | 1.79 | 1.92      | 2.78         | 2.04       | 1.79  | 1.94         | 2.93           | 2.04         |
>
> From the runtime results above, it can be observed that the runtime of SoLoRA SGD is comparable to that of LoRA-Pro SGD. Compared to LoRA-Pro SGD, its runtime is reduced by $(2.78−2.04)/2.04=36.27\%$. A  similar observation can be made for AdamW-based algorithms: the runtime of our algorithm is comparable to that of Scaled AdamW, while LoRA-Pro requires $(2.93−2.04)/2.04=43.62\%$ more runtime than our method.
>
> **Runtime of inference:** We recorded the total inference time in the following table for GPT-2 small model (rank 64).
>
> | Methods                      | SGD  | Scaled GD | LoRA-Pro SGD | SoLoRA SGD | AdamW | Scaled AdamW | LoRA-Pro AdamW | SoLoRA AdamW |
> | ---------------------------- | ---- | --------- | ------------ | ---------- | ----- | ------------ | -------------- | ------------ |
> | Total Inference Time (Hours) | 1.86 | 1.87      | 1.58         | 1.89       | 1.87  | 1.88         | 1.89           | 1.86         |
>
> **w7**
>
> Thank you for your valuable comments, which have provided us with important insights and opportunities to further improve our work.
>
> **Runtime:**
>
> + Actually, we have presented the training loss as a function of training time in **Figure 1(b) and 2(b)**.
> + For further validation, we recorded the total training time and inference time in the table above. We plotted the relationship curve between training time and training steps in **Figure 4 (page 14)**.
>
> **Memory Comsumption:**
>
> + Besides, we have provided the memory complexity in **Appendix C**, the results show that SoLoRA has no overhead.
> + To further validation, we conduct the experiments of GPT-2 with rank set as 4 when conducting E2E task to record the GPU memory consumption when the optimizer is called and after the backward is called, the results is shown in the following table. The GPU memory occupied during optimizer computation of SoLoRA AdamW is smaller than LoRA-Pro AdamW. The GPU memory occupied during optimizer computation of SoLoRA SGD only have $(1401.01-1395.62)/1395.62=0.386% $increasing compare to LoRA-SGD.
>
> | Methods                                               | SGD     | scaled GD | LoRA-Pro SGD | SoLoRA SGD | AdamW   | scaled AdamW | LoRA-Pro AdamW | SoLoRA AdamW |
> | ----------------------------------------------------- | ------- | --------- | ------------ | ---------- | ------- | ------------ | -------------- | ------------ |
> | GPU Memory occupied during optimizer computation (MB) | 1395.48 | 1395.57   | 1395.62      | 1401.01    | 1396.63 | 1396.72      | 1529.97        | 1462.51      |
> | GPU Memory occupied after backward (MB)               | 1395.48 | 1395.48   | 1395.48      | 1395.62    | 1396.60 | 1396.60      | 1510.98        | 1457.12      |
>
> **w8**
>
> Thank you very much for your thoughtful and valuable comments, which have provided us with important insights and opportunities to further improve our work.
>
> + In Section 2.1, we revised the demonstration in the updated paper.
> + We revise it to "leading to high-computation cost".
> + Rank-1 approximation: Yes, the diagonal matrix $L$ and $R$ are not rank-1, but the corresponding vector multiplied together exhibit this property.Therefore, we have revised the statement to "$l_t r_t^\top$ is a rank-$1$ approximation of $G \odot G$".
>
> Thank you again for your suggestions.
>
> We sincerely appreciate the reviewer's constructive suggestions and believe that the additional experiments, analysis, and explanations significantly improve the quality of our paper. We hope that this provides sufficient reasons to raise the score.

---

> ### Author Response · Authors · 2025-11-20
> **Response to Reviewer jxGJ: Part 5**
>
> [1] Hazan, Elad, Amit Agarwal, and Satyen Kale. "Logarithmic regret algorithms for online convex optimization." *Machine Learning* 69, no. 2 (2007): 169-192.
>
> [2] Vyas, Nikhil, Depen Morwani, Rosie Zhao, Itai Shapira, David Brandfonbrener, Lucas Janson, and Sham M. Kakade. "SOAP: Improving and Stabilizing Shampoo using Adam for Language Modeling." In *The Thirteenth International Conference on Learning Representations*.
>
> [3] Frantar, Elias, Eldar Kurtic, and Dan Alistarh. "M-FAC: Efficient matrix-free approximations of second-order information." *Advances in Neural Information Processing Systems* 34 (2021): 14873-14886.
>
> [4] Morwani, Depen, Itai Shapira, Nikhil Vyas, Sham M. Kakade, and Lucas Janson. "A New Perspective on Shampoo's Preconditioner." In The Thirteenth International Conference on Learning Representations.
>
> [5] Bian, Fengmiao, Jian-Feng Cai, and Rui Zhang. "A preconditioned Riemannian gradient descent algorithm for low-rank matrix recovery." *SIAM Journal on Matrix Analysis and Applications* 45, no. 4 (2024): 2075-2103.
>
> [6] Elsayed, Mohamed, Homayoon Farrahi, Felix Dangel, and A. Rupam Mahmood. "Revisiting Scalable Hessian Diagonal Approximations for Applications in Reinforcement Learning." In *Forty-first International Conference on Machine Learning*.
>
> [7] Zhang Y, Li B, Giannakis GB. RefLoRA: Refactored Low-Rank Adaptation for Efficient Fine-Tuning of Large Models. arXiv preprint arXiv:2505.18877. 2025.
>
> [8] Yen JN, Si S, Meng Z, Yu F, Duvvuri SS, Dhillon IS, Hsieh CJ, Kumar S. LoRA Done RITE: Robust Invariant Transformation Equilibration for LoRA Optimization. arXiv preprint arXiv:2410.20625. 2024.

---

### Official Review · Reviewer_6E1X · 2025-10-30

**Soundness:** 2
**Presentation:** 2
**Contribution:** 2
**Rating:** 4
**Confidence:** 4

**Summary:**

The paper proposes SoLoRA, a low-rank fine-tuning method that uses an adaptive metric to approximate the full fine-tuning gradient, incorporating second-order information to improve efficiency and convergence.

SoLoRA achieves performance comparable to full fine-tuning with minimal additional computational cost and demonstrates superior results compared to existing low-rank methods on large language and diffusion models.

**Strengths:**

The paper addresses an important problem of improving computational efficiency of fine-tuning methods.

Convergence rates of the proposed SoLoRA were analyzed in detail.

In some of the experimental results, SoLoRA converges faster compared to the baseline LoRA.

**Weaknesses:**

There are a few major issues with the paper.

First, one of the main claims of the paper is improving computational efficiency. However, this claim has not been justified experimentally.

Second, in some results, e.g. in Fig. 3, SoLoRA converges to a larger loss compared to the other LoRAs. To support the proposed SoLoRA better, additional analyses utilizing different models in different tasks in comparison with the other state-of-the-art LoRA methods should be given.

**Questions:**

Have you compared convergence rate of your proposed method and the other LoRA variations using other LLMs/VLMs on additional benchmarks?

Could you please provide memory consumption and running times (during training and inference) in comparison with other state-of-the-art LoRA methods?

---

> ### Author Response · Authors · 2025-11-20
> **Response to Reviewer 6E1X: Part 1**
>
> We sincerely thank the reviewer for recognizing our focus on improving fine-tuning efficiency, the detailed convergence analysis of SoLoRA, and its faster convergence compared to LoRA in some experiments. Your positive feedback is incredibly encouraging and inspires us to further improve our work.
>
> **w1**
>
> We sincerely thank you for your comments, which provide us with a valuable opportunity to clarify our experimental results and further demonstrate the effectiveness of our algorithm.
>
> **Comparison of Training Time:** In fact, we have provided the training loss curve over training time when fine-tuning using different methods in **Figure 1(b) and 2(b)**. We have also supplemented the relationship between training time and training steps, shown in **Figure 4**.
>
> **Memory Consumption:** We conduct the experiments of GPT-2 with rank set as 4 when conducting E2E task to record the GPU memory consumption when the optimizer is called and after the backward is called, the results are shown in the following table. The GPU memory occupied during optimizer computation of SoLoRA AdamW is smaller than LoRA-Pro AdamW. The GPU memory occupied during optimizer computation of SoLoRA SGD only has (1401.01-1395.62)/1395.62=0.386% increase compared to LoRA-SGD. However, with this slight increase in memory, Algorithm 1 demonstrates an effective improvement, as shown in Table 2 (Page 13). For example, for the GPT-2 medium model with rank = 4, we improved the BLEU score from 66.6 to 70.3. This improvement is primarily due to the removal of dependency on the condition number of $J_\mathcal{G}(A, B)$, as explained in Sections 2.1 and 2.2, and the utilization of second-order information, resulting in better experimental performance.
>
> | Methods                                               | SGD     | scaled GD | LoRA-Pro SGD | SoLoRA SGD | AdamW   | scaled AdamW | LoRA-Pro AdamW | SoLoRA AdamW |
> | ----------------------------------------------------- | ------- | --------- | ------------ | ---------- | ------- | ------------ | -------------- | ------------ |
> | GPU Memory occupied during optimizer computation (MB) | 1395.48 | 1395.57   | 1395.62      | 1401.01    | 1396.63 | 1396.72      | 1529.97        | 1462.51      |
> | GPU Memory occupied after backward (MB)               | 1395.48 | 1395.48   | 1395.48      | 1395.62    | 1396.60 | 1396.60      | 1510.98        | 1457.12      |
>
> **w2**
>
> We sincerely thank you for your comments, which provide us with a valuable opportunity to clarify our experimental results and further demonstrate the effectiveness of our algorithm.
>
> We would like to clarify two points and provide a more systematic explanation.
>
> First, **SoLoRA converges to a smaller loss compared to the other methods.** **Figures 1 and 2** have already shown the trends of training loss with respect to training steps and training time, respectively, and the results clearly demonstrate that SoLoRA has advantages in both convergence speed and the magnitude of loss reduction. In contrast, the purpose of **Figure 3** is not to compare the final loss under optimal learning rates but to examine the stability and robustness of different methods under varying learning rate settings. Therefore, **Figure 3** should not be directly interpreted as “a performance comparison under optimal hyperparameters.”
>
> Second, regrading the experiments on different models, we have systematically evaluated the effectiveness of SoLoRA on **GPT-2 model (Section 4.1 and Appendix A)**, and take the ablation study across different model sizes, ranks, and learning rates. We also conduct experiments on **diffusion model (Appendix B)**, testing prompts on different characters, scaling factors, learning rates. These diverse experiments validate the robustness and advantages of our algorithm from multiple perspectives.
>
> Furthermore, based on your suggestion, we have added experiments on the DART dataset [1] using GPT-2 small (rank 4) to provide additional evidence across different task scenarios, enhancing the generality and persuasiveness of our conclusions. The experimental results are as follows:
>
> | Methods/Metric    | SGD  | Scaled GD | LoRA-Pro SGD | SoLoRA SGD | AdamW | Scaled AdamW | LoRA-Pro AdamW | SoLoRA AdamW |
> | ----------------- | ---- | --------- | ------------ | ---------- | ----- | ------------ | -------------- | ------------ |
> | BLEU $\uparrow$   | 41.2 | 43.8      | 44.1         | 44.6       | 43.9  | 44.8         | 44.9           | 45.4         |
> | METEOR $\uparrow$ | 0.63 | 0.66      | 0.66         | 0.66       | 0.66  | 0.67         | 0.66           | 0.67         |
> | chrF++ $\uparrow$ | 0.59 | 0.62      | 0.61         | 0.62       | 0.60  | 0.62         | 0.62           | 0.60         |
> | TER $\downarrow$  | 0.52 | 0.50      | 0.50         | 0.49       | 0.50  | 0.49         | 0.50           | 0.49         |
> | BLEURT $\uparrow$ | 0.33 | 0.38      | 0.38         | 0.39       | 0.38  | 0.40         | 0.39           | 0.40         |

---

> ### Author Response · Authors · 2025-11-20
> **Response to Reviewer 6E1X: Part 2**
>
> From the results above, we can observe that the four metrics METEOR, chrF++, TER, and BLEURT all showed no significant differences. Regarding the BLEU metric, SoLoRA outperformed the other approaches.
>
> We greatly appreciate your insightful feedback, which has helped us improve the quality of our paper.
>
> **Q1**
>
> Thank you for your valuable questions.
>
> In fact, we have **evaluated the convergence in Figure 1 (Page 8) and 2 (Page 12)**, which demonstrate the relationship between training loss and training steps/time for the GPT-2 model using the E2E dataset. The results indicate that SoLoRA achieves faster loss reduction, demonstrating optimization efficiency. Regarding additional experiments on different tasks, in **Figure 13**, we demonstrate the relationship between training loss and training steps/time for the GPT-2 model using the DART [1] dataset.
>
> For diffusion models, we performed experiments using different prompts (see **Figures 5 to 12**), scaling factors, and learning rates. Based on the consensus in [2] regarding the instability of training losses for such models, we do not use loss curves as a basis for comparison. Instead, we adopt more representative generation quality metrics (e.g., FID, CLIP score) for evaluation. Besides, we follow the setting of Preconditioned Riemannian LoRA (scaled GD) [3], and conduct experiments within the framework of Mix-of-Show [4]. In this field, each LoRA is trained for each single character. To diversify, we perform additional experiments on the other 4 characters, with the results below:
>
> | Method/Metric    | AdamW | Scaled AdamW | LoRA-Pro AdamW | SoLoRA AdamW |
> | ---------------- | ----- | ------------ | -------------- | ------------ |
> | CLIP $\uparrow$  | 30.46 | 24.50        | 30.60          | 30.69        |
> | FID $\downarrow$ | 27.72 | 34.17        | 28.15          | 27.12        |
>
> We appreciate the reviewer’s suggestion to further expand the models and benchmarks evaluated. Despite being constrained by GPU resources (equipped with only a single Nvidia 4090 graphics card), we are actively advancing the convergence comparisons on larger-scale LLMs (e.g., the LLaMA series). We plan to include corresponding loss curves and step-time comparisons in future versions and supplementary materials to further enhance the generality and extrapolation of our conclusions. If the reviewer has suggestions for specific models or benchmarks (e.g., a particular VLM or publicly available leaderboard tasks), we would be more than happy to incorporate those into our experiments and provide timely updates in our response materials.
>
> **Q2**
>
> Thank you very much for your valuable question, which helps us present the effectiveness of our algorithm more clearly. Below, we provide the memory usage and runtime of our algorithm.
>
> **The relation of training loss and training time is provided:** In **Figure 1(b) and 2(b)**, we provide the relationship between training loss and training steps/time for the GPT-2 model using the E2E dataset. These curves consistently show that our method achieves faster loss reduction, demonstrating its superior optimization efficiency and effectiveness.
>
> **Runtime of training:** We recorded the total training time in the following table of GPT-2 small model (rank 64) and plotted the relationship between training time and training steps in **Figure 4 (Page 14).**
>
> | Methods                     | SGD  | Scaled GD | LoRA-Pro SGD | SoLoRA SGD | AdamW | Scaled AdamW | LoRA-Pro AdamW | SoLoRA AdamW |
> | --------------------------- | ---- | --------- | ------------ | ---------- | ----- | ------------ | -------------- | ------------ |
> | Total Training Time (Hours) | 1.79 | 1.92      | 2.78         | 2.04       | 1.79  | 1.94         | 2.93           | 2.04         |
>
> From the runtime results above, it can be observed that the runtime of SoLoRA SGD is comparable to that of Scaled GD. Compared to LoRA-Pro SGD, its runtime is reduced by $(2.78−2.04)/2.04=36.27\%$. A  similar observation can be made for AdamW-based algorithms: the runtime of our algorithm is comparable to that of Scaled AdamW, while LoRA-Pro requires $(2.93−2.04)/2.04=43.62\%$ more runtime than our method.
>
> **Runtime of inference:** We recorded the total inference time in the following table for GPT-2 small model (rank 64).
>
> | Methods                      | SGD  | Scaled GD | LoRA-Pro SGD | SoLoRA SGD | AdamW | Scaled AdamW | LoRA-Pro AdamW | SoLoRA AdamW |
> | ---------------------------- | ---- | --------- | ------------ | ---------- | ----- | ------------ | -------------- | ------------ |
> | Total Inference Time (Hours) | 1.86 | 1.87      | 1.58         | 1.89       | 1.87  | 1.88         | 1.89           | 1.86         |

---

> ### Author Response · Authors · 2025-11-20
> **Response to Reviewer 6E1X: Part 3**
>
> **Memory consumption:** We conduct the experiments of GPT-2 with rank set as 4 when conducting E2E task to record the GPU memory consumption when the optimizer is called and after the backward is called, the results is shown in the following table. The GPU memory occupied during optimizer computation of SoLoRA AdamW is smaller than LoRA-Pro AdamW. The GPU memory occupied during optimizer computation of SoLoRA SGD only have (1401.01-1395.62)/1395.62=0.386% increasing compare to LoRA-Pro SGD.
>
> | Methods                                               | SGD     | scaled GD | LoRA-Pro SGD | SoLoRA SGD | AdamW   | scaled AdamW | LoRA-Pro AdamW | SoLoRA AdamW |
> | ----------------------------------------------------- | ------- | --------- | ------------ | ---------- | ------- | ------------ | -------------- | ------------ |
> | GPU Memory occupied during optimizer computation (MB) | 1395.48 | 1395.57   | 1395.62      | 1401.01    | 1396.63 | 1396.72      | 1529.97        | 1462.51      |
> | GPU Memory occupied after backward (MB)               | 1395.48 | 1395.48   | 1395.48      | 1395.62    | 1396.60 | 1396.60      | 1510.98        | 1457.12      |
>
> We sincerely appreciate the reviewer's constructive suggestions and believe that the additional experiments, analysis, and explanations significantly improve the quality of our paper. We hope that this provides sufficient reasons to raise the score.
>
> [1] Nan, Linyong, Dragomir Radev, Rui Zhang, Amrit Rau, Abhinand Sivaprasad, Chiachun Hsieh, Xiangru Tang et al. "Dart: Open-domain structured data record to text generation." In *Proceedings of the 2021 Conference of the North American Chapter of the Association for Computational Linguistics: Human Language Technologies*, pp. 432-447. 2021.
>
> [2] Hang, Tiankai, Shuyang Gu, Chen Li, Jianmin Bao, Dong Chen, Han Hu, Xin Geng, and Baining Guo. "Efficient diffusion training via min-snr weighting strategy." In *Proceedings of the IEEE/CVF international conference on computer vision*, pp. 7441-7451. 2023.
>
> [3] Zhang, Fangzhao, and Mert Pilanci. "Riemannian Preconditioned LoRA for Fine-Tuning Foundation Models." In *Forty-first International Conference on Machine Learning*.
>
> [4] Gu, Yuchao, Xintao Wang, Jay Zhangjie Wu, Yujun Shi, Yunpeng Chen, Zihan Fan, Wuyou Xiao et al. "Mix-of-show: Decentralized low-rank adaptation for multi-concept customization of diffusion models." *Advances in Neural Information Processing Systems* 36 (2023): 15890-15902.

---

### Official Review · Reviewer_iQCe · 2025-10-30

**Soundness:** 3
**Presentation:** 3
**Contribution:** 3
**Rating:** 4
**Confidence:** 3

**Summary:**

Overview

This paper proposes SoLoRA, a second-order low-rank adaptation method that uses an adaptive weighted metric to approximate the full fine-tuning gradient while incorporating Hessian information. The authors demonstrate performance improvements over existing LoRA variants on GPT-2 and diffusion model fine-tuning tasks.

**Strengths:**

Strengths

The paper provides a clear theoretical framework connecting low-rank fine-tuning to optimization on Riemannian manifolds. The analysis of existing methods through the lens of condition number dependence and the explicit formulation of the Jacobian operator offers useful insights into why standard LoRA methods underperform compared to full fine-tuning.

The proposed adaptive metric based on diagonal approximations of the Hessian is computationally efficient with O(mn + (m+n)r² + r³) complexity per iteration and O(m+n) memory overhead. This makes the method practical for large-scale applications while still incorporating second-order information.

Experimental results consistently show improvements across multiple settings. The GPT-2 experiments demonstrate superior performance on the E2E benchmark across different model sizes and ranks, while the diffusion model experiments show better FID scores and comparable or better CLIP scores.

The closed-form solutions for optimal low-rank updates (Theorems 3.1 and 3.2) are mathematically rigorous and provide clear guidance for implementation. The paper includes thorough supplementary materials with detailed proofs and additional experimental results.

**Weaknesses:**

Weaknesses

The theoretical contribution is somewhat incremental, primarily combining existing ideas from AdaGrad/SOAP-style adaptive metrics with the LoRA-Pro framework. While the weighted metric approach is sensible, the novelty over simply applying preconditioned optimization to LoRA factors is limited.

The experimental evaluation has several limitations. The GPT-2 experiments use only the E2E dataset with a single evaluation protocol, making it difficult to assess generalization. The performance gains, while consistent, are often modest (e.g., 70.0 vs 69.8 BLEU for rank 16). Statistical significance testing and error bars are absent throughout.

The comparison to LoRA-Pro appears incomplete since the paper claims LoRA-Pro requires solving a Sylvester equation at each step with "extremely high computational costs," yet reports runtime comparisons showing LoRA-Pro is only marginally slower than other methods (Figures 1-2). This suggests either an unfair implementation comparison or an overstatement of LoRA-Pro's computational burden.

The adaptive metric construction using diagonal approximations Lt and Rt is heuristic rather than principled. While inspired by AdaGrad/SOAP, the specific choices (element-wise summation, normalization by l1-norm, decay factors β₁ and β₂) lack theoretical justification for why this particular formulation provides a good Hessian approximation for the LoRA optimization problem.

The claim that the low-rank approximation "can be viewed as a rank-1 approximation of the Hessian" (page 5) is not rigorously justified. The connection between the Kronecker product structure of Lt and Rt and the actual Hessian of the loss function needs more careful analysis, especially given that the Hessian is with respect to the factors B and A, not the weight matrix W.

The paper introduces multiple hyperparameters (β₁, β₂, β₃, learning rates) that appear to require careful tuning based on Tables 3-4 and 6. The sensitivity analysis is limited to learning rate variations in Figure 3, and the paper does not provide clear guidance on how to set these hyperparameters in new applications.

The diffusion model experiments use relatively small-scale personalization tasks with only two characters (Harry Potter and Hermione Granger). The qualitative improvements in image diversity mentioned for lower CLIP scores at scaling factor 0.7 are not convincingly demonstrated through systematic evaluation or user studies.

**Questions:**

see weaknesses

---

> ### Author Response · Authors · 2025-11-20
> **Response to Reviewer iQCe: Part 1**
>
> We sincerely thank you for your positive feedback and recognition of the clarity of our theoretical framework, computational efficiency, and experimental effectiveness. We also deeply appreciate your valuable and detailed comments. In the following, we provide our point-by-point response and hope our response could address your concerns. We also look forward to further discussion which may further help solve the current issues.
>
> **w1**
>
> Thank you for your constructive comments. To better clarify the core value of our paper, we would like to restate the main contributions and innovations of our paper.
>
> First, we provide a new perspective on existing factorization-based algorithms of LoRA and propose a unified factorization-based update framework:
>
> $$ W_{t+1} \approx W_t - \alpha_t J_\mathcal{G} ([B_t, A_t]) J_\mathcal{G}^* ([B_t, A_t]) \nabla_{W_t} \mathcal{L}(W_0 + W_t). $$
>
> Within this framework, we analyze how the convergence rate of factorization-based algorithms is influenced by the condition number of the Jacobian operator $J_\mathcal{G} J_\mathcal{G}^* $, which in turn depends on the condition numbers of $B$ and $A$. Based on this observation, we identify a gap between factorization-based algorithms and algorithms that directly update the full weight matrix $W$, as the latter completely eliminates the dependency on the condition number of the Jacobian operator.
>
> Second, while updates based on the full weight matrix $W$ are no longer constrained by the Jacobian condition number $J_\mathcal{G} J_\mathcal{G}^* $, they are still impacted by the condition number of the Hessian $\nabla^2 \mathcal{L}(W)$. To address this, we design an adaptive metric and employ a Taylor expansion to demonstrate its ability to mitigate the influence of the condition number of the Hessian matrix on convergence. However, directly updating the full weight matrix $W$ under this metric would involve high computations and require storing the entire weight matrix, resulting in memory overhead.  To overcome this issue, we seek an optimal low-rank update $\Delta_{B}$ and $\Delta_A$ under the new adaptive metric to approximate the preconditioned gradient $H^{-1}G$. This yields the optimal updates for the two low-rank factors, which form our algorithm SoLoRA.
>
> As a result, the updates in our algorithm are directly based on the preconditioned gradient of $W$, which narrows the gap between low-rank factor updates and full weight updates. Then our algorithm eliminates the influence of the Jacobian condition number while mitigating the effects of the Hessian condition number through the adaptive metric, making our method faster and more efficient compared to direct updates in the low-rank factor space.  In contrast, methods that only precondition $A$ and $B$ can only partially reduce the influence of the Jacobian condition number, failing to fully eliminate its dependency. Moreover, such methods do not address the Hessian condition number systematically or leverage second-order information to accelerate optimization.
>
> In conclusion, we believe this analysis not only highlights the limitations of existing low-rank factorization update methods but also provides a novel framework for algorithm design: eliminating the influence of the Jacobian condition number in the full parameter space and mitigating the Hessian condition number under an adaptive metric. This framework offers a clear starting point and innovative perspective for designing new fine-tuning methods. We consider this methodologically insightful perspective to be of significant importance.
>
> **w2**
>
> Thank you very much for your thoughtful and valuable comments, which have provided us with important insights and opportunities to further improve our work. In fact, our experiments already involve two different types of models: the language model GPT-2 and the diffusion model Mix-of-Show.
>
> **GPT-2 experiments:** Since the GPU resources (equipped with only a single Nvidia 4090) are limited, we have tried our best to increase the diversity of the experiments. For GPT-2, we conduct experiments on the E2E dataset across different rank settings (e.g. r=4,16,64), with results shown in **Table 1 (Page 9) and Table 2 (Page 13)**. Additionally, we test models of varying sizes, and the corresponding results are presented in **Table 2 (Page 13)**. To further validate the robustness of SoLoRA, we also evaluate the stability under different learning rates, with results shown in **Figure 3** (Page 14). These diverse experimental setups clearly demonstrate the effectiveness of our algorithm.

---

> ### Author Response · Authors · 2025-11-20
> **Response to Reviewer iQCe: Part 2**
>
> Moreover, about the experimental results, we think there might have been a misreading. In **Table 1**, when the rank is 16, the BLEU score for SoLoRA SGD is 70.0, compared to 68.8 for Scaled GD. This indicates that SoLoRA has an advantage in improving the BLEU score for SGD-based methods. To further demonstrate the effectiveness of our algorithm, **we add additional experiments on the DART [1] dataset. The results are shown below:**
>
> | Methods/Metric    | SGD  | Scaled GD | LoRA-Pro SGD | SoLoRA SGD | AdamW | Scaled AdamW | LoRA-Pro AdamW | SoLoRA AdamW |
> | ----------------- | ---- | --------- | ------------ | ---------- | ----- | ------------ | -------------- | ------------ |
> | BLEU $\uparrow$   | 41.2 | 43.8      | 44.1         | 44.6       | 43.9  | 44.8         | 44.9           | 45.4         |
> | METEOR $\uparrow$ | 0.63 | 0.66      | 0.66         | 0.66       | 0.66  | 0.67         | 0.66           | 0.67         |
> | chrF++ $\uparrow$ | 0.59 | 0.62      | 0.61         | 0.62       | 0.60  | 0.62         | 0.62           | 0.60         |
> | TER $\downarrow$  | 0.52 | 0.50      | 0.50         | 0.49       | 0.50  | 0.49         | 0.50           | 0.49         |
> | BLEURT $\uparrow$ | 0.33 | 0.38      | 0.38         | 0.39       | 0.38  | 0.40         | 0.39           | 0.40         |
>
> From the results above, we can observe that the four metrics METEOR, chrF++, TER, and BLEURT all showed no significant differences. Regarding the BLEU metric, SoLoRA outperformed the other approaches.
>
> We also sincerely thank you for your comments regarding the lack of statistical significance tests and error bars. We fully agree with your suggestion. Following your advice, we added experiments that include error bars, and the results are shown in the table below. However, to the best of our knowledge, no related work currently provides statistical significance tests.
>
> | Method/Metric | AdamW             | Scaled AdamW      | LoRA-Pro AdamW    | SoLoRA AdamW      |
> | ------------- | ----------------- | ----------------- | ----------------- | ----------------- |
> | BLEU          | $69.1_{\pm 0.4}$  | $69.5_{\pm0.2}$   | $69.2_{\pm 0.3}$  | $70.0_{\pm 0.2}$  |
> | NIST          | $8.75_{\pm 0.02}$ | $8.80_{\pm 0.01}$ | $8.73_{\pm 0.06}$ | $8.84_{\pm 0.03}$ |
> | MET           | $46.0_{\pm 0.2}$  | $46.2_{\pm 0.1}$  | $45.9_{\pm 0.3}$  | $46.3_{\pm 0.3}$  |
> | ROUGE-L       | $70.5_{\pm 0.4}$  | $70.9_{\pm 0.2}$  | $70.8_{\pm 0.2}$  | $71.3_{\pm 0.3}$  |
> | CIDEr         | $2.47_{\pm 0.01}$ | $2.48_{\pm 0.01}$ | $2.47_{\pm 0}$    | $2.50_{\pm 0.01}$     |
>
> We greatly appreciate your insightful feedback, which has helped us improve the quality of our paper.
>
> **w3**
>
> Thank you very much for your comments. We think that simply observing the loss curves in **Figures 1 to 2 cannot directly reflect the per‑step computation time of each algorithm**. This is because, within the same training time, different algorithms execute different numbers of iterations, and the amount of loss reduction per step also differs. To provide a more intuitive comparison of computational cost, we report in the table below the total training time required for all algorithms on the GPT‑2 small model (rank 64). In addition, we **add Figure 4 (page 14)** in our paper to illustrate the relationship between training time and training steps.
>
> | Methods                     | SGD  | Scaled GD | LoRA-Pro SGD | SoLoRA SGD | AdamW | Scaled AdamW | LoRA-Pro AdamW | SoLoRA AdamW |
> | --------------------------- | ---- | --------- | ------------ | ---------- | ----- | ------------ | -------------- | ------------ |
> | Total Training Time (Hours) | 1.79 | 1.92      | 2.78         | 2.04       | 1.79  | 1.94         | 2.93           | 2.04         |
>
> The results in the table show that for SGD‑based methods, LoRA‑Pro SGD requires (2.78 − 2.04) / 2.04 = 36.27$\%$ more computation time than SoLoRA SGD. For AdamW‑based methods, LoRA‑Pro AdamW requires (2.93 − 2.04) / 2.04 = 43.62$\%$ more computation time. Furthermore, the implementation of SoLoRA and baseline methods is attached in `Supplementary Material` for reviewers to check. For baseline methods, we strictly followed the official code to ensure fairness. Detailed training logs for each method can be provided upon request.
>
> We have revised the statement to: “In comparison, LoRA‑Pro incurs higher computational overhead due to solving a Sylvester equation at each step.”  Thank you again for your suggestions regarding the clarity of our presentation. Your feedback significantly improved the precision of our paper's presentation.

---

> ### Author Response · Authors · 2025-11-20
> **Response to Reviewer iQCe: Part 3**
>
> **w4**
>
> Thank you for your suggestions regarding the clarity of our presentation. Your feedback significantly improved the precision of our paper's presentation.
>
> To better clarify the origin of the weighted metric, we would like to re-examine two existing approximations of the Hessian matrix. First, the Kronecker product $\mathbb{E}[GG^\top] \otimes \mathbb{E}[G^\top G]$ (where $G$ represents gradients with respect to per batch) can be regarded as an approximation of the Hessian matrix, a conclusion validated in [4]. Second, the element-wise product of the gradient $\mathbb{E}(G \odot G)$ similarly represents a diagonal approximation of the Hessian matrix, as employed by AdamW. The relevant theoretical analysis is detailed in [5], which demonstrates that Adam's second-order matrix estimates, acting as an adaptive diagonal preconditioner, can approximate the curvature of the loss function in Transformer models.
>
> Subsequently, Adafactor performs the row/sum summation on the element-wise gradient product $\mathbb{E}(G \odot G)$ used by AdamW to obtain $l=\mathbb{E}(\text{vec}(\text{diag}(GG^\top)))$ and $r=\mathbb{E}(\text{vec}(\text{diag}(G^\top G)))$, where $lr^\top$ is a rank-1 approximation of the element-wise gradient product $\mathbb{E}(G \odot G)$. Thus, Adafactor **employs two steps to approximate the Hessian**.
>
> In the following, we will explain how our algorithm approximates the Hessian using diagonal matrices $L$ and $R$. Since shampoo computes matrix multiplication, this results in high computational costs.  Therefore, we attempt to leverage Adafactor's simplicity by computing only the diagonal elements of $GG^\top$ and $G^\top G$, this can be realized by performing row/column summation on $G \odot G$. Therefore, in our algorithm, the weighted metric $L$ and $R$ also employ two steps to approximate the Hessian.
>
> **w5**
>
> Thank you very much for your comments.
>
> First, we need to clarify that the Hessian we analyzed is exactly with respect to the full weight matrix $W$. Regarding the statement, we have revised the statement to “$l_t r_t^\top$ is a rank-$1$ approximation of $G \odot G$" as stated in Adafactor [6].
>
> Thank you again for your suggestions regarding the clarity of our presentation. Your feedback significantly improved the precision of our paper's presentation.
>
> **w6**
>
> Thank you very much for your suggestions regarding the hyperparameter settings. We agree that our algorithm involves tuning the four hyperparameters you mentioned. However, to the best of our knowledge, all baseline methods (SGD, AdamW, Scaled GD, Scaled AdamW, and LoRA-Pro) require hyperparameter optimization on different tasks (i.e., first-order momentum parameter $\beta_3$, second-order momentum parameters $\beta_1 = \beta_2$, and the learning rate). In all experiments covered in our paper, we adjusted these parameters while fixing the remaining ones to achieve optimal performance across all methods, thereby ensuring fairness.
>
> Tables 3, 4, and 6 present all hyperparameters used in the experiments of our paper. The purpose is to enable readers to reproduce our work, while also fulfilling the `Reproducibility Statement` requirement of the ICLR 2026.
>
> In fact, **Figures 3(a) and 3(b)** illustrate the stability of SoLoRA under different learning rates, and the results clearly demonstrate that our algorithm does not require overly meticulous fine-tuning of the learning rate.
>
> We provide the implementation of SoLoRA and baseline methods in `Supplementary Material` for reproducing. For similar tasks, it is not necessary to perform extensive hyperparameter tuning. The default settings for these four parameters are demonstrated in the code: $\beta_1=\beta_2=0.98, \beta_3=0.9, \text{learning rate}=1e-3$.

---

> ### Author Response · Authors · 2025-11-20
> **Response to Reviewer iQCe: Part 4**
>
> **w7**
> Thank you very much for your suggestion. It is highly valuable for enriching our experimental results, improving the structure of our paper, and providing a more comprehensive validation of our algorithm's effectiveness.
>
> For the setting of diffusion model experiments, we follow the setting of Preconditioned Riemannian LoRA (scaled GD) [2], and conduct experiments within the framework of Mix-of-Show [3]. In this field, each LoRA is trained for each single character.
>
> We diversified the experiments as much as possible, conducting across 8 prompts, containing different backgrounds, different clothing styles, and different framing sizes.  We also perform ablation studies across different scaling factors and learning rates to validate the effectiveness of SoLoRA.
>
> **Additional experiments:** We provide the additional experimental results of the other four characters in the following table.
>
> | Method/Metric    | AdamW | Scaled AdamW | LoRA-Pro AdamW | SoLoRA AdamW |
> | ---------------- | ----- | ------------ | -------------- | ------------ |
> | CLIP $\uparrow$  | 30.46 | 24.50        | 30.60          | 30.69        |
> | FID $\downarrow$ | 27.72 | 34.17        | 28.15          | 27.12        |
>
> We sincerely appreciate the reviewer's constructive suggestions and believe that the additional experiments, analysis, and explanations significantly improve the quality of our paper. We hope that this provides sufficient reasons to raise the score.
>
> [1] Nan, Linyong, Dragomir Radev, Rui Zhang, Amrit Rau, Abhinand Sivaprasad, Chiachun Hsieh, Xiangru Tang et al. "Dart: Open-domain structured data record to text generation." In *Proceedings of the 2021 Conference of the North American Chapter of the Association for Computational Linguistics: Human Language Technologies*, pp. 432-447. 2021.
>
> [2] Zhang, Fangzhao, and Mert Pilanci. "Riemannian Preconditioned LoRA for Fine-Tuning Foundation Models." In *Forty-first International Conference on Machine Learning*.
>
> [3] Gu, Yuchao, Xintao Wang, Jay Zhangjie Wu, Yujun Shi, Yunpeng Chen, Zihan Fan, Wuyou Xiao et al. "Mix-of-show: Decentralized low-rank adaptation for multi-concept customization of diffusion models." *Advances in Neural Information Processing Systems* 36 (2023): 15890-15902.
>
> [4] Morwani, Depen, Itai Shapira, Nikhil Vyas, Sham M. Kakade, and Lucas Janson. "A New Perspective on Shampoo's Preconditioner." In The Thirteenth International Conference on Learning Representations.
>
> [5] Zhang, Yushun, Congliang Chen, Tian Ding, Ziniu Li, Ruoyu Sun, and Zhiquan Luo. "Why transformers need adam: A hessian perspective." *Advances in neural information processing systems* 37 (2024): 131786-131823.
>
> [6] Shazeer, Noam, and Mitchell Stern. "Adafactor: Adaptive learning rates with sublinear memory cost." In *International Conference on Machine Learning*, pp. 4596-4604. PMLR, 2018.

---

### Official Review · Reviewer_i9ma · 2025-10-30

**Soundness:** 2
**Presentation:** 2
**Contribution:** 2
**Rating:** 2
**Confidence:** 5

**Summary:**

The paper studies LLM fine-tuning with LoRA, which receives great attention recent years due to its efficacy and efficiency for fine-tuning LLMs to adapt to downstream tasks. Although LoRA works well in general, there is still some potential gap compared to the performance of full parameter fine-tuning. Motivated from the previous paper, LoRA-Pro, this work proposes to let LoRA updates match with the updates using an approximation of Newton's method on the full model. The new algorithm, termed SoLoRA, achieves better empirical performance on fine-tuning GPT-2.

**Strengths:**

The paper provides a good summary of related works on LoRA using their framework. The derivation of the proposed algorithm is explained in details and mathematically consistent. The studied topic is well-motivated and is interesting to the general community.

**Weaknesses:**

1. Although the authors call their algorithm as SoLoRA, it is not a second-order method. There are lots of approximations happening in the derivation, and the Hessian is replaced by $L$ and $R$. The authors say in the paper that such rank-1 approximations of Hessian are optimal w.r.t. the generalized KL divergence and refer to the AdaFactor paper. This statement of the authors is wrong and very misleading. Indeed, what AdaFactor paper says is that $L$ and $R$ are optimal rank-1 approximations of the second moments of the gradients $G_t^2$, **but not the Hessian matrix**. There is still a gap between $G_t^2$ and the true Hessian. In this regard, the algorithm is approximating preconditioned GD with $G_t^2$ but not Newton. Therefore, it should not be called as second-order. Although there exists some work trying to approximate Hessian with Fisher information, when and why this approximation makes sense remain unclear for LLMs.

2. The major weakness of the algorithm is that the full gradients $G_t$ are always required. In Algorithm 1, $G_t$ is used to update $l_t$ and $r_t$. In Algorithm 2, $G_t$ is required to maintain the momentum $M_t$. In fact, the computation of $G_t$ is exactly what LoRA tries to avoid. It increases the memory for optimizer states and also activation storage for backpropagation. By only updating using gradients of $A$ and $B$, the savings of LoRA are made possible. In fact, according to the motivation that there is still a gap between LoRA and full parameter fine-tuning, if $G_t$, **the full gradient, is already computed, why not just do full parameter fine-tuning using it**? Moreover, the storage of the momentum $M_t$ also increases memory compared to the original LoRA, from $(m+n)r$ to $mn$.

3. The experiments are too limited. Only one model, GPT-2, is tested on a single task, E2E. What about other model family? How about even larger models? What about other tasks? The current experiments do not suffice to show that the new method achieves good performance. Also, the paper claims that their method adds no overhead, which is not verified. What are the memory consumptions compared to LoRA, say for example on the considered GPT-2 with E2E, and other larger models with different tasks?

**Questions:**

Other questions not asked in weaknesses:

1. In the original LoRA-Pro derivation, there is a constraint $dL\leq0$. Can authors provide some details why this is dropped in this paper?

2. Why are $B^\top L^{1/2}B$ and $AR^{1/2}A^\top$ invertible? In particular, since $B_1=0$, what is $(B_1^\top L_1^{1/2}B_1)^{-1}$?

3. Are there typos in line 8 of Algorithm 2? Should be $(1-\lambda\eta) B$ and also for A?

---

> ### Author Response · Authors · 2025-11-20
> **Response to Reviewer i9ma: Part 1**
>
> Thank you for your valuable and detailed comments! We are delighted that that you found our summary of related works on LoRA comprehensive and well-structured, and that the detailed explanation and mathematically consistency of our algorithm were recognized. In the following, we provide our point-by-point response and hope our response could address your concerns. We also look forward to the subsequent discussion which may further help solve the current issues.
>
> **w1** Thank you for your insightful question, which helps us improve the presentation of our paper.
>
> In fact, we employed the second-order information to construct the weighted metric, but did not label it as a second-order method. To better clarify the core perspective of our paper, we would like to re-examine two existing approximations of the Hessian matrix. First, the Kronecker product $\mathbb{E}[GG^\top] \otimes \mathbb{E}[G^\top G]$ (where $G$ represents gradients with respect to per batch) can be regarded as an approximation of the Hessian matrix, a conclusion validated in [2]. Second, the element-wise product of the gradient $\mathbb{E}(G \odot G)$ similarly represents a diagonal approximation of the Hessian matrix, as employed by AdamW. The relevant theoretical analysis is detailed in [3], which demonstrates that AdamW's second-order matrix estimates, acting as an adaptive diagonal preconditioner, can approximate the curvature of the loss function in Transformer models.
>
> Subsequently, Adafactor performs the row/sum summation on the element-wise gradient product $\mathbb{E}(G \odot G)$ used by AdamW to obtain $l=\mathbb{E}(\text{vec}(\text{diag}(GG^\top)))$ and $r=\mathbb{E}(\text{vec}(\text{diag}(G^\top G)))$, where $lr^\top$ is a rank-1 approximation of the element-wise gradient product $\mathbb{E}(G \odot G)$. Thus, Adafactor **employs two steps to approximate the Hessian**.
>
> In the following, we will explain how our algorithm approximates the Hessian using the diagonal matrix $L$ and $R$. Since shampoo computes matrix multiplication, this results in high computational costs.  Therefore, we attempt to leverage Adafactor's simplicity by computing only the diagonal elements of $GG^\top$ and $G^\top G$, this can be realized by performing row/column summation on $G \odot G$. Therefore, in our algorithm, the weighted metric $L$ and $R$ also employ two steps to approximate the Hessian.
>
> Regarding the Newton direction, for a twice differentiable function $f: \mathbb{R} \rightarrow \mathbb{R}$, the Newton direction is $f'(x)/f''(x)$. For high dimension cases, the newton direction is $H^{-1}\nabla f(x)=(\nabla^2 f(x))^{-1}\nabla f(x)$. Since the high computation cost of Hessian, some methods approximate the Hessian by $G G^\top$, as described in [4], which proposes an approximate Newton step using the inverse of the accumulated matrix $G G^\top$, applied to a vector constructed from gradient terms. Consequently, the optimal solution presented in Equation 6 of our paper represents an approximation of the Newton direction.
>
> Regarding the relation between Fisher information and Hessian approximations, [5] demonstrates that the Hessian of the negative log-likelihood loss equals the Fisher Information Matrix (FIM) under regularity conditions, making FIM a natural positive semi-definite approximation of curvature.
>
> We greatly appreciate your insightful feedback, which has helped us improve the quality of our paper.
>
> **w2**
>
> **The computation of $G_t$:** The baseline methods vanilla LoRA, Preconditioned Riemannian LoRA (scaled GD), and LoRA-Pro are all require computing the gradient $G_t$ during the backward process due to the chain rule.
>
> **Regarding the memory usage of Algorithm 1:** In Algorithm 1 (SoLoRA SGD), we only need to store two vectors $l_t$ and $r_t$, resulting in a memory complexity of $O(m+n)$. To further validate this, we record GPU memory consumption when the optimizer is called and after the backward is called (fine-tune GPT-2 small model with rank as 4). The results are summarized below.
>
> | Methods                                               | SGD     | scaled GD | LoRA-Pro SGD | SoLoRA SGD | AdamW   | scaled AdamW | LoRA-Pro AdamW | SoLoRA AdamW |
> | ----------------------------------------------------- | ------- | --------- | ------------ | ---------- | ------- | ------------ | -------------- | ------------ |
> | GPU Memory occupied during optimizer computation (MB) | 1395.48 | 1395.57   | 1395.62      | 1401.01    | 1396.63 | 1396.72      | 1529.97        | 1462.51      |
> | GPU Memory occupied after backward (MB)               | 1395.48 | 1395.48   | 1395.48      | 1395.62    | 1396.60 | 1396.60      | 1510.98        | 1457.12      |
> | Memory Complexity                                     | 0       | 0         | 0            | m+n        | (m+n)r  | (m+n)r       | 2mn            | mn+m+n       |

---

> ### Author Response · Authors · 2025-11-20
> **Response to Reviewer i9ma: Part 2**
>
> The results confirm that the memory usage of Algorithm 1 is comparable to other algorithms. Specifically, SoLoRA SGD (Algorithm 1) increases memory usage by only (1401.01-1395.62)/1395.62 = $0.386\%$ compared to LoRA-SGD. However, with this slight increase in memory, Algorithm 1 demonstrates an effective improvement, as shown in Table 2 (Page 13). For example, for the GPT-2 medium model with rank = 4, we improved the BLEU score from 66.6 to 70.3. This improvement is primarily due to the removal of dependency on the condition number of $J_\mathcal{G}(A, B)$, as explained in Sections 2.1 and 2.2, and the utilization of second-order information, resulting in better experimental performance.
>
> **Regarding the memory usage of Algorithm 2:** For Algorithm 2 (based on Adam-like optimization methods), we acknowledge that all gradient-based momentum optimization algorithms require additional memory to store first-order momentum $M_t$. Compared to LoRA, which stores the low-rank momentum, our algorithm stores the first-order momentum for $G_t$. It is worth noting that, if memory consumption is a primary concern, we could alternatively use the moving average of the low-rank factor gradients as the first-order momentum to reduce memory usage. However, we chose not to do so because the essence of our algorithm is to approximate the Hessian of the entire matrix $W$ rather than the Hessian of its low-rank factors $B$ and $A$. The Hessian of low-rank factors is non-invertible, and calculating their pseudoinverse introduces parameter redundancy, leading to errors. Therefore, we construct the first-order momentum using the full gradient $G_t$. The GPU memory usage recorded in the table above shows that we incur a moderate increase in memory usage compared to LoRA. However, we believe that this trade-off between memory and performance is acceptable. By sacrificing a small amount of memory, we achieve notable performance improvements, as demonstrated in Table 2 (Page 13). For instance, on GPT-2 medium with the E2E task (BLEU scores), our method improves AdamW (Vanilla LoRA) from 68.9 to 70.3.
>
> Totally speaking, the memory complexity of SoLoRA SGD (Alg 1) and SoLoRA AdamW (Alg 2) is m+n and mn+m+n, respectively. The above table shows the GPU memory occupation of SoLoRA does not have a significant growth than baseline methods. The most important is that we got a better performance in GPT-2 and diffusion model experiments. Figure 1, 2 shows the relation of training loss and step, training loss and time, both showing the effectiveness of SoLoRA. The implementation of SoLoRA and baseline methods is attached in `Supplementary Material` for reviewers to check.
>
> **w3**
>
> Thank you very much for your valuable feedback on our paper. Your suggestions have greatly helped us improve our paper. We have added additional experiments and optimized the presentation in the paper.
>
> **In fact, our experiments already cover two major types of models: GPT-2 (language model) and Mix-of-Show (diffusion model).** For GPT-2, we performed experiments under different rank settings (e.g., $r=4, 16, 64$), with results shown in **Table 1** (Page 9) and **Table 2** (Page 13). Additionally, we tested models of different sizes (small and medium), with results presented in **Table 2** (Page 13). To further validate the robustness of our algorithm, we also tested its sensitivity to different learning rates, as shown in **Figure 3** (Page 14). These diverse experimental setups clearly demonstrate the effectiveness of our algorithm.To further verify the performance of our method, **we add experiments on the DART [1] dataset, with the results provided in the following table.**
>
> | Methods/Metric    | SGD  | Scaled GD | LoRA-Pro SGD | SoLoRA SGD | AdamW | Scaled AdamW | LoRA-Pro AdamW | SoLoRA AdamW |
> | ----------------- | ---- | --------- | ------------ | ---------- | ----- | ------------ | -------------- | ------------ |
> | BLEU $\uparrow$   | 41.2 | 43.8      | 44.1         | 44.6       | 43.9  | 44.8         | 44.9           | 45.4         |
> | METEOR $\uparrow$ | 0.63 | 0.66      | 0.66         | 0.66       | 0.66  | 0.67         | 0.66           | 0.67         |
> | chrF++ $\uparrow$ | 0.59 | 0.62      | 0.61         | 0.62       | 0.60  | 0.62         | 0.62           | 0.60         |
> | TER $\downarrow$  | 0.52 | 0.50      | 0.50         | 0.49       | 0.50  | 0.49         | 0.50           | 0.49         |
> | BLEURT $\uparrow$ | 0.33 | 0.38      | 0.38         | 0.39       | 0.38  | 0.40         | 0.39           | 0.40         |
>
> The four metrics METEOR, chrF++, TER, and BLEURT all showed no significant differences. Regarding the BLEU metric, SoLoRA outperformed the other approaches.

---

> ### Author Response · Authors · 2025-11-20
> **Response to Reviewer i9ma: Part 3**
>
> For Mix-of-Show, we follow the setting of Preconditioned Riemannian LoRA (scaled GD), and conduct experiments within the framework of Mix-of-Show. In this field, each LoRA is trained for each single character. We conducted fine-tuning for two characters with a variety of prompts. The results are shown in **Table 5** (Page 16) and **Figures 5 to 12**. To further verify the performance of our method, **we add experiments on 4 more characters, with the results provided in the following table.**
>
> | Method/Metric    | AdamW | Scaled AdamW | LoRA-Pro AdamW | SoLoRA AdamW |
> | ---------------- | ----- | ------------ | -------------- | ------------ |
> | CLIP $\uparrow$  | 30.46 | 24.50        | 30.60          | 30.69        |
> | FID $\downarrow$ | 27.72 | 34.17        | 28.15          | 27.12        |
>
> **These supplementary experiments span different models, tasks, and datasets, collectively proving the adaptability and effectiveness of our algorithm.** However, for the experiments of even larger models, we are constrained by GPU resources (equipped with only a single Nvidia 4090), experiments with larger models are difficult to implement.
>
> **Regarding memory consumption**, we record the GPU memory in the table above. From these results, it can be observed that, for SGD-based algorithms, the memory consumption of our algorithm is comparable to that of other methods. For AdamW-based algorithms, while our method does incur slightly higher memory usage, it also achieves performance improvements. For example, as shown in Table 2 (Page 13), for the GPT-2 medium model tested on the E2E task (BLEU score), our method improves the performance of AdamW (Vanilla LoRA) from 68.9 to 70.3. This performance gain highlights the effective trade-off between memory and performance achieved by our method.
>
> We sincerely thank you again for your feedback, which has helped us more comprehensively validate the effectiveness of our algorithm and improve the overall quality of the paper.
>
> **Q1**
>
> In fact, we did not overlook this constraint. By combining the approximate preconditioned gradient and the balance constraint, we have already determined the specific form of our algorithm. After the form of the algorithm was derived, we can verify that it also satisfies the constraint $d\mathcal{L} \leq 0$. As a result, the loss function of our algorithm is guaranteed to decrease. Since this constraint does not affect the derivation process of the algorithm, we omitted it in the main text for the sake of brevity. However, we confirm that SoLoRA also satisfies $d\mathcal{L} \leq 0$. **The proof has now been included in Appendix D.4 of the updated paper.**
>
> **Q2**
>
> Thank you very much for raising this insightful question, which has provided an excellent opportunity to further clarify the theoretical assumptions underlying our algorithm. Consistent with the settings adopted in LoRA‑Pro, our derivations and algorithmic design rely on the assumption that both $B$ and $A$ are full‑rank matrices. Under this assumption, $B^\top L^{1/2} B$ and $A R^{1/2} A^\top$ are positive definite on their respective subspaces, thereby guaranteeing their invertibility.
>
> **Proof of $B^\top L^{\frac{1}{2}}B$  is invertible:**  For any non-zero vector $x \in \mathbb{R}^{r}$, $x^\top B^\top L^{\frac{1}{2}}Bx = \langle Bx,Bx \rangle_H = \|Bx \|_H^2 > 0$. Thus, $B^\top L^{\frac{1}{2}}B$ is a positive definite matrix and is therefore invertible. The proof for the invertibility of $A R^{\frac{1}{2}}A^\top$ is similar.
>
> **Regarding the specific case where $B_1=0$:** we introduce a small regularization term $\epsilon I$ to the matrix $B^\top L^{\frac{1}{2}}B$, which ensures numerical stability and prevents singularity. In the experiments, we set $\epsilon = 1e-6$. This technique is also employed in Scaled GD and LoRA-Pro.
>
> We have revised the paper to include a more detailed explanation of these assumptions and experimental settings. The implementation of SoLoRA and baseline methods is attached in `Supplementary Material` for reviewers to check. We sincerely appreciate your thoughtful feedback again, which has been instrumental in improving the clarity of our presentation.
>
> **Q3**
>
> Thank you very much for your careful reading of our paper. This is a typo, and we have made the correction to line 8 of Algorithm 2 (Page 8). Thank you again for pointing this out.
>
> We sincerely appreciate the reviewer's constructive suggestions and believe that the additional experiments, analysis, and explanations significantly improve the quality of our paper. We hope that this provides sufficient reasons to raise the score.

---

> ### Author Response · Authors · 2025-11-20
> **Response to Reviewer i9ma: Part 4**
>
> [1] Nan, Linyong, Dragomir Radev, Rui Zhang, Amrit Rau, Abhinand Sivaprasad, Chiachun Hsieh, Xiangru Tang et al. "Dart: Open-domain structured data record to text generation." In *Proceedings of the 2021 Conference of the North American Chapter of the Association for Computational Linguistics: Human Language Technologies*, pp. 432-447. 2021.
>
> [2] Morwani, Depen, Itai Shapira, Nikhil Vyas, Sham M. Kakade, and Lucas Janson. "A New Perspective on Shampoo's Preconditioner." In The Thirteenth International Conference on Learning Representations.
>
> [3] Zhang, Yushun, Congliang Chen, Tian Ding, Ziniu Li, Ruoyu Sun, and Zhiquan Luo. "Why transformers need adam: A hessian perspective." *Advances in neural information processing systems* 37 (2024): 131786-131823.
>
> [4] Hazan, Elad, Amit Agarwal, and Satyen Kale. "Logarithmic regret algorithms for online convex optimization." *Machine Learning* 69, no. 2 (2007): 169-192.
>
> [5] Barshan, Elnaz, Marc-Etienne Brunet, and Gintare Karolina Dziugaite. "Relatif: Identifying explanatory training samples via relative influence." In *International Conference on Artificial Intelligence and Statistics*, pp. 1899-1909. PMLR, 2020.

---

### Author Response · Authors · 2025-12-03
**Final Remarks**

Dear AC,

We thank you and all the reviewers for the time. We summarize the key points below.

| Reviewer | Strength                                                     | Questions                                                    | Our reply                                                    |
| -------- | ------------------------------------------------------------ | ------------------------------------------------------------ | ------------------------------------------------------------ |
| i9ma     | A good summary, mathematically consistent, and well-motivated. | **1.** Why SoLoRA is a second-order method. **2.** The full gradient is required. **3.** Experiments are limited. **4.** A proof of loss reduction is dropped. **5.** Why the matrices $B^\top L^{\frac{1}{2}}B$ and $A R^{\frac{1}{2}}A^\top$ are invertible. **6.** A typo. | **1.** Existing works approximate Hessian by Kronecker approximation or diagonal approximation (element-wise product of gradient). Our method uses these both. **2.** Memory usage is provided, which verifies SoLoRA have no significant overhead. **3.** We have conducted experiments on GPT-2 with different rank, different model size, different learning rates, and additional datasets. We also extended diffusion model to more characters. Both follow and extended the setting of existing papers. **4.** In Appendix D.4, we verify that SoLoRA satisfies the constraint $d \mathcal{L}<0$ to guarantee the loss exhibits a decreasing trend. **5.** The matrices $B^\top L^{\frac{1}{2}}B$ and $A R^{\frac{1}{2}}A^\top$ are positive definite and thus invertible. **6.** Corrected. |
| iQCe     | Clear theoretical framework, computationally efficient, and mathematically rigorous. | **1.** Novelty is limited. **2.** Experiments is limited. **3.** Comparison of runtime is incomplete. **4.** The approximation of Hessian lack theoretical justification. **5.** Hyperparameters  require careful tuning. **6.** Characters of diffusion model are limited. | **1.** We design an adaptive metric and employ a Taylor expansion to demonstrate its ability to mitigate the influence of the condition number of the Hessian matrix on convergence. **2.** We have conducted experiments on GPT-2 with different rank, different model size, different learning rates, and additional datasets. We also extended diffusion model to more characters. Both follow and extended the setting of existing papers. **3.** The comparison of runtime are provided to support the efficiency of our method. **4.** Existing works approximate Hessian by Kronecker approximation or diagonal approximation (element-wise product of gradient). Our method uses these both. **5.** Our method require 3 hyperparameters, same as baseline methods. **6.** Following existing practice, each character require training a separate LoRA model. We have extended experiments to 6 characters under various prompts. |
| 6E1X     | Computational efficiency, convergence faster.                | **1.** Computational efficiency is not be justified experimentlly. **2.** Experiments of additional models/tasks should be given. **3.** Please provide memory consumption and  runtime. | **1&3.** The training loss -- time curve was provided in the paper. The memory consumption and runtime are supplemented in the response & revision, both showing the efficiency of our method. **2.** We have conducted experiments on GPT-2 with different rank, different model size, different learning rates, and additional datasets. We also extended diffusion model to more characters. Both follow and extended the setting of existing papers. |
| jxGJ     | Generalize existing methods, improved performance.           | **1.** What is the motivation. **2.** The specific design of SoLoRA  are not justified. **3.** The distinction between SoLoRA and existing works are unclear. Some related works are not cited. **4.** Experiments is limited. **5.** Performance gain is limited. **6.** Please provide memory consumption and  runtime. **7.** Some writing typos. | **1.** Existing methods are affected by the condition number of Jacobian and Hessian matrix. Through second-order Taylor expansion of the loss function, we show the weighted metric is crucial to solve these issues. **2.** We rephrased the rationale of our design in the full response for clarity. **3.** Our method directly employs the full gradients, which is invertible and enable a more accurate and direct approximation of the inverse Hessian. **4.** We have conducted experiments on GPT-2 with different rank, different model size, different learning rates, and additional datasets. We also extended diffusion model to more characters. Both follow and extended the setting of existing papers. **5.** The experimental results show our method not only outperforms baseline in metric score, but also in computational efficiency. **6.** The memory consumption and runtime are supplemented, both showing significant efficiency improvements over LoRA-Pro. **7.** Corrected. |

---

### Meta-Review · Area_Chair_Ep1y · 2025-12-16

**Summary:**

The paper studies LLM fine-tuning with LoRA. Motivated by LoRA-Pro, this work proposes to let LoRA updates match with the updates using an approximation of Newton's method on the full model. The new algorithm, termed SoLoRA, achieves better empirical performance on fine-tuning GPT-2. However, the novelty is limited and the experiments are still limited after the rebuttal, as pointed out by several reivewers. I suggest to reject this paper.

Btw, the authors provide a very nice summary about the comments, and the response.

**Reviewer Concerns:**

Reviewer i9ma concerns the fake of second-order methods, the full gradient is required at each step, and the limited experiments. The first issue was addressed by the rebuttal, but the last two issues still exist.

Reviewer iQCe concerns the limited novelty and experiments. The rebuttal partially addressed a bit but not fully addressed.

Reivewer 6E1X concerns the experimental parts, e.g., incomplete comparison, computational cost. The rebuttal addressed most of them.

Reviewer jxGJ cares about the motivation of this work (as well as the algorithm design), limited experiments and performance. The rebuttal cannot fully address these issues.

**Reviewer Scores:**

Reviewer i9ma could increase the score and wouldn't give positive support.

Reviewer iQCe may keep the socre unchanged.

Reivewer 6E1X would increase the score.

Reviewer jxGJ may not change their socre.

---

### Decision · Program_Chairs · 2026-01-26

Reject